# Paternal microbiome perturbations impact offspring fitness

Ayele Argaw-Denboba[1], Thomas S. B. Schmidt[2], Monica Di Giacomo[1], Bobby Ranjan[1], Saravanan Devendran[2], Eleonora Mastrorilli[2], Catrin T. Lloyd[1], Danilo Pugliese[1], Violetta Paribeni[1], Juliette Dabin[1], Alessandra Pisaniello[1], Sergio Espinola[1], Alvaro Crevenna[1], Subhanita Ghosh[3,4], Neil Humphreys[1], Olga Boruc[1], Peter Sarkies[3,4], Michael Zimmermann[2], Peer Bork[2,5,6] & Jamie A. Hackett[1✉]

The gut microbiota operates at the interface of host–environment interactions to influence human homoeostasis and metabolic networks[1–4]. Environmental factors that unbalance gut microbial ecosystems can therefore shape physiological and disease-associated responses across somatic tissues[5–9]. However, the systemic impact of the gut microbiome on the germline—and consequently on the $F_1$ offspring it gives rise to—is unexplored[10]. Here we show that the gut microbiota act as a key interface between paternal preconception environment and intergenerational health in mice. Perturbations to the gut microbiota of prospective fathers increase the probability of their offspring presenting with low birth weight, severe growth restriction and premature mortality. Transmission of disease risk occurs via the germline and is provoked by pervasive gut microbiome perturbations, including non-absorbable antibiotics or osmotic laxatives, but is rescued by restoring the paternal microbiota before conception. This effect is linked with a dynamic response to induced dysbiosis in the male reproductive system, including impaired leptin signalling, altered testicular metabolite profiles and remapped small RNA payloads in sperm. As a result, dysbiotic fathers trigger an elevated risk of in utero placental insufficiency, revealing a placental origin of mammalian intergenerational effects. Our study defines a regulatory 'gut–germline axis' in males, which is sensitive to environmental exposures and programmes offspring fitness through impacting placenta function.

Sperm transmit heritable information to the next generation in the form of genetic (DNA sequence) and epigenetic (non-DNA sequence-based) material[11,12]. Evidence across phyla indicates that the epigenetic component, including chromatin states, small RNAs and macromolecules, has the potential to be modified by the preconception environment and influence offspring phenotype[13–21]. Nonetheless, the extent and underlying mechanisms of paternally inherited epigenetic effects in mammals remain opaque[22], while how environmental factors converge and signal to germ cells is also unclear. The gut microbiota is increasingly appreciated to play a principal role in integrating environmental signals into host responses[1,9]. Indeed, the mammalian lifecycle has evolved in the presence of gut microbial communities, which have assumed an important role in metabolic, hormonal and immune function, implying that hosts exploit a healthy microbiome to maximize fitness. The composition of gut microbial ecosystems is nonetheless profoundly shaped by environmental factors, such as diet and medication[23–25], and thus loss of gut biodiversity associated with modernization poses a risk to human health. Yet, despite accumulating evidence that an imbalance (dysbiosis) of the microbiome triggers physiological responses across somatic tissues[5–8,26–29], little is known of the effect of microbiome perturbations on the germline.

To investigate this, we established an inducible model of gut microbiota dysbiosis in isogenic male mice using ad lib non-absorbable antibiotics (nABX) (Fig. 1a). These nABX cannot cross the gastrointestinal epithelium and thus any reproductive responses reflect acute perturbation of gut microbial communities rather than systemic drug exposure[30]. Applying 16S ribosomal RNA sequencing showed that 6 weeks of low-dose nABX treatment (denoted 6 wk) led to a marked reduction in diversity, abundance and richness of gut microbiota ($P = 0.000003$, Wilcoxon rank sum), which is reversible and progressively recovers after 8 weeks of nABX withdrawal (Fig. 1b and Extended Data Fig. 1a,b). The dysbiosis after 6 weeks of nABX had no significant effects on male weight, fertility or survival (Extended Data Fig. 1c–h). Moreover, nABX were undetectable in circulating serum and testis using mass spectrometry, confirming their specific action in the gut (Extended Data Fig. 1i–l).

### Offspring of dysbiotic fathers

To assess the impact of induced microbiota dysbiosis on offspring, we mated nABX-treated males with naive (untreated) females and scored $F_1$ phenotypes, applying a rigorous nested statistical analysis for intergenerational significance. We found that offspring sired by

[1]European Molecular Biology Laboratory (EMBL), Epigenetics & Neurobiology Unit, Rome, Italy. [2]European Molecular Biology Laboratory (EMBL), Structural & Computational Biology Unit, Heidelberg, Germany. [3]MRC London Institute for Medical Science (LMS), London, UK. [4]Department of Biochemistry, University of Oxford, Oxford, UK. [5]Department of Bioinformatics, Biozentrum, University of Würzburg, Würzburg, Germany. [6]Yonsei Frontier Lab (YFL), Yonsei University, Seoul, South Korea. ✉e-mail: jamie.hackett@embl.it

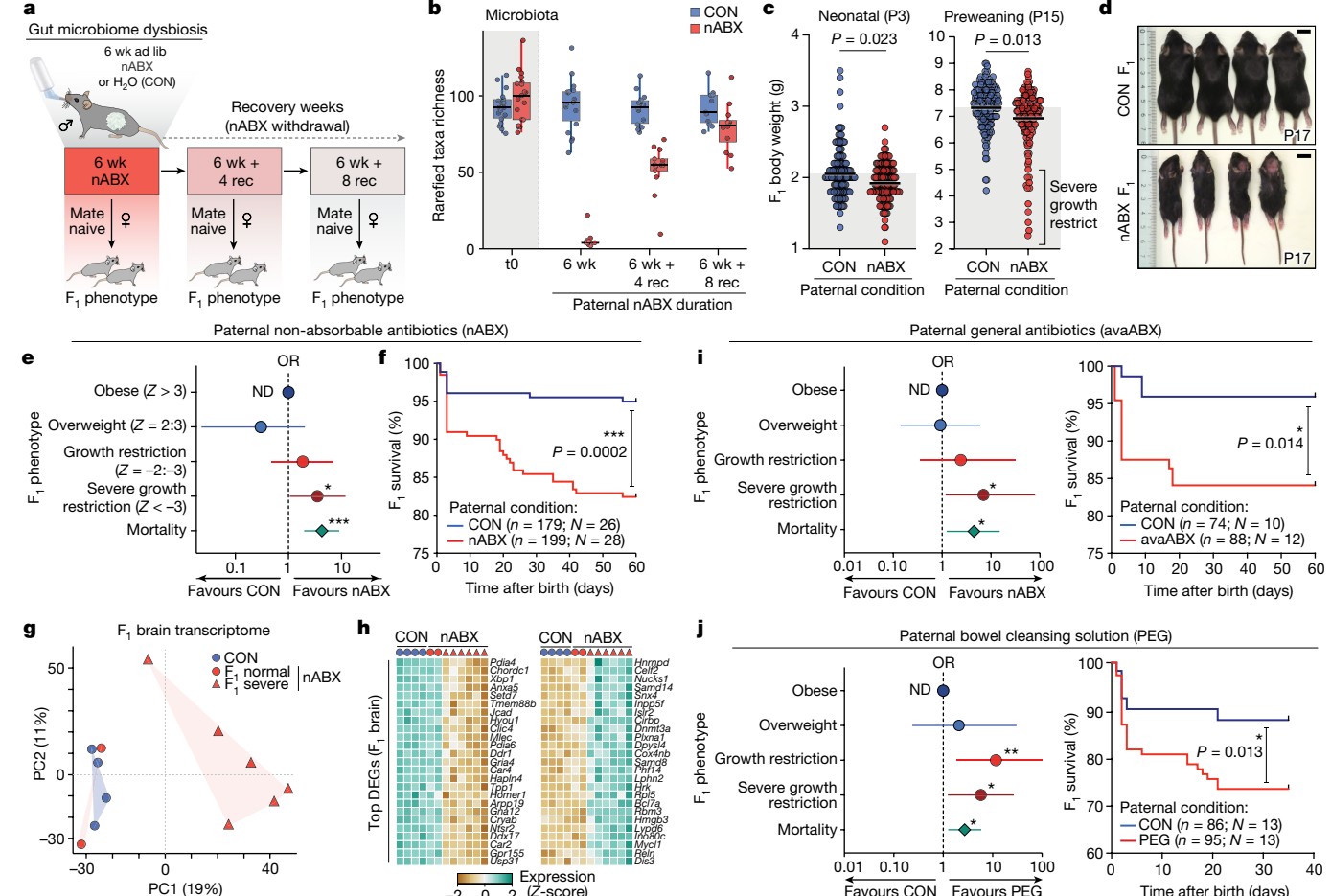

**Fig. 1 | Paternal gut dysbiosis probabilistically triggers major F₁ phenotypes.**
**a**, Schematic showing the strategy for induced paternal dysbiosis and recovery using nABX. **b**, Quantification of microbial taxa richness in males after 6 weeks of nABX treatment and during the recovery (rec) time course, by 16S rRNA sequencing (CON t0 = 18, 6 wk = 18, 6 wk + 4 rec = 14, 6 wk + 8 rec = 11; nABX t0 = 19, 6 wk = 12, 6 wk + 4 rec = 13, 6 wk + 8 rec = 12 individuals per timepoint). Bar represents median, whiskers 1.5× interquartile range. **c**, Body weight of F₁ offspring at postnatal days P3 and P15 according to paternal nABX treatment. *P* value by two-tailed nested (hierarchical) *t*-test (CON *n* = 172, nested into *N* = 26 fathers; nABX *n* = 181 nested into *N* = 28 fathers). Bar indicates mean. **d**, Representative images of SGR phenotype in F₁ offspring from dysbiotic fathers (nABX-treated). **e**, Forest plot showing the log OR of risk for abnormal body-weight classes in offspring that survive to P15. Null effect is represented by a vertical line for which OR value is 1. Whiskers indicate 95% CI, *P* value by two-sided Chi-square test (mortality *P* = 0.0001; SGR *P* = 0.044). **f**, Kaplan–Meier plot showing postnatal survival of F₁ progeny depending on paternal nABX

treatment regime (CON *n* = 179; nABX *n* = 199). *P* value by Mantel–Cox (log-rank) test. **g**, PCA of transcriptomes from F₁ brains derived from control or nABX sires. The nABX offspring are stratified by normal or SGR phenotype. SGR not observed in control offspring. **h**, Heatmap showing expression of top upregulated and downregulated genes in F₁ SGR brains, from independent litters sired by nABX-treated fathers. **i**, Left, OR of F₁ susceptibility for abnormal body weight and mortality when sired by males with dysbiosis induced by 6 weeks of treatment with general antibiotics (avaABX). Whiskers indicate 95% CI, *P* value by two-sided Chi-square test (mortality *P* = 0.014; SGR *P* = 0.038). Right, Kaplan–Meier plot showing postnatal survival of F₁ progeny from avaABX-treated males. **j**, Left, OR of F₁ risk for abnormal body weight when sired by males with dysbiosis induced by 6 weeks of bowel cleansing with PEG laxative. Whiskers indicate 95% CI, *P* value by two-sided Chi-square test (mortality *P* = 0.013; SGR *P* = 0.014). Right, Kaplan–Meier plot showing survival of F₁ progeny from PEG-treated males. *n*, offspring; *N*, litters; ND, not detected.

nABX fathers had significantly lower neonatal birth weight (P3) relative to offspring from control fathers (*P* = 0.023, nested unpaired *t*-test; control (CON) *n* = 172 (26 litters), nABX *n* = 181 (28 litters)) (Fig. 1c). Both female (*P* = 0.017) and male (*P* = 0.029) offspring were affected, while litter size was constant (Extended Data Fig. 2a,b). Moreover, mean body weight of F₁ offspring sired by nABX males remained significantly lower throughout postnatal development (*P* = 0.013 at P15 (preweaning); *P* = 0.015 at P21 (weaning), nested unpaired *t*-test) (Fig. 1c and Extended Data Fig. 2c,d).

Among offspring fathered by dysbiotic males, we also observed a major but partially penetrant postnatal phenotype that manifested as severe growth restriction (SGR; body-weight *Z*-score < −3), which was not observed among control offspring (Fig. 1d and Extended Data Fig. 2e,f). Quantification of this showed a significantly increased

odds-ratio (OR = 3.52; *P* = 0.044, Chi-square) of SGR amongst nABX-derived progeny (Fig. 1e). This was further reflected by a negative skew and excess kurtosis of nABX offspring body weights (skew −1.98; Rku 5.0). Most strikingly, however, F₁ offspring sired by nABX-treated males were associated with a highly significant increased rate of postnatal mortality (*P* = 0.0002, Mantel–Cox test) relative to offspring from control sires (Fig. 1f). This occurred preferentially amongst SGR offspring, suggesting that elevated mortality is linked with increased F₁ susceptibility to growth restriction.

Transcriptome profiling of SGR offspring fathered by independent nABX males showed that they clustered together by principal component analysis (PCA) and separately from offspring of control sires, indicating a reproducible F₁ molecular response (Fig. 1g). Indeed, 2,973 and 1,563 differentially expressed genes (DEGs) are

detected in $F_1$ brain and brown adipose tissue (BAT), respectively. DEGs are preferentially enriched for reactome pathways relating to metabolic processes (top terms: metabolism $P = 2.51^{-9}$; metabolism of lipids $P = 0.000018$) and are robust between SGR offspring from independent litters (Fig. 1h and Extended Data Fig. 2g,j). These data support an intergenerational impact of nABX-mediated paternal dysbiosis on offspring growth, metabolic networks and survival. Importantly, these phenotypes arise as probabilistic rather than deterministic responses to paternal status, and thus manifest as altered risk.

We next asked whether orthologous strategies of paternal microbiota perturbation also elicit an $F_1$ response. First, we used an alternative combination of antibiotics (avaABX) and observed that paternal ab lib administration of avaABX replicated the increased susceptibility of $F_1$ offspring to reduced body weight (SGR OR = 7.0; $P = 0.038$, Chi-square) and increased mortality ($P = 0.014$, Mantel–Cox test) (Fig. 1i). Second, we perturbed the paternal gut microbiota without any exposure to antimicrobial drugs, by performing gastrointestinal cleansing with osmotic laxatives (polyethylene glycol (PEG)), which induces widespread dysbiosis[31,32]. We observed progeny fathered by PEG-treated males had significantly lower $F_1$ body weight ($P = 0.021$, nested unpaired $t$-test; CON $n = 76$ (13 litters), nABX $n = 76$ (13 litters)), with increased SGR susceptibility (OR = 5.8; $P = 0.0142$ Chi-square) and premature mortality ($P = 0.013$, Mantel–Cox test) (Fig. 1j). Thus, multiple distinct perturbations of the gut microbiota in prospective fathers increase the risk of offspring presenting with developmental impairment and premature mortality, supporting a direct link between induced paternal dysbiosis and offspring fitness.

## Reversibility of paternal effects

Next, we investigated whether paternal recovery from gut dysbiosis can rescue the $F_1$ phenotypic effects. Following 4 weeks of nABX withdrawal (6 wk + 4 rec (recovery)), when a significant perturbation of the paternal microbiome persists, $F_1$ offspring again showed significantly lower neonatal weight ($P = 0.012$, nested unpaired $t$-test; CON $n = 160$ (24 litters), nABX $n = 146$ (22 litters)), increased susceptibility to SGR (OR = 8.1; $P = 0.020$ Chi-square) and impaired growth trajectory (Fig. 2a and Extended Data Fig. 3a,b). However, upon recovery of the paternal microbiome after 8 weeks of nABX withdrawal (6 wk + 8 rec) (Fig. 1b and Extended Data Fig. 1a,b), we observed a concurrent recovery of the $F_1$ neonatal weight phenotype ($P = 0.55$, nested unpaired $t$-test; CON $n = 87$ (13 litters), nABX $n = 89$ (13 litters)) and normal developmental growth (SGR OR = 0.97; $P = 0.98$, Chi-square) (Fig. 2b and Extended Data Fig. 3a,b). Moreover, progeny of dysbiotic 6 wk + 4 rec nABX fathers reproduced the significantly elevated $F_1$ mortality rate ($P = 0.0009$, Mantel–Cox test) (Fig. 2c) but progeny from the same fathers following microbiota recovery at 6 wk + 8 rec had no excess mortality ($P = 0.73$) (Fig. 2d,e). These data imply that the increased probability of $F_1$ effects persists during the period of paternal dysbiosis (6 wk, 6 wk + 4 rec) but is reversible and reverts coincident with recovery of the father's gut microbiome (6 wk + 8 rec) and a spermatogenic cycle (about 5 weeks) (Fig. 2e).

Of note, transcriptomics showed SGR offspring sired by 6 week nABX fathers clustered with independent offspring from 6 wk + 4 rec fathers and exhibited highly similar gene ontology enrichments (Fig. 2f and Extended Data Fig. 3c,d). This suggests that affected offspring conceived during the window of paternal dysbiosis funnel into a consistent molecular phenotype, implicating a common underlying aetiology. Moreover, de novo genome sequencing of SGR offspring showed no elevated mutational load, nor explanatory differences in structural variants, single-nucleotide polymorphisms (SNPs) or small insertions and deletions (INDELs) relative to controls (Extended Data Fig. 4a,b). We were further unable to detect transmitted $F_2$ effects (Extended Data Fig. 4c–f). These data establish that $F_1$ phenotypes arising from dysbiotic fathers are not due to inheritance of genetic differences and do not propagate beyond the first generation.

## Mode of intergenerational transmission

To examine the modality of intergenerational inheritance, we first asked whether there is paternal transmission of a dysbiotic gut microbiome[33]. We found that postpartum mothers exhibited no significant compositional changes in their microbiome ($P = 0.77$, alpha diversity Wilcoxon; paternal CON $n = 8$, nABX $n = 11$) and did not cluster by paternal exposure, judged by faecal 16S profiling (Extended Data Fig. 5a). We further detected no significant effects on the maternal microbiome in the days after mating with nABX males, nor differences in offspring or seminal microbiota, suggesting against transmission of altered paternal microbiomes to mother and/or offspring (Extended Data Fig. 5b–h). Indeed, offspring phenotype correlated with the microbiome of its father rather than its own microbiota (Fig. 2g), with some specific taxa showing preferential associations (Extended Data Fig. 5e). We also examined potential coprophagic effects of residual nABX during parental cohousing, noting a small but non-significant trend which resolved in days (Extended Data Fig. 5f–h). To empirically test whether this or other indirect (non-germline) paternal factors impact offspring phenotype, we performed a series of cohousing experiments whereby females were maintained with a control or nABX male in its environment but then mated with independent treatment-naive males. We found no difference in birth weight ($P = 0.85$; nested $t$-test), growth ($P = 0.98$) or survival ($P = 0.35$, Mantel–Cox) of offspring, irrespective of previous maternal exposure to dysbiotic nABX males (Extended Data Fig. 6a–d). We conclude that neither paternal transfer of altered microbiota nor indirect maternal responses underlie $F_1$ effects.

We therefore asked whether $F_1$ phenotypes are transmitted specifically through paternal gametes, by performing in vitro fertilization (IVF). Here, isogenic oocytes are fertilized with sperm from either nABX-treated or control males and then implanted into CD1 high-quality surrogate dams, precluding any parental contact (Extended Data Fig. 6e). We found IVF progeny derived from dysbiotic sperm donors had significantly reduced neonatal birth weight ($P = 0.034$, nested unpaired $t$-test; CON $n = 65$, nABX $n = 80$), impaired postnatal growth ($P = 0.047$) and elevated SGR incidence relative to controls (Fig. 2h and Extended Data Fig. 6f). Independent IVF using BL6 recipient dams, which are relatively poor surrogates, reproduced $F_1$ birth weight effects from nABX sperm donors with greater effect size (Fig. 2i and Extended Data Fig. 6g) ($P = 0.050$, nested unpaired $t$-test; CON $n = 33$, nABX $n = 41$). These data suggest that paternally induced $F_1$ phenotypes arise in independent in utero genetic backgrounds and are transmitted primarily through the gametes and copurifying molecules.

## The gut–germline axis

Transmission through the germline prompted us to investigate the physiological changes in the father's reproductive system induced by acute gut microbiome dysbiosis. We first observed that dysbiotic males from 6 weeks of nABX exposure had significantly smaller testes by mass than did controls ($P = 0.001$, unpaired $t$-test; CON $n = 31$, nABX $n = 32$), which correlated with lower sperm count (Fig. 3a and Extended Data Fig. 7a,b). Histological analysis showed architectural changes in a subset of seminiferous tubules, including vacuoles formed by partial loss of germ cells, which was not observed in control testes (Fig. 3b and Extended Data Fig. 7a). Indeed, nABX males exhibited a significantly increased number of abnormal testis tubules ($P = 0.032$, nested Mann–Whitney; CON mean 0.64%, nABX mean 3.84%) and reduced epithelial thickness ($P = 0.016$; nested unpaired $t$-test) (Fig. 3c,d and Extended Data Fig. 7b–d). These data indicate that testicular physiology is impacted by gut microbiota perturbation.

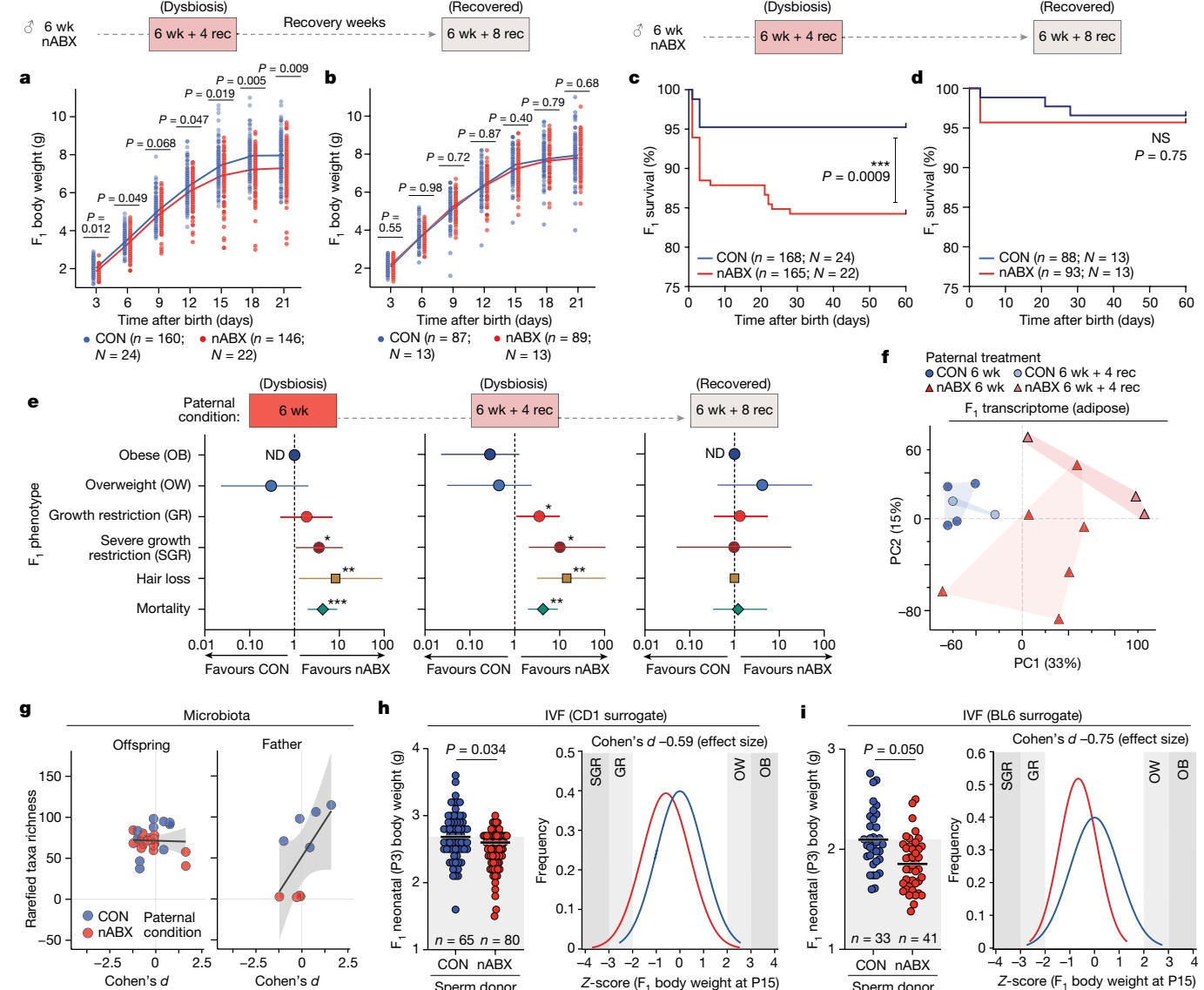

**Fig. 2 | Recovery of paternal gut microbiota rescues susceptibility to $F_1$ phenotypes. a,b,** Growth curves of $F_1$ offspring sired by males during a time course of nABX withdrawal which still retain gut microbiota dysbiosis (6 wk + 4 rec) (**a**) and offspring of the same males after microbiota recovery (6 wk + 8 rec) (**b**), as well as aged-matched control sires. Line indicates mean, *P* value by nested (hierarchical) two-tailed *t*-test. **c,d,** Kaplan–Meier plot showing postnatal survival of $F_1$ progeny sired by dysbiotic (**c**) or recovered nABX males (**d**). *P* value by Mantel–Cox (log-rank) test. **e,** Forest plots showing OR of $F_1$ susceptibility to abnormal body weight and premature mortality when sired by dysbiotic (6 wk, 6 wk + 4 rec) or recovered males (6wk + 8 rec). Null effect is represented by a vertical line for which the OR value is 1. Whiskers indicate 95% CI, *P* value by two-sided Chi-square test (mortality: 6 wk *P* = 0.0001, 6 wk + 4 rec *P* = 0.0014, 6 wk + 8 rec *P* = 0.76; SGR: 6 wk *P* = 0.044, 6 wk + 4 rec

*P* = 0.020, 6 wk + 8 rec *P* = 0.99). **f,** PCA of transcriptomes from $F_1$ SGR adipose tissue sired by 6 wk or 6 wk + 4 rec nABX or control males, each from several independent litters. **g,** Gut microbiota richness of offspring (left) is not affected by paternal gut microbial status and does not correlate with $F_1$ phenotype. Paternal microbiota richness (right) does correlate with $F_1$ offspring phenotype. Shaded areas indicate 95% CI. **h,i,** Left, neonatal $F_1$ body weight following IVF of isogenic oocytes using control or nABX-treated sperm donors. Right, $F_1$ body-weight distribution at P15 following IVF. Data are independently derived from CD1 strain surrogate mothers (CON *n* = 66 offspring, nested into *N* = 12 fathers; nABX *n* = 75, nested into *N* = 12) (**h**) or BL6 strain surrogate mothers (CON *n* = 33 offspring, nested into *N* = 8 fathers; nABX *n* = 41, *N* = 7 fathers) (**i**). *P* value by nested (hierarchical) two-tailed *t*-test.

To characterize the reproductive system response to dysbiosis at the molecular level, we first performed untargeted metabolomic profiling of father's testes, annotating 3,803 features[34]. Global PCA of annotated metabolites showed testes cluster according to gut microbiota status, with clear separation between nABX and control males at both 6 wk and 6 wk + 4 rec, which corresponds to the period of transmitted $F_1$ effects (Fig. 3e). By contrast, testicular metabolomic profiles at 6 wk + 8 rec are indistinguishable, indicating that metabolites dynamically recover coincident with restoration of the gut microbiota and with reversal of transmitted $F_1$ effects. We identified 68 significant differentially

abundant metabolites in testes of dysbiotic males, which included the fatty acid, anandamide, which acts in the endocannabinoid pathway and the signalling lipid, sphingosine-1-phosphate (S1P) (Fig. 3f and Extended Data Fig. 8a). Differentially abundant metabolites are specifically enriched for sphingolipids and glycerophospholipids, as well as endocannabinoids, which have all been implicated in germ cell function[35] (Extended Data Fig. 8b). We further investigated transcriptomic profiles in testes of dysbiotic males, observing limited expression changes at both bulk and single-cell levels (Fig. 3g and Extended Data Figs. 8c–f and 9a–g). Gene set enrichment indicated

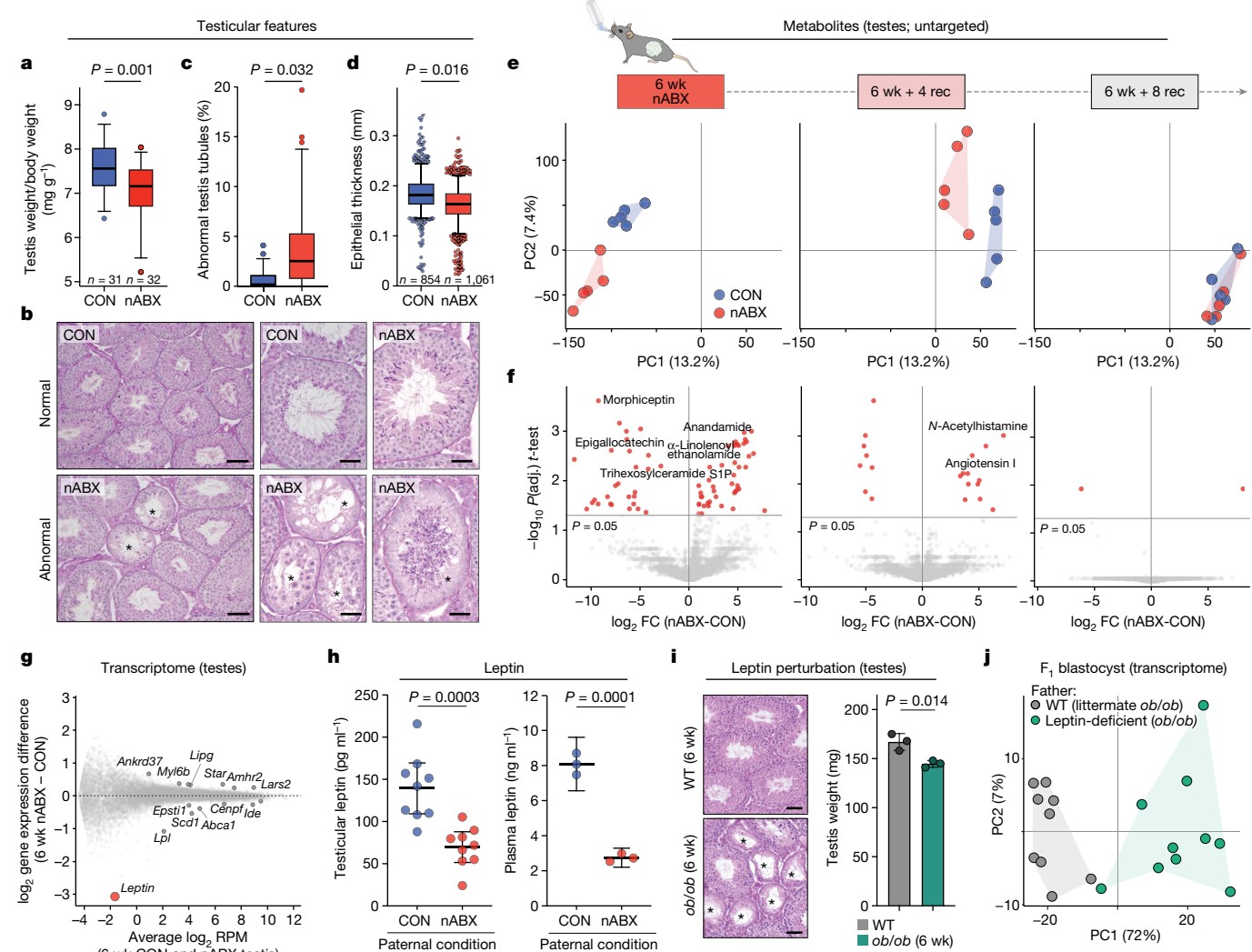

**Fig. 3 | Testicular responses to gut microbiota dysbiosis indicate a regulatory gut–germline axis. a**, Boxplot showing testis mass to body weight ratio after 6 weeks of nABX treatment in males. Bar represents median and whiskers indicate 5th–95th percentile. *P* value by unpaired two-tailed *t*-test (CON *n* = 31; nABX *n* = 32). **b**, Representative haematoxylin and eosin stained histological sections of testes from control and nABX males. Seminiferous tubules from dysbiotic males show incidence of vacuoles formed by loss of epithelium and absence of mitotic (spermatogonial) compartments. Asterisks indicate abnormal tubules. **c**, Quantification of abnormal testis tubules in control and dysbiotic males. *P* value by nested Mann–Whitney test (CON *n* = 54 sections, nested into *N* = 4 males; nABX *n* = 72, N = 5; *P* = 0.032). Bar represents median and whiskers indicate 5th–95th percentile. **d**, Quantification of epithelial thickness of seminiferous tubules. *P* value by nested *t*-test (CON *n* = 854 tubules, *N* = 4 males; nABX *n* = 1,061, *N* = 4; *P* = 0.016). Bar represents median and whiskers indicate 5th–95th percentile. **e**, PCA of untargeted metabolomics profiles from independent testis of control or dysbiotic males (6 wk, 6 wk + 4 rec) and after gut microbiome recovery (6 wk + 8 rec). **f**, Volcano plot highlighting differentially

abundant metabolites in testes after 6 weeks of nABX and during recovery to restore the gut microbiome. *P* value by two-tailed *t*-test adjusted for multiple testing. **g**, MA ((M (log₂ ratio) and A (mean average)) plot showing gene expression changes in testes from independent (*n* = 5) nABX males. **h**, Quantitation of leptin hormone level in testes (left) and circulating plasma (right) of dysbiotic males after 6 weeks of nABX, by ELISA. Bar indicates mean with 95% CI whiskers. *P* value by unpaired two-tailed *t*-test (CON *n* = 9; nABX *n* = 9). **i**, Testes features in males that are leptin deficient (*ob/ob*) for 6 weeks and wild-type controls (WT *n* = 3; *ob/ob n* = 3). Left, haematoxylin and eosin stained tubule sections; stars indicate abnormal histo-architecture. Right, testis weight, *P* value by unpaired two-tailed *t*-test and bar indicates mean with 95% CI. **j**, Transcriptome PCA of blastocysts derived from leptin-deficient or control fathers. Data points represent single-embryos from duplicate independent IVF experiments, each with triplicate independent fathers and littermate fathers as controls. Scale bars, 100 μm (left panel) or 50 μm (right panel) (**b**); 50 μm (**i**). FC, fold change.

that glycerophospholipids- and steroidogenesis-related genes were preferentially dysregulated, consistent with altered metabolomic profiles (Extended Data Fig. 8e–g). The most sensitive gene was *Leptin*, however (Fig. 3g and Extended Data Fig. 9h), which encodes a hormone produced primarily by adipocytes but also by germ cells, with key roles in energy homoeostasis and reproduction[36,37].

The cumulative evidence indicates a significant change in the testicular environment in response to gut microbiome perturbation, including altered metabolite profiles, physiology and hormones. This

suggests the existence of a gut–germline axis in mammals which carries an important homoeostatic function, analogous to recent reports in *Drosophila*[38,39].

Among the paternal responses to nABX was strong dysregulation of *Leptin* (Fig. 3g). Validation with ELISA showed that nABX-mediated dysbiosis leads to significantly reduced levels of leptin hormone, both systemically in circulating blood (*P* = 0.0001, unpaired *t*-test; CON *n* = 3; nABX *n* = 3) and specifically in testes (*P* = 0.0003; CON *n* = 9; nABX *n* = 9) (Fig. 3h). To investigate the implications of this, we used 6-week-old

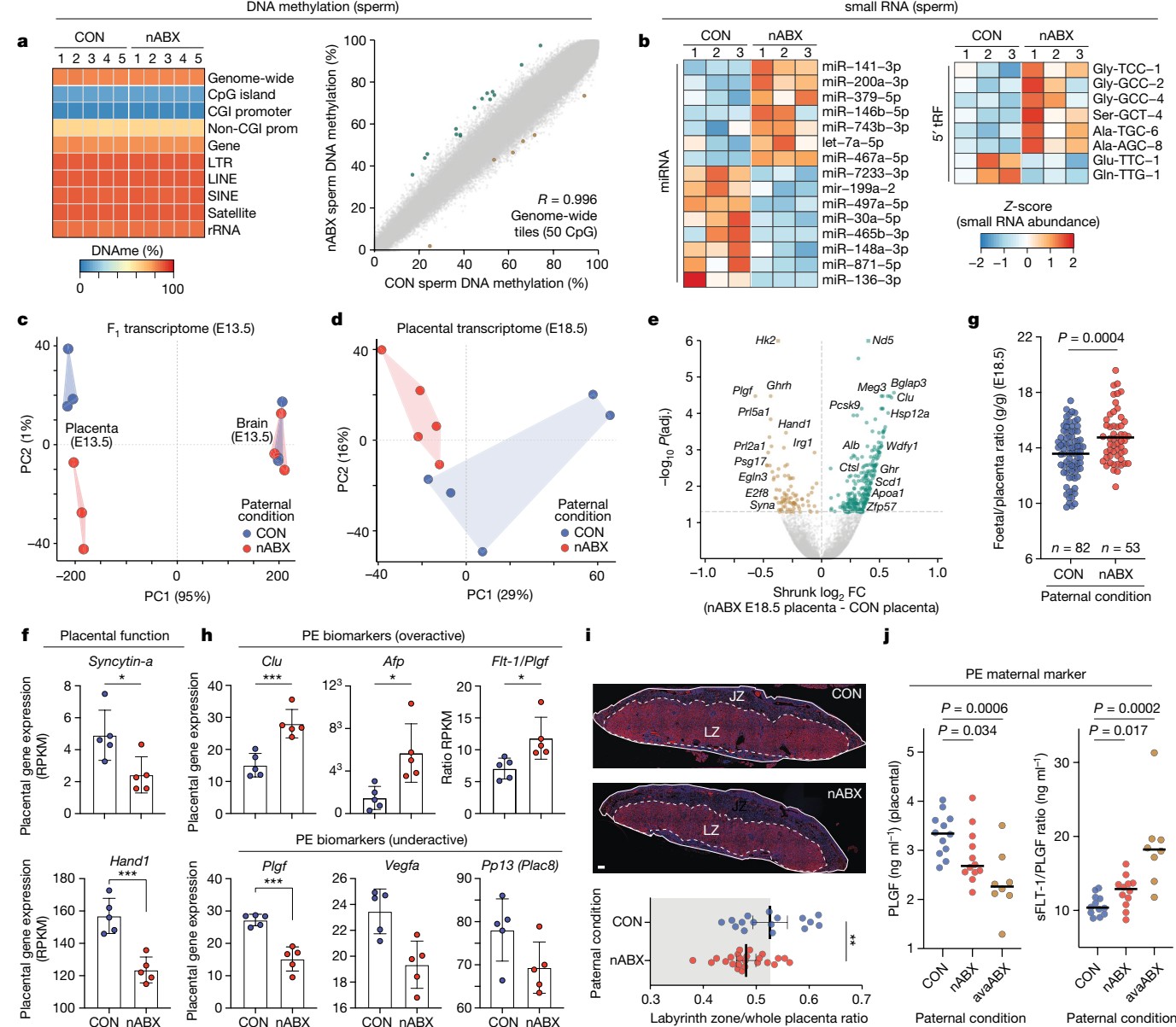

**Fig. 4 | Paternal dysbiosis induces F₁ placental insufficiency. a**, Left, heatmap showing DNA methylation levels across genomic features in sperm from control or nABX-treated males (*n* = 5 methylomes per condition) by whole-genome bisulfite-seq. Right, scatter plot of genome-wide DNA methylation (50 CpG tiles), with differential tiles highlighted. **b**, Heatmaps showing differential abundance of selected miRNA (left) and tRNA fragments (right) in pooled purified sperm from independent (*n* = 9) control or nABX males. **c**, PCA of transcriptomes from embryonic (E13.5) brain and placenta according to paternal exposure. **d**, PCA of placenta transcriptomes at E18.5 from independent litters. **e**, Volcano plot showing DEGs in E18.5 placenta sired by 6-week nABX-treated males. *P* value adjusted for multiple testing. **f**, Expression of key genes for placental development in E18.5 placenta. Bar indicates mean and whiskers are s.d. Each data point is an independent placenta (CON *n* = 5 placenta (3 litters); nABX *n* = 5 placenta (3 litters)). **g**, Ratio of fetal mass to placental mass at E18.5, depending on preconception paternal condition. Bar indicates mean, *P* value by unpaired two-tailed *t*-test (CON *n* = 82, nABX *n* = 53). **h**, Expression of biomarkers of pre-eclampsia (PE) in E18.5 placenta derived from nABX fathers. Bar indicates mean and whiskers are s.d. *P* value by multiple-testing corrected DESeq2. CON *n* = 5 placenta (3 litters); nABX *n* = 5 placenta (3 litters). **i**, Representative placenta sired by control or dysbiotic males (nABX), stained for DAPI (blue) and VE-cadherin (red) to demarcate the labyrinth zone (LZ). Quantification below shows LZ as a ratio of total area. Bars indicate mean ± 95% CI. Each data point is an independent placenta tissue section (CON *n* = 4 placenta (4 litters); nABX *n* = 6 placenta (6 litters)). JZ, junctional zone. *P* value by unpaired two-tailed *t*-test. **j**, Levels of placental growth factor (PLGF) protein (left), a marker of PE and the sFLT1/PLGF ratio (right), in F₁ placenta depending on paternal regime. *P* value by two-tailed *t*-test. Bars indicate median. Each data point is an independent placenta (CON *n* = 12 placenta (7 litters); nABX *n* = 12 placenta (6 litters); avaABX *n* = 8 placenta (3 litters)). Asterisks indicate *P* values of 0.05 or less* and 0.0001 or less***. Scale bar, 100 μm. Prom, promoter.

*ob/ob* mice (*Leptin*-null), which models the timespan of leptin deficiency induced by 6 weeks of nABX exposure. Such young *ob/ob* males had reduced testes weight (*P* = 0.014) and tubule abnormalities interspersed with normal histo-architecture, similarly to dysbiotic 6-week nABX males (Fig. 3i and Extended Data Fig. 10a–c). We therefore assessed the intergenerational impact of short-term leptin deficiency. Although natural matings failed, IVF using 6-week *ob/ob* sperm fertilized wild-type oocytes as efficiently as control sperm, implying underlying germ cell viability (Extended Data Fig. 10d). Single-embryo transcriptomics of the resulting blastocysts showed that offspring sired by leptin-deficient

fathers strongly cluster away from embryos sired by control fathers (Fig. 3j and Extended Data Fig. 10e,f). Indeed, we detected more than 500 $F_1$ high-confidence DEGs, with chromatin pathways preferentially deregulated (Extended Data Fig. 10g–i). Importantly, *Leptin* itself is not expressed during early development ruling out a haploinsufficient effect of $F_1$ heterozygosity (Extended Data Fig. 10g). Taken together, these data show that leptin is systemically dysregulated by induced microbiome dysbiosis and that direct perturbation of paternal *Leptin* before conception has an intergenerational legacy on offspring gene expression programmes, implicating leptin as an important signalling component in the gut–germline axis.

We next sought to understand the impact of the gut–germline axis on mature gametes by searching for molecular changes in sperm. We initially charted DNA methylation at base resolution. Independent methylomes ($n = 5$) of purified sperm from nABX males were highly comparable to controls, with no change in DNAme globally or at genomic features (Fig. 4a and Extended Data Fig. 11a,b). We identified only 21 differentially methylated regions (DMRs) genome-wide (logistic regression $P$(adjusted (adj)) < 0.05 and >20% absolute change, 50 CpG tiles) (Fig. 4a and Extended Data Fig. 11c,d), which typically overlapped epivariable CpG shore regions, implying that they may reflect natural variation. Genomic imprints were all correctly established in both control and nABX sperm (Extended Data Fig. 11e). We next assayed sperm-borne small RNA with high-quality libraries from independent sperm collections ($n = 9$ males, pooled into $N = 3$). We detected significant abundance changes in several microRNA, including miR-141 ($P$(adj) = $1.15 \times 10^{-9}$) and miR-200a ($P$(adj) = 0.008), which act together to regulate epithelial–mesenchymal transition and placental development (Fig. 4b and Extended Data Fig. 11f,g)[40,41]. We also observed changes in the abundance of 5′ transfer RNA fragments (tRF) and in particular upregulation of tRF-Gly-GCC in dysbiotic males (Fig. 4b and Extended Data Fig. 11f–h), which has been implicated in intergenerational effects[42,43]. Quantitative differences in small RNA abundance were confirmed in independent samples by TaqMan quantitative PCR (Extended Data Fig. 11i). Overall, although DNA methylation is relatively stable, the composition of small RNAs in sperm is modified in response to nABX-mediated dysbiosis. Considered with the altered metabolite and hormonal profiles, this suggests that compound changes in macromolecule composition are transmitted to offspring.

## Placental responses to dysbiotic fathers

To understand the mechanism through which sperm influence offspring phenotype, we sought to identify the initial source of embryonic defects. At E13.5 (about mid-gestation) we found no DEGs in embryos sired by nABX-treated males relative to control and embryonic transcriptomes could not be distinguished by PCA (Fig. 4c). By contrast, transcriptomes of placenta at E13.5 from independent matings clustered strongly depending on the paternal nABX regime (Fig. 4c), with 538 DEGs, which included downregulation of several *Prolactin* genes (Extended Data Fig. 12a,b). To examine this further, we assayed mature placenta at E18.5 and observed their transcriptomes clustered according to paternal microbiome status, with 348 high-confidence DEGs (Fig. 4d and Extended Data Fig. 12c). Upregulated DEGs are enriched for steroid metabolism ($P$(adj) = $2.31 \times 10^{-7}$) whereas downregulated DEGs are associated with glycolysis ($P$(adj) = 0.00039) (Fig. 4e). Notably, they also include downregulation of several factors important for placenta development, such as *Hand1* and *Syna*, suggestive of impaired placental ontogeny (Fig. 4f). To investigate this possibility further, we scored the ratio of placental mass relative to embryo at E18.5. We found a significant change in the $F_1$ fetal-to-placental ratio when sired by nABX males ($P$ = 0.0004, unpaired *t*-test; CON $n = 82$ nABX $n = 53$), which is driven specifically by reduced placenta mass (Fig. 4g). This is consistent with a placental defect derived from dysbiotic fathers.

To decipher the molecular aetiology underlying this, we noted that the top ten DEGs in nABX-derived placenta included several clinical markers for human placental insufficiency disorders, such as pre-eclampsia (Fig. 4d and Extended Data Fig. 12c). For example, *Plgf* was significantly downregulated in mature placentae sired by dysbiotic fathers, whereas upregulated pre-eclampsia markers include the *Flt:Plgf* ratio, *Clu* and *Afp* (Fig. 4h). To further investigate the potential for dysbiotic fathers to induce placental insufficiency, we examined placenta structure. Here, placenta derived from nABX fathers were associated with a significantly reduced labyrinth zone ($P$ = 0.0098) (Fig. 4i), which is a frequent underlying cause of placental disorders[44]. Indeed, we further found significantly impaired vascularization ($P$ = 0.0076) and increased placental infarction ($P$ = 0.0296) (Extended Data Fig. 12d–g). Finally, we assayed the level of placental growth factor (PLGF) hormone, which when low, is a prime diagnostic marker for pre-eclampsia in humans[45]. We found that PLGF was significantly lower in placenta when offspring were sired by dysbiotic males induced by either nABX ($P$ = 0.034) or avaABX ($P$ = 0.0006), while the sFLT/PLGF ratio was significantly elevated (Fig. 4j).

## Discussion

Taken together, cumulative evidence suggests that environmentally induced perturbations to the gut microbiota elicit a significant reproductive response in prospective fathers. This supports a regulatory gut–germline axis, which when perturbed can propagate to affect offspring disease risk, at least in part by impacting forthcoming placental function. The gut microbiota may therefore act as a principal interface wherein distinct environmental inputs, such as antibiotic regimes or diet, can converge and signal to male germ cells directly or indirectly, ultimately exerting influence on progeny. Such paternal $F_1$ effects manifest probabilistically however, with multifactorial molecular mechanisms acting in/on sperm probably interacting to underpin transmission. Although we identify altered lipid metabolites, hormones such as leptin and small RNA payloads here, future work is necessary to deconvolve the phenotypically relevant modalities of inheritance and their applicability beyond murine models. However, our observation that paternal conditioning can impact placental ontogeny provides a mechanistic grounding for mammalian intergenerational effects[46]. Moreover, because restoration of paternal gut microbiota before conception rescues emergent $F_1$ phenotypes, the effect is reversible and thus remedially tractable. Given the prevalence of lifestyle and antibiotic practices that (re)shape human microbial communities[23–25,47], this could prove an area of interest towards mitigating against adverse pregnancy outcomes. More generally, our data highlight the importance of understanding how environmental factors can modify complex biological systems across scales; from direct molecular responses to intergenerational disease susceptibility.

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

## Methods

### Animal husbandry

All experiments involving mice were carried out in accordance with the approved protocol and guidelines by the laboratory animal management and ethics committee of the European Molecular Biology Laboratory (EMBL) under license no. 20190708_JH and the Italian Ministry of Health under authorization code no. 308/2021-PR. The inbred C57BL/6J strain was used as the primary mouse model, whereas CD1 IGS or C57BL/6J dams were used as surrogate mothers for IVF. C57BL/6J mice deficient in leptin (*ob/ob* mice) were used to study the intergenerational effects of paternal testicular leptin deficiency. Mice were housed under a 12 h light/dark cycle (from 07:00 to 19:00), with ad libitum access to regular diet and water. Standard chow contained 18.5% protein, 5.3% fat, 4% fibre and other nutritional additives in pellet form and is suitable for long-term maintenance, breeding, lactation and gestation periods (NFM18, Mucedola).

### Experimental strategy for induced dysbiosis in animals

At 5 weeks of age, specific pathogen free (SPF) male C57BL/6J inbred mice were transferred to a conventional colony facility. The male mice were divided into separate cages (one mouse per cage) and randomly assigned to two groups: (1) control group, which received regular diet and sterile filtered (0.2 µm) water for six successive weeks (abbreviated as CON mice); (2) treated group, which received regular diet and sterile filtered (0.2 µm) water supplemented with a cocktail of FDA-approved nABX (neomycin 2.5 mg ml$^{-1}$, bacitracin 2.5 mg ml$^{-1}$ and pimaricin 1.25 µg ml$^{-1}$; abbreviated as nABX) for 6 successive weeks. The water bottle and nABX cocktail were replaced freshly every 5 days. At the end of 6 weeks of treatment, male mice from both groups were mated with treatment-naive 6-week-old C57BL/6J inbred female mice. Subsequent to mating, male mice were maintained for an extra 8 weeks without nABX (recovery period) and they had ad libitum access to regular diet and sterile filtered (0.2 µm) water. During the 8 week recovery period, each male mated at two time points: at 4 weeks after nABX withdrawal and at 8 weeks postwithdrawal of nABX (denoted as 6wk + 4rec and 6wk + 8rec, respectively). All the female mice received regular diet and water throughout gestation and lactation period. Of note, the dosage of each antibiotic used is lower than the standard human-equivalent dosage and half of the standard dose administered to mice in previous studies and therefore constitutes a relatively low dosage regime designed to perturb rather than ablate gut microbiota communities[30,48,49].

An equivalent experimental setup and 6 week time course was used for alternative strategies to induce acute gut dysbiosis. First, ad libitum administration of a cocktail of absorbable antibiotics (ampicillin 0.5 mg ml$^{-1}$, vancomycin 0.25 mg ml$^{-1}$ and amphotericin B 12.5 µg ml$^{-1}$; abbreviated as avaABX) in filtered (0.2 µm) water. Second, through bowel cleansing using 5% PEG (PEG 4000, EMD Millipore). Herein, the sire males were given avaABX or 5% PEG in drinking water for 6 weeks to provoke acute gut microbiota dysbiosis, whereas the randomly assigned control group received only sterile filtered (0.2 µm) water. At the end of 6 weeks, male mice from each group mated with treatment-naive 6-week-old C57BL/6J inbred female mice. For each antibiotic, the administered dose was adjusted to half the standard antibiotic dose that was used in previous studies[50–52]. Likewise, the PEG concentration was lower than the standard dose used for human bowel cleansing or for medicating constipation and in previous mouse studies[31].

### Male fertility parameters

To evaluate the effect of nABX-induced dysbiosis on breeding males, the testis-to-body weight ratio and standard male fertility parameters were characterized on the basis of a quantitative descriptive analysis which includes, sire body weight, testis weight, sperm count and fecundity potential:

1. Testes-to-body weight ratio: to assess the effect of nABX treatment, the body weights of sires treated for 6 weeks were measured and compared with those of age-matched CON sires. Furthermore, to assess the impact on reproductive organs, testes were carefully collected from CON and nABX-treated mice, weighed and the testes-to-body weight ratio difference between the two groups was calculated.

2. Sperm count: the method described by ref. 53 was used for sperm counts. In brief, sperm suspensions were prepared by mincing cauda in 1 ml of 1× PBS (pH 7.2). The suspension was pipetted and filtered through 80 µm nylon mesh to remove tissue fragments. An aliquot (0.05 ml) from the sperm suspension (1 ml) was diluted with 1:40 PBS (pH 7.2) and mixed thoroughly. The sperm suspension was counted using Bürker chamber after discharging a few drops of the suspension. To determine sperm count per millilitre, sperm counts in each of the four squares (1 mm$^2$) were averaged and multiplied by the dilution factor ×10$^4$ to calculate the total number of sperm per mouse.

3. Fecundity potential: 11-week-old sire males (both CON and nABX-treated) were mated with treatment-naive 6-week-old C57BL/6J females. The females were checked for the presence of a copulatory plug over the first 4 days and moved to a holding cage once they were plug positive, then checked again at 3 weeks for pregnancy and litter size.

### Offspring generation

To generate $F_1$ offspring both natural mating and artificial (IVF) were used. A coding system was used to ensure that researchers and animal husbandry staff were double-blind to the paternal treatment regime.

**Natural mating.** To generate $F_1$ offspring, each CON or nABX male mouse was placed with a single 6-week-old treatment-naive virgin female. During the mating period (1–4 days), mice were housed under standard optimized environmental conditions with ad libitum access to regular diet and water. Each morning females were checked for vaginal status, all females with positive signs of mating (vaginal plugs) were removed from the male cage and housed in a new cage, whereas all females that did not show a vaginal plug were returned to the male cages in the afternoon and examined on each successive morning. To minimize microbial transfers between the pairs housed together, the mating hours of the pairs was restricted from late afternoon to early morning (15:00–09:00) throughout the mating period.

**In vitro fertilization.** As described previously[54], this method of $F_1$ offspring generation follows five critical steps: (1) superovulation of oocyte-donor females; (2) preparation of fresh sperm from sire males; (3) collecting isogenic oocyte from donor females; (4) in vitro sperm–oocyte fertilization; (5) embryo transfer to pseudopregnant CD1 or C57BL/6J surrogate mothers. This process was performed as follows:

1. Superovulation: 4-week-old C57BL/6J oocyte-donor female mice were injected intraperitoneally with 0.2 ml of PMSG plus inhibin antiserum (IAS). Precisely 48 h later, hCG (0.15 ml) was injected intraperitoneally into oocyte donors. Herein, hCG injection was done according to the planned sperm–oocyte fertilization time (approximately 13–14 h after hCG injection).

2. Preparation of fresh sperm: cauda epididymis were dissected from 11-week-old mice, with any excess adipose tissue and blood cleaned away. Subsequently, cauda epididymis was transferred into a centre-well organ culture dish (Falcon) containing 120 µl of cryoprotective agent (CPA). Five to six incisions were made across each cauda epididymis with optimized scissors and the surface of the cauda was gently pressed to release the sperm. Fresh sperm were pre-incubated in CPA for 3 min at 37 °C with gentle stirring of the Petri dish every minute. A 10 µl sperm suspension was transferred into the centre drop of 90 µl of TYH-MBCD medium and the fresh sperm were capacitated by incubation for at least 10–30 min at 37 °C

in the 5% $CO_2$ incubator. Dishes remained undisturbed until the sperm were moving rapidly and equilibrated in the medium.

3. Oocyte collection: a day before oocyte collection, 90 µl of HTF medium containing 1 mM GSH was covered with paraffin oil and placed overnight at 37 °C in the 5% $CO_2$ incubator to allow equilibration. The next day, the oviduct isolated from each superovulated female (ten donors in total) was divided into two, half used for fertilizing with sperm from CON mice and half with sperm from nABX-treated mice, which assured isogenic oocytes source and eliminated oocyte-donor quality as a confounding effect. Once each half of the oviduct ampullae were transferred into fertilization dish containing HTF drops, swollen ampullae were clipped with a 18G needle to release the cumulus masses into the oil and then transferred into the fertilization drop.

4. Sperm–oocyte fertilization: from capacitated sperm in TYH-MBCD drops, a 10 µl of sperm suspension was taken from the peripheral part of the pre-incubation drops which contain the most motile sperm. Aliquoted sperm were then transferred either from CON mice or nABX-treated mice into the centre of the drops in the fertilization dish containing oocytes. After insemination of the oocytes with sperm, sperm–oocyte fertilization dishes were cultured at 37 °C under 5% $CO_2$ for 3–4 h. After 3–4 h of incubation, the presumptive zygotes were washed three to four times with 150 µl drops of M2 medium and cultured overnight in four-well NUNC plates with 500 µl of KSOM medium supplemented with amino acids (KSOMaa) at 37 °C under 5% $CO_2$ and 5% $O_2$ incubator. After 18–24 h, the fertilized two-cell stage embryos were transferred to the oviducts of pseudopregnant CD1 or C57BL/6J female surrogate mice.

5. Embryo transfer: embryo transfer was conducted as described previously[55]. Vasectomized males were mated with 6–8-week-old CD1 females or C57BL/6J (27–30 g) to produce 0.5 days postcoitum (dpc) pseudopregnant recipients. Plug-positive females were selected and anaesthetized by inhalation of isofluorane. To transfer two-cell stage embryos, the following subsequent steps were applied: the dorsal midline opened, ovary body wall incised and held outside the body, after breaking the bursa the tip of the capillary inserted into the infundibulum to transfer ±20 embryos per single pseudopregnant CD1 or C57BL/6J mouse and finally a gentle push applied through mouth pipette to dislodge embryos from the glass capillary tube into the infundibulum of one oviduct.

**$F_1$ phenotype after co-housing to examine indirect effects.** To separate the potential indirect impact on offspring of maintaining females with a dysbiotic male (such as microbial transfer to mothers or coprophagy of nABX faeces), we used a cohousing strategy coupled with control (independent) male mating. A male with either 6 weeks of previous nABX exposure or a control male was placed with one 6-week-old treatment-naive virgin female during the day (09:00–16:00), recapitulating full parental co-exposure. Males were removed from the cages at 16:00 to prevent mating during the night, with females remaining in the male cages to maximize the potential to detect microbial transfer or coprophagic effects. Each day at 16:00, females were checked for vaginal status, all females with positive signs of mating (vaginal plugs) were excluded from the study, whereas all females that were vaginal plug negative were held in the male cages. In the 4 day cohousing period, mice were maintained in standard optimized environmental conditions with ad libitum access to food and water. After 4 days of cohousing, all plug-negative females (exposed to nABX or CON males) were mated with conventionally raised 11-week-old C57BL/6J male mice (independent naïve males) to determine whether any indirect effects of previous nABX male exposure could be detected in $F_1$ phenotypes.

**$F_2$ offspring generation.** $F_2$ offspring were produced by intercrossing two siblings from the $F_1$ generation derived from either dysbiotic (nABX) or CON $F_0$ sires.

## Phenotyping and tissue collection

**Offspring body weight and survival metrics.** A comparative growth phenotype analysis was applied to evaluate $F_1$ prenatal to postnatal development. Matings were setup using a blinded code system and $F_1$ phenotypes were recorded by individuals and/or husbandry staff without knowledge of the paternal condition. For postnatal analysis, the litter size at birth was recorded and still-born pups were monitored. The body weight of pups was measured every 3 days from postnatal day P3 until weaning age (P3–P21) and every week up until P56. $F_1$ offspring morbidity and mortality were also recorded daily in this age range (P0–P60). To account for outliers due to extreme litter size effects, only litters within 2 s.d. of the average litter size (6 ± 2 pups per litter) were included in the study (that is, litters of 4–8 pups), which corresponded to about 81% of total litters. Likewise, the postnatal body-weight trajectories of $F_1$ progeny generated by IVF were characterized and analysed using aforementioned methods. Note, $F_0$ pregnant females as well as $F_1$ offspring were fed a regular diet and sterilized water throughout gestation, lactation and the postweaning period. To phenotype the prenatal $F_1$ conceptus, derived from CON or nABX-treated males, pregnant dams were killed either at E13.5 or at E18.5. Placenta and fetus were carefully isolated, cleaned and weighed. Subsequently, embryonic tissues and placentae were isolated for transcriptome profiling. Placental tissue was further collected for immunofluorescent or ELISA analysis.

**Tissue and sperm isolation for omics.** Testes and epididymal sperm were isolated as described previously[42], to ensure maximum purity and quality. Briefly, testes were carefully dissected from 11-week-old mice, with all fat pads removed, before being individually weighed, snap-frozen in liquid nitrogen and stored at −80 °C until processing for transcriptomics or ELISA assay. The contralateral testis was processed for histological staining. For sperm collections, cauda epididymis was dissected from mice and placed in M2 media at warmed 37 °C. Two small incisions were made at the proximal end of cauda using a 26G needle and holes were poked in the rest of the tissue to let the sperm swim out and epididymal fluid release. Sperm-containing media were incubated for 30 min at 37 °C, then transferred to fresh tubes and incubated for another 15 min at 37 °C. After incubation, sperm were collected by centrifugation at 2,000$g$ for 5 min, followed by a 1× PBS wash and a second wash with somatic lysis buffer (contains 0.1% SDS and 0.5% Triton-100X diluted in distilled $H_2O$) for 20 min in ice, which is important to eliminate any potential somatic cell contamination. Somatic lysis buffer-treated sperm samples were then collected by centrifugation at 3,000$g$ for 5 min and finally washed twice with 1× PBS and stored at −80 °C. Purification of sperm cells was confirmed by microscopic examination.

**Testis single-cell isolation for 10× single-cell RNA sequencing.** Single testicular cells were isolated using a two-step enzymatic digestion protocol described previously[56], with minor modifications. In brief, testes from nABX and CON males were excised, washed twice in PBS, the tunica albuginea removed, each testis transferred to 1 ml of lysis buffer (1 mg ml$^{-1}$ of collagenase A diluted in PBS) and triturated five times using a P1000 tip to disrupt the tissue and incubated for 5 min at 37 °C with gentle shaking at 300 rpm. The tubules were gravity sedimented, centrifuged for 5 min at 300$g$ at room temperature. The supernatant was removed and remaining tissue washed with PBS and digested in TrypLE (1×) and DNase I. The suspension was triturated vigorously ten times, incubated at 37 °C for 5 min, followed by repeat trituration and incubation until the end of 15 min. The digested single-cells suspension was sequentially size-filtered and washed through 70 and 40 µm strainers and pelleted by centrifugation at 300$g$ for 5 min. The pelleted cells were resuspended in 1 ml of cold PBS + 0.04% BSA, pooled into a Falcon tube (four testis per group) and counted using Countess II (average viability 85–90%). Target cell concentration was adjusted in the range

of about 1,100 cells μl$^{-1}$ using cold PBS with 0.04% BSA and placed on ice until loading on the 10x Chromium chip.

**Seminal fluid collection.** Seminal vesicles were collected in a laminar flow-hood as described previously[57] with slight modifications. To prevent cross contamination, one experimental animal (CON or nABX) was placed in the hood per sample collection. To ensure maximum sterilization, the experimenter wore a sterile gown, gloves, mask and scrubbed their hands with 70% ethanol and followed the following procedures. In brief, the mice were placed in a laminar flow-hood that had been previously cleaned with 70% ethanol and 2% Virkon followed by 2 h of ultraviolet radiation. To collect seminal vesicles aseptically, mice were euthanized by cervical dislocation and their abdominal areas were cleaned with 70% ethanol and incised with sterilized tweezers and scissors. Seminal vesicles were then excised using new sterilized tweezers and scissors and placed in a sterilized 2 ml Eppendorf tube, snap-frozen in dry ice and stored at −80 °C until DNA extraction for seminal fluid microbiome profiling. Note, dissection tools used in this sample collection were used only once per mouse.

**Blood collection.** Blood samples were collected from saphenous vein microvette blood collection tubes (Microvette 100K3E SARSTEDT, catalogue no. 20.1278.100) and centrifuged at 5,000 rpm for 10 min at 4 °C; plasma was obtained and stored at −80 °C until assayed.

**Placenta tissue collection.** The placental tissues were collected according to the protocol described in ref. 58. Briefly, samples were collected at E13.5 and E18.5, counting from E0.5 as the day the copulation plug was found. The conceptuses were removed from a pregnant mouse and the uterine horn containing a single conceptus was segmented. Using microscissors, the uterine muscle was gently peeled away from the antimesometrial side (the side that is least vascularized), the yolk sac was cut to expose the amniotic sac and the placenta was carefully dissected and separated from the umbilical cord and yolk sac to collect the entire placental disc without cut or tear remnants. Samples were snap-frozen in liquid nitrogen and stored at −80 °C until they were processed for RNA-seq or fixed in 4% paraformaldehyde for histology and immunohistochemical analysis.

**Pre-implantation embryos.** To generate blastocyst-stage embryos, IVF was used. Briefly, 4-week-old C57BL/6J oocyte-donor female mice were superovulated with intraperitoneal injections of PMSG (0.1 ml) and hCG (0.1 ml) after 48 h. Capacitated sperm suspensions were prepared from either control or leptin-deficient (*ob/ob*) mice aged 6 weeks and transferred to the centre of prepared drops in a fertilization dish containing oocytes. Note control (*wt/wt*) and leptin-deficient (*ob/ob*) male sperm donors were littermates derived from heterozygous crosses (*wt/ob*) and thus near-isogenic. After insemination of the oocytes with isolated sperm, sperm–oocyte fertilization dishes were cultured at 37 °C under 5% CO$_2$ for 3–4 h. After 3–4 h of incubation, the presumptive zygotes were washed 3–4 times with 150 μl drops of M2 medium and cultured for 4.5 days in four-well NUNC plates with 500 μl of KSOM medium supplemented with amino acids (KSOMaa) at 37 °C in a 5% CO$_2$ and 5% O$_2$ incubator. After 4.5 days, each blastocyst-stage embryo was collected in 0.2 ml PCR tube strips (containing 2 μl of lysis buffer + 1 μl of dNTPs + 1 μl of oligo-dT primer) for single-embryo RNA-seq analysis (Smart-Seq). Note, lysis buffer composition: 0.5% Triton-X100 (vol/vol) in H$_2$O + 2 U μl$^{-1}$ of SUPERase In RNase inhibitor (20 U μl$^{-1}$; Thermo, AM2694) and Oligo-dT primer: 5′-AAGCAGTGGTATCAACGCAGAGT ACT30VN-3′.

**Histological staining.** Testis tissue collected from 6-week-treated (nABX) or CON sire male mice were fixed in Bouin's fixative, then treated through graded ethanol and clearing agent (xylene), followed by embedding in paraffin. Subsequently, the samples were sectioned

serially at a thickness of 5–7 μm and stained with standard haematoxylin and eosin stain. Testicular morphometric analyses of the seminiferous tubules (that is, spermatogonium, spermatocyte, spermatid and Sertoli cells) and Leydig cells were performed using LMD 7000 laser microdissector microscope (Leica). The quantitative evaluation of the abnormal seminiferous tubules was done as follows: for each sample at least ten sections were analysed with more than 1,000 seminiferous tubules evaluated (normal or abnormal) and the percentage of abnormal tubules was calculated accordingly. For this analysis, the images were take using NanoZoomer S60 Digital slide scanner (Hamamatsu Photonics, C13210-01).

**Immunofluorescent staining.** Placenta samples collected at E18.5 were fixed in 4% paraformaldehyde at 4 °C, followed by embedding in paraffin and sectioned into 7-μm-thick slices. Sectioned samples were mounted on slides and dried at 40 °C on hot plate. The dried sections were subjected to heat-induced antigen retrieval using citrate buffer at pH 6.0, incubated overnight at 4 °C with anti-mouse VE-cadherin (Thermo Fisher Scientific; eBiosciences. catalogue no.14-1441-81; 2.5 μg ml$^{-1}$) and detected by anti-Rat-Alexa Fluor 568 (Thermo Fisher Scientific; catalogue no. A-11077; 4 μg ml$^{-1}$). Nuclei were counterstained with 4′,6- diamidino-2-phenylindole (DAPI) (Thermo Fisher Scientific; catalogue no. D1306; 5 μg ml$^{-1}$). Slides were washed before being mounted with ProLong Glass (Thermo Fisher Scientific; catalogue no. P36983) and imaged on a Leica THUNDER Imager Live Cell with a ×20 objective (numerical aperture = 0.8). QuPath v.0.2.1 image analysis software was used to measure areas of labyrinth zone and whole placenta.

Placental vessels at E13.5 and E18.5 were stained with isolectin B4. Briefly, sagittal 7 μm sections of placenta samples were deparaffinized with xylene and rehydrated through decreasing ethanol concentrations. Slide sections were subjected to heat-induced antigen retrieval for 10 min in 10 mM citrate pH 6.0 buffer. Thereafter, these sections were permeabilized with 0.3% Triton-X100, blocked in 5% donkey serum and incubated overnight at 4 °C with a rabbit Isolectin HRP (Sigma, catalogue no. L5391) at 2 μg ml$^{-1}$. Chromagen detection was performed using reagents from an ABC DAB kit (Vector Laboratories no. PK-6100) as per manufacturer's instructions

## Macromolecule analysis

**ELISA assay.** ELISA kits were used to determine the concentration of a specific target protein in biological samples. Frozen plasma and testis samples were used for the measurement of leptin (Lep) in F$_0$ sire males, whereas F$_1$ placenta tissues were used for the measurement of placental growth factor (Plgf) and soluble fms-like tyrosine kinase-1 (sFlt1). Mouse Leptin ELISA kit MOB00, mouse PlGF-2 ELISA Kit MP200 and mouse VEGFR1/Flt-1 ELISA Kit MVR100 (R&D Systems), were used for quantification of Lep, Plgf and sFlt1 following the manufacturer's instruction. Reagents, standard curve dilutions and samples were prepared as described on the kits and all samples, standards and control were assayed in technical duplicate, with several biological replicates. To quantify the target protein concentrations, the data generated were interpolated by nonlinear regression model curve fit.

To quantify plasma leptin, an aliquot of plasma samples sourced from F$_0$ sire male blood and stored in −80 °C were thawed in ice, diluted 20-fold with Calibrator Diluent and assayed according to manufacturer's instruction.

To quantify leptin, Plgf and sFlt1 in tissue, testis or placenta samples were thawed on ice and washed in PBS. This was followed by homogenization with pestle 'A' (about ten strokes) and pestle 'B' (about 20 strokes) in 2 ml of Dounce tissue grinder containing 1 ml of RIPA buffer (Sigma-Aldrich) mixed with protease inhibitor cocktail, kept in ice for 5 min with gentle agitation and centrifuged at 15,000 rpm for 20 min at 4 °C. After centrifugation an aliquot of clear supernatant was used for ELISA assay.

To screen for antibiotic residues and test whether nABX or its residues are detectable in distal tissues or the circulatory system we took independent and complementary approaches.

**Mass spectrometry.** Samples for the quantification of antibiotics in testis tissues were prepared as described in the section on Metabolomics below. Extracted and dried samples were resuspended in 20 µl of methanol:water (1:1) and 10 µl of samples were directly injected into an Agilent 6550 iFunnel qToF mass spectrometer. Calibration curves (twofold dilutions) were prepared using chemical standard resulting in the following ranges of injected amounts: bacteriocin (5.6–90.0 pmol), primaricin (7.0–112.5 pmol) and neomycin (5.3–85.0 pmol). Quantification of antibiotics was performed using the MassHunter Quantitative Analysis Software (Agilent Technologies, v.10.0) and limit of detection was determined by linear regression using R.

**Functional residue screening.** PremiTest kit (R-Biopharm) was used for the determination of nABX or avaABX residues in testis samples stored at −80 °C, which was performed in accordance with the manufacturer's instructions for use in meat. The principle of the test is based on thermophilic bacterium spores (*Bacillus stearothermophilus*) which are highly sensitive to several antibiotics and sulfa compounds. Briefly, testis collected from 6-week-treated (nABX or avaABX) and untreated (CON) sire males were thawed in ice and washed in 1× PBS, followed by homogenization using 2 ml of Dounce tissue grinder containing 200 µl of 1× PBS with pestle 'A' (about ten strokes) and pestle 'B' (about strokes). Herein, sterilized water was used as negative control, whereas nABX cocktail diluted in water was used as a positive control at a concentration of the minimum residue detection limit (0.5 µg ml$^{-1}$). From the homogenized sample solution, 100 µl was pipetted onto the agar in the ampoule and allowed to stand at room temperature for 20 min for prediffusion. After the prediffusion, the sample solution was flushed out of the ampoule by washing twice with water and covered with foil to avoid evaporation. Subsequently, the test ampoules were incubated in Eppendorf block heater at 64 ± 0.5 °C for approximately 3–3.5 h until the negative control turned from purple to yellow. At this stage, ampoules were removed from the block heater and results interpreted on the basis of the kit bicolour indicator.

## Microbiota analysis
**Faecal sample collection.** Fresh faecal samples were collected from each individually housed $F_0$ sire males at several different time points: (1) before nABX administration at day 0; (2) at the end of the 6 week treatment before setting up mating; and (3) during the recovery period following nABX withdrawal. Faecal sample from dams and $F_1$ offspring were collected at weaning stage (P21). To collect fresh faecal samples, each parent or offspring for which faeces were to be collected was put into clean, bedding-free autoclaved cages with food and water for 2–3 h, faecal pellets collected with sterilized tweezer and stored immediately in −80 °C freezers until DNA extraction for microbiome profiling.

**Microbial DNA extraction.** All samples were stored at −80 °C until processing by 16S rRNA sequencing. Microbial DNA extraction from faecal specimens (about 200 mg from $F_0$ parent and about 100 mg from $F_1$ offspring) was performed using QIAamp PowerFecal Pro DNA Kit (QIAGEN) according to the manufacturer's instructions. Briefly, faecal samples processed with the DNA extraction kit were added to a PowerBead beating tube and rapidly homogenized on a Vortex Adapter (QIAGEN catalogue no. 13000-V1-24) using Vortex-Genie 2 mixer (Scientific Industries). Once cells are lysed and potential inhibitors removed, total genomic DNA is captured on a silica membrane, washed and eluted for downstream gut microbiota profiling.

**Determination of 16S rRNA gene copy number.** The relative abundance of 16S rRNA gene copy number in mice faeces was determined from the standard curve generated on the basis of serial dilutions of the control sample by quantitative real-time PCR using F341 5'-CCTACGGGAGGCAGCAG-3' and R534 5'-ATTACCGCGGCTGCTGG-3' primers, as described previously[59].

Targeted amplification of the 16S rRNA V4 region (primer sequences F515 5'-GTGCCAGCMGCCGCGGTAA-3' and R806 5'-GGACTACHVGGGTWTCTAAT-3' (ref. 60), was performed using the KAPA HiFi HotStart PCR mix (Roche) in a two-step barcoded PCR protocol (NEXTflex 16S V4 Amplicon-Seq Kit; Bioo Scientific) with minor modifications from the manufacturer's instructions. PCR products were pooled, purified using size-selective SPRIselect magnetic beads (0.8 left-sized) and then sequenced at 2 × 250 base pairs (bp) on an Illumina MiSeq (Illumina) at the Genomics Core Facility, European Molecular Biology Laboratory, Heidelberg.

Raw 16S rRNA reads were trimmed, denoised and filtered to remove chimaeric PCR artefacts using DADA2 (ref. 61). The resulting amplicon sequence variants were then clustered into operational taxonomic units (OTUs) at 98% sequence similarity using an open-reference approach: reads were first mapped to a preclustered reference set of full-length 16S rRNA sequences at 98% similarity using MAPseq[62]. Reads that did not confidently map were aligned to bacterial and archaeal secondary structure-aware rRNA models using Infernal[63] and clustered into OTUs with 98% average linkage using hpc-clust[64], as described previously[65]. The resulting OTU count tables were noise filtered by asserting that samples retained at least 1,000 reads and taxa were prevalent in at least two samples; these filters removed 58% of spurious OTUs but only 0.09% of total reads from the dataset.

Local sample diversities were calculated as OTU richness, exponential Shannon entropy and inverse Simpson index (corresponding to Hill diversities of order 0, 1 and 2 (ref. 66) as average values of 100 rarefaction iterations to 5,000 reads per sample. Between-sample community diversity was calculated as Bray–Curtis dissimilarity. Trends in community composition were quantified using ordination methods (principal coordinate analysis, distance-based redundancy analysis) and tested using permutational multivariate analysis of variance (PERMANOVA[67], as implemented in the R package vegan[68].

## Cohousing to measure microbiota cross-transfer during mating
To directly test whether microbiota cross-transfer occurred during the mating window period, we sampled faeces at two time points In this 4 day experiment, males from each group (CON and 6-week nABX) were cohoused at a 1:1 ratio in a single cage with a treatment-naive 6-week-old female C57BL/6J mouse continuously, regardless of plug positivity (single breeding pairs). During the mating period, both groups were maintained on regular diet and sterile water ad libitum. We sampled faeces (as above) at two time points: at day 0 precohousing to determine baseline microbiota and at day 4 postmating (denoted as pre- and post-mating, respectively) to examine the effect of female exposure/cohabitation with nABX males. Furthermore, to determine whether microbiota transfer occurs at mating, we sampled a copulatory plug from the female's vaginal area directly following mating. Using swabs, control samples were collected from the working area and sterile sharp-toothed tweezers were used to collect plug samples.

## RNA sequencing (transcriptomics)
**mRNA-seq.** For messenger RNA sequencing, total RNA was extracted from four different tissue types: sire testis, $F_1$ placenta, $F_1$ brain and $F_1$ (BAT), using RNeasy Protect Mini Kit (Qiagen) following the manufacturer's instructions. The quantity and quality of total RNA was evaluated on a NanoDrop Spectrophotometer and Agilent TapeStation system (Agilent) to ensure RNA integrity number > 8.5. For library preparation, mRNA was first enriched using the NEBNext Poly(A) mRNA magnetic isolation module with 1 µg of input RNA. Purified mRNA was subsequently prepped into stranded libraries using the NEBnext Ultra II directional RNA library prep kit, following all manufacturer's guidelines.

Amplified libraries were multiplexed and sequenced on an Illumina NextSeq 500 (PE40).

**Small RNA-seq.** For sperm small RNA analysis, total RNA was isolated using miRVana microRNA isolation kit (Thermo Fisher Scientific). The procedure was performed according to the manufacturer's instructions with a slight modification at the cell lysis step to ensure complete lysis of sperm. Briefly, purified sperm samples were thawed on ice and then the miRVana lysis buffer was added for 5 min at room temperature, with 40 mM dithiothreitol and 12 μl ml$^{-1}$ of proteinase K subsequently added into the sample. The sample was then incubated at 56 °C for 20 min on a thermoshaker at 450 rpm. After this step, miRNA homogenate was added to the lysate and the kit instructions recommended for total RNA isolation were followed. Small RNA libraries were prepared from 1 μg of this total RNA using the Illumina Truseq small RNA library preparation kit according to the manufacturer's instructions. Libraries were sequenced on an Illumina Hiseq 2000.

**Single-cell RNA-seq (10X).** Testicular single-cells resuspended in PBS + 0.04% BSA were adjusted at a concentration of about 1,100 cells μl$^{-1}$ for loading on the 10x Chromium chip. Cell capturing and library preparation was carried out as per kit instructions (Chromium Next GEM Single Cell 3′ v.3.1 (Dual Index) User Guide). In brief, about 6,000 cells were targeted for capture per sample; after complementary DNA synthesis, 12–14 cycles were used for library amplification. The resultant libraries were size selected, pooled and sequenced using Illumina NextSeq 2000 P2 flowcell (100CYC) sequencing protocol.

**Hybridization in situ sequencing.** The protocol was followed as described in ref. 69. Briefly, mRNA transcripts were targeted from testis tissue cryosectioned at 10 μm thickness mounted on SuperFrost Plus adhesion slides. Sections were fixed in 3% formaldehyde, permeabilized with 0.1 M HCl for 5 min and washed with PBS. By applying SecureSeal Hybridization Chambers (Grace Bio-Labs) around tissue sections, mRNA was reverse transcribed overnight at 37 °C using random hexamers, RNase inhibitor and reverse transcriptase (BLIRT). Tissue sections were fixed for 40 min following reverse transcription, washed with PBS, phosphorylated padlock probes (PLPs) were hybridized at a final concentration of 10 nM/PLP and ligated with Tth Ligase (BLIRT) and RNaseH. After sections were washed with PBS, rolling circle amplification was performed overnight at 30 °C using phi29 polymerase and exonuclease I (Thermo). After incubation, reagents were removed, samples washed twice in PBS and the SecureSeal chamber was removed with forceps, followed by hybridizing Bridge-probes (10 nM) for 1 h at room temperature in hybridization buffer (2× SSC, 20% formamide) in the dark, on a rocker. Following this, readout detection probes (0.1 μM) were hybridized in the dark at room temperature for 1 h in hybridization buffer, washed twice with PBS and stained with DAPI (0.5 μg ml$^{-1}$) for 5 min at room temperature. Sections were washed with PBS and mounted with about 20 μl of Fluoromount-G Mounting medium, overlayed with a cover slip and stored at 4 °C until imaged. Images were taken with a slideview vs200 slide scanner (Olympus) and analysed using TissUUmaps.

**TaqMan assay for small RNA.** Independent males from the small RNA-seq samples were used to extract purified sperm total RNA with the miRVana microRNA isolation kit, applying minor modifications to ensure complete lysis. Quantification of miRNA and tRNA was performed using predesigned (miR-141-3p) and custom-designed (tRNA-Gly-GCC-2) TaqMan assays, according to the manufacturer's protocol (Applied Biosystems) using 7 ng of total RNA from CON ($n = 4$) and nABX ($n = 4$) mice. The quantitative PCR with reverse transcription was performed in 15 μl reactions using TaqMan Fast Advanced Master Mix, following the standard programme (10 min at 95 °C; 15 s at 95 °C; 1 min at 60 °C, for 40 cycles). snoRNA202 (Applied Biosystems) was used as the endogenous control for normalization of miRNA and tRNA levels. Relative quantities were normalized to endogenous control values and foldchange calculated by means of the 2$^{-\Delta\Delta Ct}$ method. Amplification efficiency of TaqMan probes was confirmed by serial dilution of template runs.

**Single-embryo RNA-seq (Smart-Seq).** Single-embryo RNA-seq was carried out using the Smart-Seq protocol, as described in ref. 70. For quality control and data preprocessing, raw reads were aligned using STAR v.2.7.10a (ref. 71) at default parameter settings to mm10 (GRCm38) primary genome assembly. The data were quantified using the feature-Counts module of Subread v.2.0.1 (ref. 72). Differential expression and PCA were performed in R (v.4.1.2). Differential expression was performed by passing the raw counts into the DESeq2 (v.1.34.0) package[73]. The vst function was used to normalize the counts at default settings and the top 500 variable genes were used for PCA. For gene ontology analysis, the Metascape web application (https://metascape.org/gp/index.html) was used to infer enrichment of gene ontology terms[74]. The list of genes to be tested was uploaded to Metascape as a text file. The 'Express Analysis' option was selected after setting both the 'Input as species' and 'Analysis as species' parameters to *M. musculus*.

## DNA sequencing

**Whole-genome bisulfite-seq.** Extraction of total sperm DNA was performed using the DNeasy Blood & Tissue Kit (QIAGEN Hilden) optimized for purification of sperm genomic DNA. The DNA extraction was performed as follows. Frozen pure sperm samples stored at −80 °C were thawed in ice; 100 μl of DNA extraction buffer (20 mM Tris Cl pH 8, 20 mM EDTA, 200 mM NaCl, 4% SDS) containing 80 mM dithiothreitol and 12 μl ml$^{-1}$ of proteinase K was added to the sample; and then incubated at 56 °C on Eppendorf thermomixer until complete sperm lysis was assured (about 1 h), shaking occasionally during incubation to disperse the sample. After incubation, the user-developed protocol DY03 (QIAGEN) for purification of total DNA from animal sperm was applied. BS-Seq libraries were constructed according to the manufacturer's instructions using TruSeq DNA Methylation Library Prep Kit (Illumina). Amplified libraries were multiplexed and sequenced on an Illumina NextSeq 500 (PE75).

**De novo gDNA-seq.** The gDNA was extracted from control or nABX-derived F$_1$ offspring (normal or SGR) liver using DNeasy Blood & Tissue Kit (Qiagen) following the manufacturer's instructions. The quantity and quality of DNA was evaluated on a NanoDrop Spectrophotometer and Agilent TapeStation system (Agilent) to ensure DNA integrity number > 7. For library preparation NEBNext Ultra II DNA Library Prep kit was used, following all manufacturer's guidelines. Amplified libraries were multiplexed and sequenced on an Illumina Hiseq 4000 (PE150) to appropriate depth.

## Metabolomics

**Untargeted metabolomics measurements.** All chemicals for liquid chromatography–mass spectrometry (LC–MS) analysis including water and acetonitrile (LC–MS grade) were purchased from Fisher Scientific. Standards for online mass calibration were purchased from Agilent Technologies.

**Sample preparation.** Testis samples were collected from both CON and nABX-treated sire males across the three breeding time points: immediately after 6 weeks of treatment (11-week-old mice), 4 weeks after treatment withdrawal (15-week-old mice) and 8 weeks after treatment withdrawal (19-week-old mice). Immediately after collection, samples were snap-frozen in liquid nitrogen and stored at −80 °C. On retrieval, 300 μl of ice-cold solvent mixture (acetonitrile:methanol:water, 2:2:1) and two Tungsten Carbide Beads, 3 mm from Qiagen were added to each testicle sample. Samples were homogenized using Qiagen tissueLyser II

at 30% strength for 5 min. The lysed samples were centrifuged at 12700 RCF at 4 °C for 10 min. An equal volume (125 µl) of extraction supernatant was transferred to two Nunc 96-well, V-shape plates and evaporated in the Genevac centrifugal concentrator (Genevac) at 25 °C for 2 h. Concentrated samples were stored at −80 °C and resuspended in 20 µl of same solvent mixture for LC–MS analysis.

**LC–MS measurements.** Chromatographic separation was performed using an Agilent InfinityLab Poroshell 120 EC-C18, 3.0 × 150 mm², 2.7 µm column and an Agilent 1290 Infinity II LC system coupled to a 6550 iFunnel qToF mass spectrometer. Column temperature was maintained at 45 °C with a flow rate of 0.4 ml min⁻¹. The following mobile phases were used: mobile phase A—water with 0.1% formic acid; and mobile phase B—acetonitrile with 0.1% formic acid. The 5 µl of sample were injected at 5% mobile phase B, maintained for 0.10 min, followed by a linear gradient to 30% B in 0.5 min, followed by a linear gradient to 95% B in 10 min and maintained at 95% B for 1 min. The column was allowed to re-equilibrate with starting conditions for 3 min before each sample injection. The mass spectrometer was operated in both positive and negative scanning mode (50–1,700 $m/z$) with the following source parameters: VCap, 3,500 V; nozzle voltage, 2,000 V; gas temperature, 275 °C; drying gas, 13 l min⁻¹; nebulizer, 45 psi; sheath gas temperature, 275 °C; sheath gas flow, 12 l min⁻¹; fragmentor, 130 V; and skimmer, 0 V. Online mass calibration was performed using a second ionization source and a constant flow (10 µl min⁻¹) of reference mass solvent (119.0363 and 1033.9881 $m/z$ for negative operation mode and 121.0509 and 922.0098 for positive operation mode, respectively). Each sample was measured separately in both positive and negative ionization modes.

**Metabolic feature extraction.** The MassHunter Qualitative Analysis Software (Agilent Technologies, v.10.0) was used to extract metabolic feature from the acquired LC–MS data. The following settings were applied: peak filter of absolute height: 5,000 counts, limit assigned charge states to 1, only ±H⁺ charged molecules were included with compound quality scores greater than 80%. Peak alignment and identification were carried out using Mass Profiler Professional (Agilent, v.15.1) with default parameters: mass tolerance of 2 mDa or 20 ppm and retention time tolerance of 0.2 min or 2%. Extracted and aligned features were exported as .csv file for further data analysis

**Statistical analyses**
Statistical analyses was performed using Graphpad Prism v.8.4.3 graphical software and in R v.3.6.2. Significant differences in $F_1$ offspring body weight and phenotypes were determined with 'nested' analyses, whereby the replicate number is dictated by the number of uniquely exposed fathers ($N$), not by the number of offspring ($n$), which is greater. Thus, although we typically generate $n > 150$ offspring per experiment to robustly capture effect size and partially penetrant phenotypes, statistically speaking each individual derived from the same father is hierarchically nested into a single $N$ value. For example, a nested $t$-test compares the means of two unmatched groups (all $F_1$ offspring from control or dysbiotic fathers), for which there is a nested factor in those treatment groups (shared father amongst each litter). This is necessary as using individual offspring as independent variables in intergenerational studies will lead to inflated alpha error rates and spurious significance. Testes-to-body weight ratio, fetoplacental ratio and labyrinth zone were analysed by two-tailed unpaired $t$-test. Odds ratios (ORs) and 95% confidence intervals (CIs) were computed using with Baptista–Pike method and the statistical significance of the ORs determined using chi-squared test. Kaplan–Meier method was applied to generate survival analysis curves compared by the log-rank (Mantel–Cox) test. ELISA calibration curves were interpolated with a Hyperbola ($X$ is concentration) nonlinear regression model fit ($R^2 > 0.99$ was acceptable curve fit).

**Bioinformatics analyses**
**Single-cell RNA-seq alignment and mapping.** Raw reads were aligned and mapped using the count module in 10x Genomics Cellranger 6.1.2 (ref. 75) to the mm10 transcriptome assembly (2020-A) with default parameters.

**Single-cell RNA-seq quality control.** All subsequent steps were performed in R (v.4.1.2) using the Seurat package[76]. First, cells were filtered on the basis of three parameters—number of unique molecular identifiers (nCount_RNA; 1,000:5,000), number of unique genes detected (nFeature_RNA; 500:5,000) and mitochondrial rate (percent.mito) and ribosomal rate (percent.ribo; 0:20)—as described below. The nABX and CON samples were then clustered separately using the default Seurat clustering approach. In both conditions, clusters having no uniquely expressed genes (using the FindMarkers function with logfc.threshold = 0.5 and min.pct = 0.5) were discarded.

**Single-cell RNA-seq integration.** The nABX and CON samples were then integrated using the canonical correlation analysis approach at default parameter settings as recommended by the Seurat package.

**Single-cell RNA-seq cell-type annotation.** The integrated dataset was then clustered and annotated using uniquely expressed marker genes. As above, clusters showing no uniquely expressed genes were ignored. To define more fine-grained annotations, the somatic and germ cells were split into different Seurat objects and clustered separately.

**Single-cell RNA-seq cell-type-specific differential expression.** For each cell type identified in the dataset, the FindMarkers function was used to identify DEGs between nABX and CON cells using logfc. threshold = 0.25 and p_val_adj < 0.01.

**RNA-seq quality control and data preparation.** Raw reads were quality trimmed using Trim Galore (0.4.3.1, -phred33 --quality 20 --stringency 1 -e 0.1 --length 20). These were mapped to the mouse mm10 (GRCm38) genome assembly using RNA Star (2.5.2b-0, default parameters except for --outFilterMultimapNmax 1000) and reads with a MAPQ score less than 20 were discarded to ensure that only unique-mapping high-quality alignments were used for analysis of gene expression. The data were quantified using the RNA-seq quantification pipeline for directional libraries in seqmonk software to generate $\log_2$ reads per million (RPM) or gene-length-adjusted (RPKM) gene expression values.

**RNA-seq differential analysis.** DEGs were determined using the DESeq2 package (v.1.24.0), inputting raw mapping counts and applying a multiple-testing adjusted P value (false discovery rate (FDR)) < 0.05 significance threshold. An extra foldchange filter of more than 2 was applied to generate final DEGs.

**RNA-seq principal component analysis.** Principle component analyses of transcriptomes were computed in seqmonk and R statistical software using all expressed genes as input. These were defined as having an RPKM > 0.1 in at least two replicates across all assayed samples.

**RNA-seq gene ontology class enrichment.** DEGs or gene lists of interest, were inputed into the STRING v.11.0 database[77] and extracting enrichment analysis related to Reactome and KEGG pathways, filtering by FDR rank.

**Small RNA-seq quality control and data preparation.** Adaptors were removed using fastx-clipper; fastq files were converted to fasta using a custom perl script and reads of 18–33 nucleotides in length were retained using a custom perl script. Reads were aligned to the mouse genome version mm10 using bowtie, reporting only the best alignment

and requiring 0 mismatches (parameters -v 0 -k 1 -- best --sam). Alignment sam files were converted to bam files using samtools v.1.9 and bam files were converted to bed files using bedtools v.2. To quantitate miRNAs intersectBed -c was used to count the number of reads overlapping the positions of the known *M. musculus* miRNAs (from miRbase www.miRBase.org). The tRNAs were obtained by using intersectBed -c to count the number of reads overlapping a bed file documenting predicted mouse tRNA coordinates downloaded from the tRNA scan database (http://gtrnadb.ucsc.edu/genomes/eukaryota/Mmusc10/). The piwi-interacting RNA coordinates were taken from ref. 78 and converted to mm10 using liftover. The piRNAs were quantitated by selecting RNAs between 26 and 32 nucleotides long with a U as the first nucleotide and intersecting these RNAs with the piRNA coordinates using intersectBed -c.

**Small RNA-seq differential analysis.** To investigate significant differences, data were processed using DESeq2 and the negative binomial test used to identify significant differences after Benjamani–Hochberg multiple test correction.

**Whole-genome bisulfite-seq.** Raw fastq sequences were quality- and adaptor-trimmed using Trim Galore (0.4.3.1) and reads aligned to mm10 using Bismark (0.20.0), discarding the first 8 bp from the 5′ end and the last 2 bp from the 3′ of a single-end reads. Cytosine methylation status was extracted from mapped reads using the Bismark methylation extractor tool. Genome-wide methylation calls were analysed using Seqmonk software (1.44.0) with five biological independent replicate datasets for each condition. To identify DMRs, the genome was first binned into sliding tiles containing 50 consecutive CpGs and their methylation status determined using the DNA methylation pipeline. DMRs were identified by running read-depth sensitive logistic regression ($P$(adj) < 0.05), with minimum of ten reads and applying a threshold for an absolute change in DNA methylation of 20%. The methylation level at specific genomic features (for example, imprints) was calculated using the DNA methylation pipeline in Seqmonk over target features.

**De novo gDNA-seq.** Alignment reads were trimmed using Trim Galore (v.0.6.3)[79], then the first ten 5′ bases of both reads removed with Cutadapt (v.2.3)[80]. Reads were aligned to *M. musculus* refence genome (mm10) using bwa mem (BWA-0.7.17)[81] before filtering with samtools (v.1.10)[82] view with the flags '-h -F 256 -f 2 -q 30' and deduplicated with Picard toolkit (v.2.9.0) MarkDuplicates[83]. SNPs and small INDELs were called using GATK (v.4.1.6.0)[84] HaplotypeCaller. Variants were filtered to remove those with a PHRED-called site quality (QUAL) of less than 30, an allele frequency of less than 0.2 or low site coverage (sliding scale) in more than one individual. Variant functional region was annotated using ANNOVAR (2020Jne07) annotate_variation.pl[85]. Structural variants were called with Delly2 call (v.0.8.7)[86].

**Metabolomics quality control and data preparation.** All statistical analyses and plotting were performed in R v.3.6.2 (ref. 87). To exclude bad sample injection from downstream analyses, the sum of all extracted metabolite features (TIC) was compared between samples (mean = $1.600 \times 10^{10}$, range = [$1.419 \times 10^{10}$; $1.808 \times 10^{10}$]) and samples were excluded, if their TIC was not within 3 s.d. from the mean value (0 excluded samples). Missing data were imputed to a fixed threshold, set at 5,000 counts. Exact duplicated features or features falling within (1) a 0.002 amu (absolute threshold) or 20 ppm (relative threshold) window and (2) a 0.15 min (absolute threshold) or 2% (relative threshold) retention time window, were considered to be split peaks and therefore collapsed together. Correlation between testis weight and animal body weight was computed, to verify that there was no significant correlation between the two values (cor = 0.1870, $P$ = 0.2477). Therefore, testis weight $Z$-scores were used to normalize area-under-the-curve intensity values, to consider variation in signal intensity derived from

testis size. Features (1) at zero variance; (2) being singletons; and (3) present in less than 75% of the samples for each class, were removed. Finally, feature tables derived from positive and negative mode were collapsed together after checking for exact duplicated features or if feature falling in a 0.002 Da or 20 ppm window existed; if so, the feature with higher average intensity was retained and both annotations were retained. The resulting table included all 30 samples and a total of 10,582 metabolic features.

**Metabolomics metabolite annotation.** Annotation was retrieved from the Human Metabolome Database (HMDB, https://hmdb.ca/)[88,89], by searching for metabolites with the exact same mass or falling in a 0.002 amu or 20 ppm window from the mono-isotopic mass of a metabolite. When present, several annotations were retrieved. Only features being annotated were retained for the downstream analysis. Moreover, for each feature, metabolite class and superclass were retrieved, when present, from the HMDB.

**Metabolomics principal component analysis.** PCA was computed[90] both for the complete dataset and after stratifying it by sampling week for all metabolic features and annotated features only.

**Metabolomics differential analysis.** Statistical significance of the feature intensity differences was assessed using a two-sided t-test (stats::t.test function in R) of log-scaled data and $P$ values were FDR-corrected for several hypotheses testing using the Benjamini–Hochberg procedure (stats::p.adjust function in R with BH parameter). Adjusted $P$ values and foldchanges were visualized using Volcano plots. Mass, retention time, HMDB annotation, class, superclass and composite spectrum were retrieved for all features showing significantly different intensity between the treated and the control group and an absolute foldchange greater than 2.

**Metabolomics metabolite class enrichment analysis.** Metabolite class and superclass enrichment analysis between the treated and control group was calculated using a Fisher's exact test. All $P$ values were FDR-corrected for several hypotheses testing using the Benjamini–Hochberg procedure (stats::p.adjust function in R with BH parameter).

**Reporting summary**

Further information on research design is available in the Nature Portfolio Reporting Summary linked to this article.

## Data availability

The data supporting the findings of this study are available and have been deposited in ArrayExpress relating to RNA-seq (E-MTAB-10034), bisulfite-seq (E-MTAB-10033), gDNA-seq (E-MTAB-10273); 16S rRNA-seq datasets are deposited in ENA (PRJEB43500); metabolomics is deposited in MetaboLights (MTBLS1629). Source data are provided with this paper.

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

**Acknowledgements** We thank EMBL core facilities, particularly E. Perlas (Histology), V. Benes (Genecore) and C. Giradot (GBCS), as well as S. Kandels, for experimental assistance. We thank I. Adams, A. Boskovic and C. Gross for comments on the manuscript. This study was funded by an EMBL programme grant no. 50650 (J.A.H.), the EMBL Interdisciplinary Postdoc (EIPOD) programme under Marie Curie CoFund Actions MSCA-COFUND-FP 664726 (A.A.-D.) and the Medical Research Council MC-A652-5PY80 (P.S.)

**Author contributions** A.A.-D. and J.A.H. conceived the study and designed the experiments. T.S.B.S. and A.A.-D. performed and analysed 16S rRNA-seq experiments. M.D.G. and A.A.-D. performed and analysed testicular histological experiments. A.A.-D. and B.R. performed and analysed single-cell RNA-seq experiments. A.A.-D., S.D., E.M. and M.Z. performed and analysed the metabolomics experiments. A.A.-D. and C.T. L. performed and analysed the whole-genome bisulfite experiments. J.A.H. analysed the whole-genome bisulfite-seq experiments. D.P. and V.P. worked with A.A.-D. on offspring phenotyping and husbandry. J.D. worked with A.A.-D. on placental phenotypic experiments. A.A.-D. and A.P. performed IVF experiments. A.A.-D., S.E. and A.C. conducted and analysed in situ sequencing experiments. A.A.-D., S.G., J.A.H and P.S. performed and analysed small RNA-seq experiments. N.H. and O.B. supervised offspring phenotypes and IVF experiments. A.A.-D. and J.A.H. interpreted the data and wrote the manuscript. P.B. cosupervised and J.A.H. supervised the study.

**Funding** Open access funding provided by European Molecular Biology Laboratory (EMBL).

**Competing interests** The authors declare no competing interests.

**Additional information**
**Correspondence and requests for materials** should be addressed to Jamie A. Hackett.

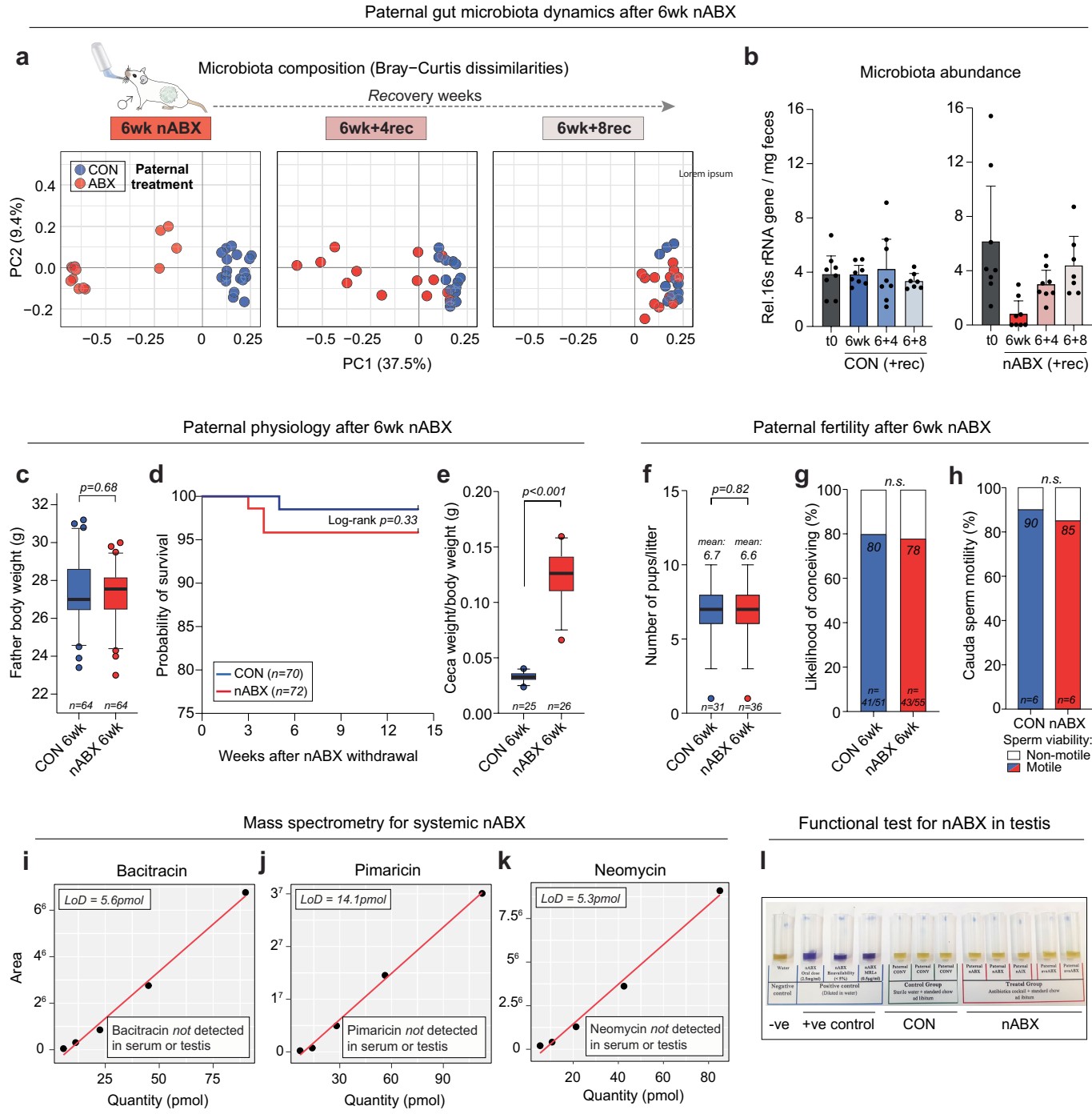

**Extended Data Fig. 1** | See next page for caption.

**Extended Data Fig. 1 | Paternal microbiome and physiological responses to nABX-induced dysbiosis.** (**a**) Principal component analysis (PCA) showing gut microbiota composition (Bray–Curtis) in males after 6 weeks of nABX or in controls. Additional panels show gut microbiota composition after a 4- or 8-weeks recovery period following nABX withdrawal, demonstrating rescue of gut microbiota after 8 weeks nABX withdrawal. (**b**) qPCR quantifying the total abundance of microbes after 6wk nABX and following 4- and 8- weeks recovery (+rec), determined by 16S rRNA. (CON t0 = 8, 6wk = 8, 6wk + 4rec = 8, 6wk + 8rec = 7; nABX t0 = 7, 6wk = 8, 6wk + 4rec = 8, 6wk + 8rec = 7 individuals per timepoint). Error bars indicate S.D. (**c**) Boxplot showings body weights of males after 6wk nABX treatment. Bar represents median and whiskers indicate 5-95th percentile, p-value by unpaired two-tailed t-test. (**d**) Survival curve of males treated with nABX showing no subsequent effect on mortality. *p-value* by Mantel–Cox (log-rank). (**e**) Ratio of ceca to body weight in males treated with 6wk nABX. An enlarged ceca is symptomatic of major dysbiosis and/or reduced microbial abundance, further demonstrating changes in gut communities in nABX males. Bar represents median and whiskers indicate 5-95th percentile, p-value by unpaired two-tailed t-test (CON n = 25 males; nABX n = 26 males) (**f**) Number of pups per litter derived from nABX-treated male sires. Bar represents median and whiskers indicate 5-95th percentile. *p-value* by unpaired two-tailed t-test (CON n = 31 litters; nABX n = 36 litters). (**g**) Likelihood of males siring offspring following 6wk nABX, illustrating no difference in fertility relative to control (CON = 41 conceived from 51 plugs positive; nABX = 43 conceived from 55 plugs positive). (**h**) Sperm viability as judged by scoring the number of motile sperm per million after 6wk nABX exposure (CON n = 6 males; nABX n = 6 males, harvested sperm samples). (**i**–**k**) Mass spectrometry analysis of each component of the nABX cocktail in tissues of treated male mice to confirm they are non-absorbable and do not reach distal tissues. Shown are standard curves demonstrating the high sensitivity and limit of detection (LoD). We were unable to detect (**i**) Bacitracin (**j**) Neomycin and (**k**) Pimaricin in either testis or blood (serum) of nABX-treated mice (**l**) Functional assay for the detection of antibiotic residues in testis of nABX-treated males indicates no bioavailable nABX can be detected (LoD detection threshold is below maximum residual level (MRL)). All assays indicate systemic and testicular responses to nABX are not due to direct chemical interactions with drugs, because nABX remain in the gastrointestinal tract and do not reach the systemic level or the testes.

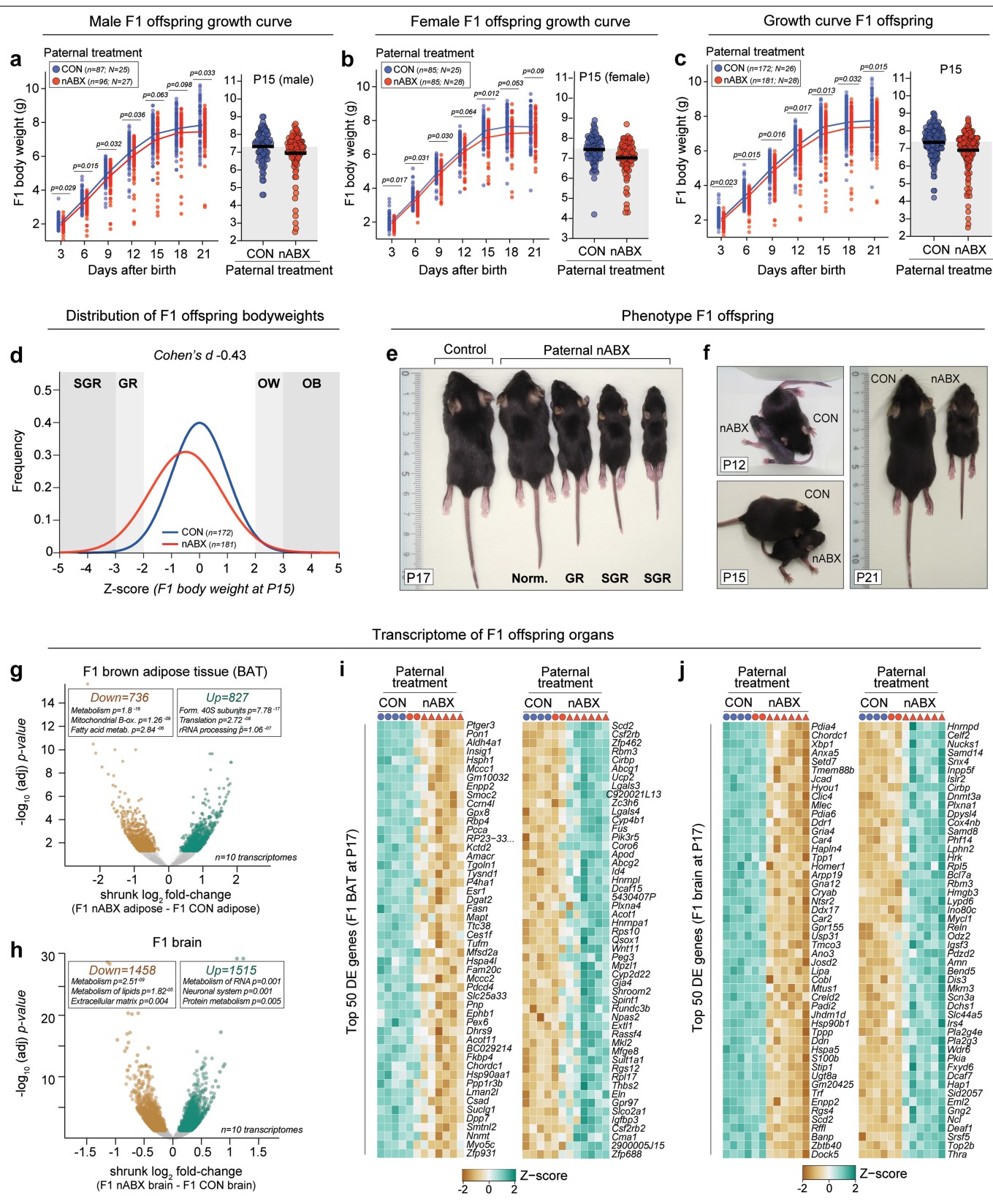

**Extended Data Fig. 2** | See next page for caption.

**Extended Data Fig. 2 | Growth and molecular phenotypes in F1 offspring from nABX-induced dysbiotic fathers.** (**a & b**) Growth curves for F1 offspring fathered by control or nABX-treated sires, stratified for (**a**) male offspring (CON n = 87; nABX n = 96) and (**b**) female offspring (CON n = 85; nABX n = 85). Shown right is a representative higher-resolution timepoint (P15) of offspring body weights. Significance by a hierarchical (nested) two-tailed t-test that considers the number of litters (N) as degrees of freedom (CON = 25; nABX=28 litters). (**c**) Growth curves and P15 for all F1 offspring (male & female) combined (P3 n = 172/181 (CON/nABX), N = 26/28 fathers; P15 n = 164/180 (CON/nABX)). Significance by a hierarchical (nested) two-tailed t-test. (**d**) Distribution of fitted F1 offspring body weights at P15. There is a population shift leading to an increase in severe growth restriction (SGR) individuals amongst nABX-derived offspring. Body-weight categories are determined by Z-score relative to controls: < −3 = severe growth restriction (SGR); −3:−2 = growth restriction (GR); 2:3 = overweight (OW); >3 = obese (OB). (**e**) Representative images showing the range of F1 offspring growth phenotypes from nABX-treated sires at P17, spanning from normal to severe growth restriction (SGR). Obese individuals were not observed. (**f**) Additional phenotype comparisons between control and nABX-derived F1 offspring at indicated postnatal (P) days.

Shown are three independent pairs, each from a different litter, with control and nABX born on same day. (**g**) Volcano plot showing significant DEGs (highlighted ochre for downregulated, green for upregulated) in brown adipose tissue (BAT) of offspring sired by dysbiotic fathers. The top enriched gene ontology terms for up- and down- regulated DEGs are shown above. *p-value* adjusted for multiple testing. (**h**) Heatmap showing the top 50 upregulated and top 50 downregulated genes in BAT from independent F1 offspring. Each column is an F1 individual from either a control or nABX-exposed father; circles indicate no observable phenotype whilst triangles indicate a physiological phenotype (SGR) observed in that individual. Data obtained from 3 litters per paternal treatment group. (**i**) Volcano plot showing significant differentially expressed genes in brain from F1 offspring. P-value adjusted for multiple testing. (**j**) Heatmap showing expression of the top 50 upregulated and downregulated genes in brain of F1 offspring according to paternal treatment. Each column is an F1 individual from either a control or nABX-exposed father; circles indicate no observable phenotype whilst triangles indicate a physiological phenotype (SGR) observed in that individual. Data obtained from 3 litters per paternal treatment group.

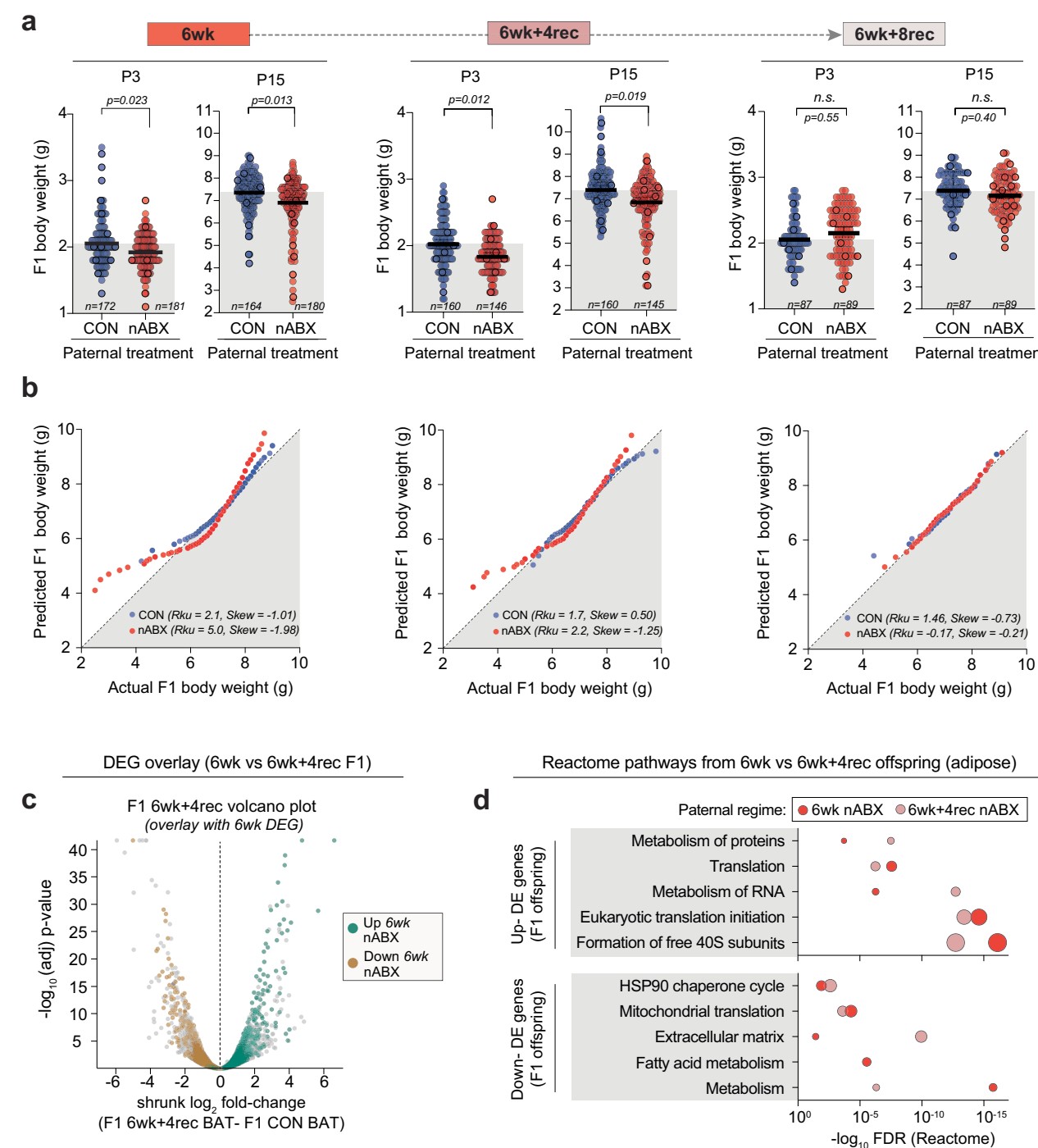

Offspring growth phenotype after restoration of paternal gut microbes

**a**

6wk — 6wk+4rec — 6wk+8rec

DEG overlay (6wk vs 6wk+4rec F1)

Reactome pathways from 6wk vs 6wk+4rec offspring (adipose)

**Extended Data Fig. 3 | Reversion of F1 phenotypes coincident with restoration of preconception paternal gut microbiota.** (**a**) Body weights of F1 offspring derived from males exposed to nABX for 6wk (dysbiotic) and after 4 weeks recovery following nABX withdrawal (6wk + 4rec; still dysbiotic) and 8 weeks recovery (6wk + 8rec; recovered). The same males are used throughout the time course. *p-value* indicates nested (hierarchical) unpaired two-tailed t-test that calculates significance based on the number of treated males (fathers) rather than number of shown offspring. (**b**) QQ normality plots for F1 offspring body weights when sired by 6wk nABX (left), 6wk + 4rec (centre) or recovered 6wk + 8rec fathers (right). The plots indicate that SGR (low weight) offspring from dysbiotic fathers occur at a higher frequency than expected, indicating there is a change in the distribution (in addition to the mean) of F1 body weights, as indicated by excess kurtosis values. This implies a probabilistic

affect that increases the frequency of outliers (SGR). (**c**) Volcano plot of gene expression in adipose of F1 offrpsing from 6wk + 4rec nABX fathers. Overlayed in green/ochre are differentially expressed genes from independent offspring derived from 6wk nABX fathers, indicating an equivalence in expression changes and directionality between cohorts sired from independent dysbiotic fathers. *p-value* adjusted for multiple testing. (**d**) Bubble plots showing gene ontology analysis of differentially expressed genes in F1 SGR offspring sired from 6wk or 6wk + 4rec fathers shows a striking similarity of enrichments. This suggests these independent dent offspring acquire a reproducible molecular response to paternal dysbiosis, whereas no SGR offspring were observed from 6wk + 8rec time points, indicating the effect is robust during the period of paternal gut microbiota perturbation but reverts coincident with recovery.

Mutational load and *de novo* genome sequencing of F1 offspring

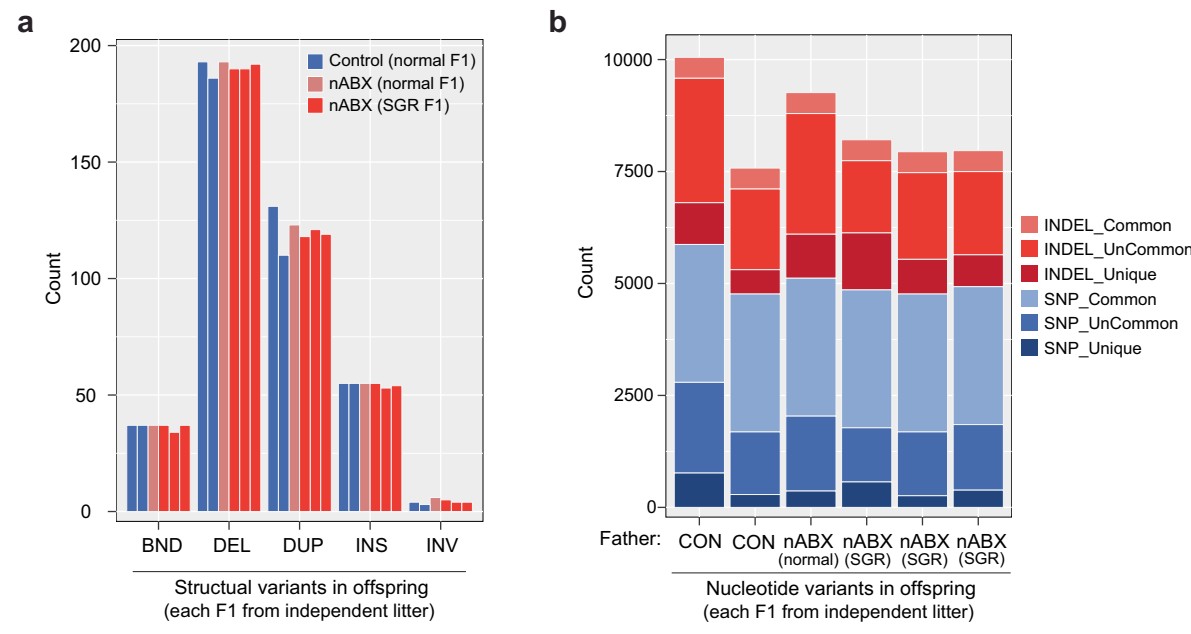

Transgenerational (F2) impact of paternal nABX

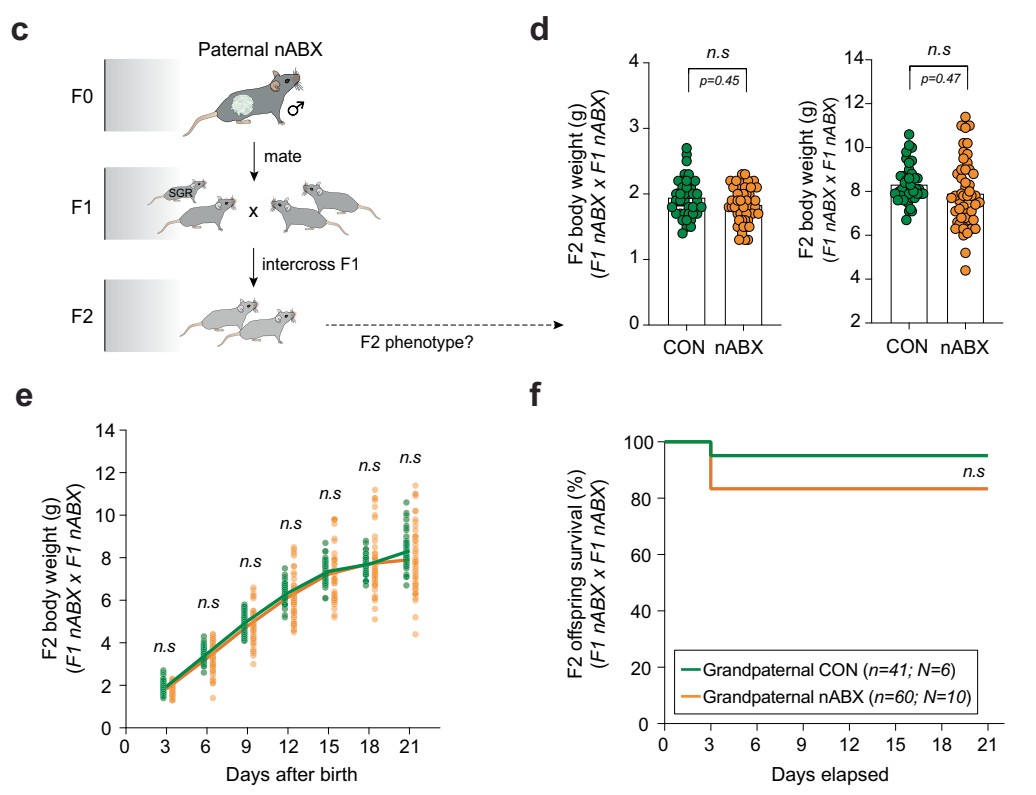

**Extended Data Fig. 4** | See next page for caption.

**Extended Data Fig. 4 | No evidence for increased structural or nucleotide variants in F1 offspring or transgenerational F2 phenotypes.** (**a**) Analysis of structural variants in independent F1 offspring sired by control or nABX-treated fathers reveals no change in their frequency irrespective of father's status or offspring phenotype (normal or severe growth restriction: SGR). Each bar is an independent F1 offspring. (**b**) Analysis of nucleotide variants amongst independent F1 offspring from control or nABX fathers, showing no change in their frequency. These data indicate that the SGR F1 phenotype derived from nABX fathers is not associated with any increased rate of genetic abnormalities. We also did not identify any coding mutations underlying SGR phenotype. Note, most identified indels are 'common' (present in n = 6 (all) offspring), or 'uncommon' (present in n = 2–5 offspring), indicating they represent baseline nucleotide polymorphisms inherent throughout our C57BL/6J mouse colony relative to the reference genome. (**c**) Schematic showing the experimental design. Control or nABX-treated F0 males were mated with naive females and their F1 offspring were intercrossed to examine potential F2 effects. Note, the subset of F1 offspring with a severe growth restriction (SGR) phenotype could not be intercrossed, as they typically exhibited mortality prior to sexual maturity. Because of this reason and irrespective of F2 phenotype, nABX-exposed F0 males are predicted to have reduced F2 (transgenerational) fitness, as judged by number of grand-progeny. This reflects fewer sexually mature F1 offspring, which in turn would produce fewer F2 in absolute terms. (**d**) Dot plot showing neonatal (P3) and post-weaning (P21) body weight of F2 offspring is not altered (F2 n = 39 offspring from 6 intercrossed F1 CON offspring, F2 n = 50 offspring from 10 intercrossed F1 nABX offspring). Significance by nested two-tailed t-test. (**e**) Growth curves of all F2 offspring from grandpaternal control or nABX conditions (CON n = 39 F2 (6 litters); nABX n = 50 F2 (10 litters)). Significance by nested two-tailed t-test. (**f**) Survival plot of F2 offspring. Significance by Mantel–Cox test (log-rank). n = number of F2 offspring, N = number of independent F1 parents.

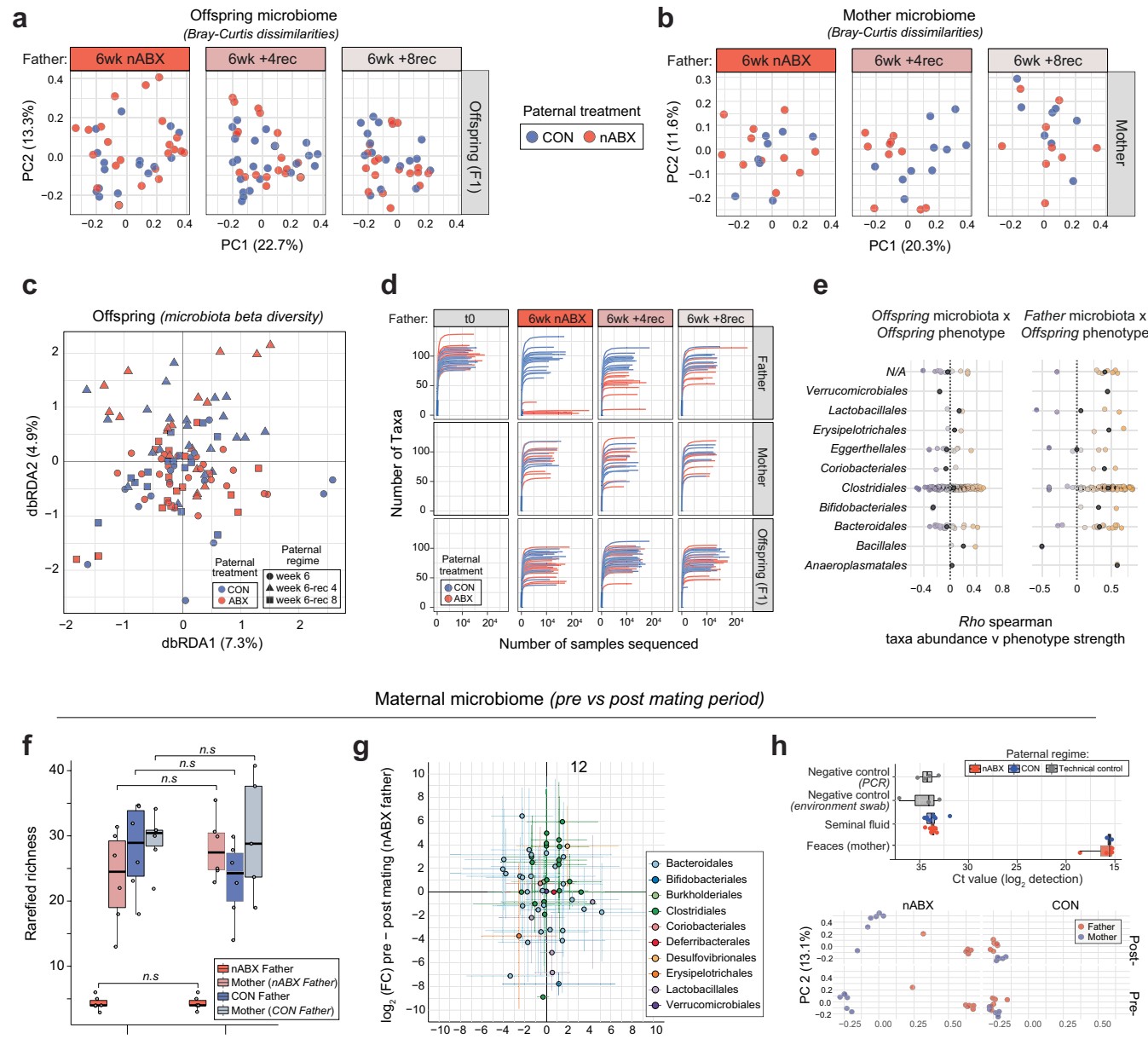

**Extended Data Fig. 5 | No significant changes in maternal or offspring microbiome.** (**a** & **b**) Bray–curtis dissimilarities showing no change in the composition of the gut microbiome in (**a**) F1 offspring sired by nABX fathers or (**b**) in mothers mated with nABX males, indicating transmitted F1 effects are independent of influencing the F1 or maternal microbiota *per se*. (**c**) Microbiota beta diversity confirming no consistent alteration in F1 microbiota as a function of the fathers microbial status. (**d**) Rarefaction analysis in fathers, mothers and offspring stratified by treatment of the father. (**e**) Spearman analysis between *offspring* taxa abundance and offspring phenotype shows no correlation (left), whereas spearman analysis between *paternal* microbiota taxa abundance and offspring phenotype shows associations of specific taxa in fathers with positive or negative correlations to F1 phenotype. Grey dot indicates mean correlation. (**f**) Analysis of microbiota richness of mothers and fathers prior to - and

following − 4d cohousing and mating with either a control or nABX male. Pre- and post-mating faecal samples were collected from both mating pairs (CON = 6; nABX = 6). The results indicate there is no significant change in mothers following 4 days exposure to nABX males. Bar represents median and whiskers indicate 1.5x IQR. (**g**) Foldchange in specific taxa in mothers during pre- to post- mating with either a control or nABX male, revealing high variance and no significant effects linked with nABX males. Faecal samples collected pre- and post-mating (CON = 6; nABX = 6, females). Error bars indicate 95% C.I. (**h**) Upper; qPCR analysis showing unaltered microbiota abundance in mothers post-mating and after 4d exposure to nABX males. Also shown is failure to detect a seminal fluid microbiome ruling out this modality of information transfer (CON = 8; nABX = 7, seminal fluid samples). Error indicates S.D. Lower: Bray–curtis dissimilarities in fathers and mothers from pre- and post-mating periods.

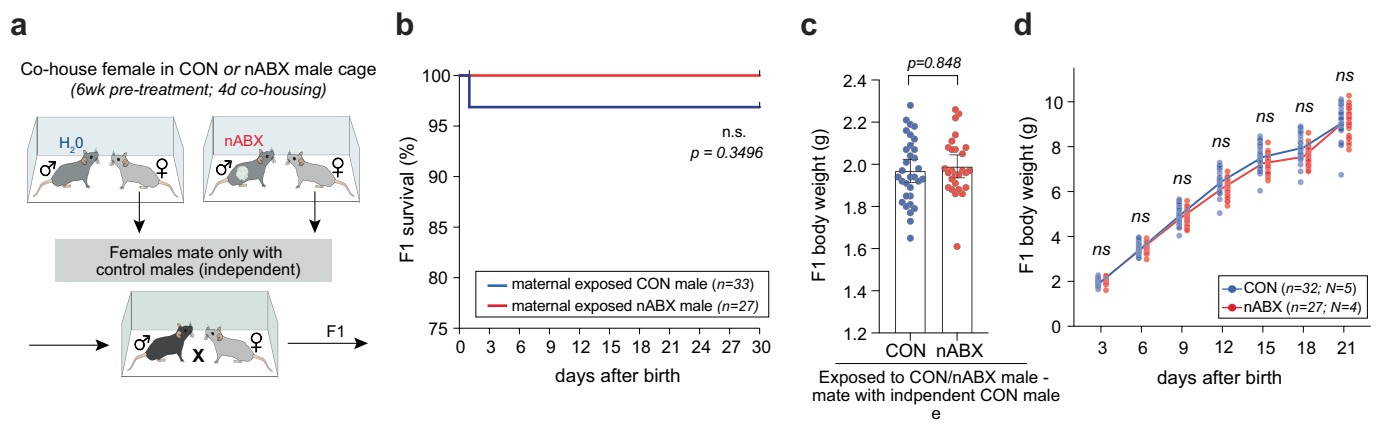

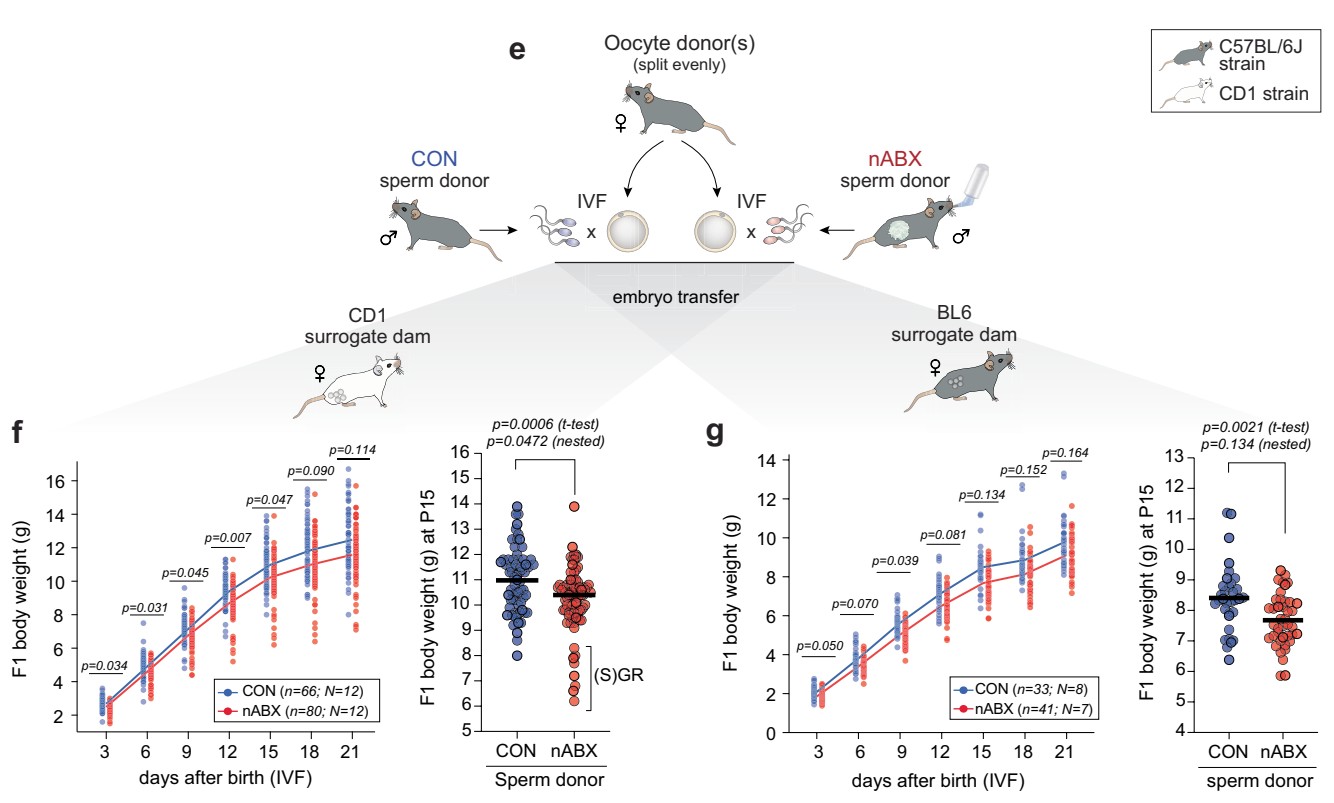

**Extended Data Fig. 6** | See next page for caption.

**Extended Data Fig. 6 | *In vitro* fertilization and cohousing suggest germline transmission directly contributes to F1 phenotypes.** (**a**) Schematic of the cohousing strategy to empirically determine whether prior maternal exposure to an nABX male or his environment influences offspring phenotype independently of germ cells via, for example, coprophagy or microbiome transfer. Females were maintained within a control or nABX male cage for 4 days, which recapitulates the maximum time period used for natural matings throughout the study (mating within 1–4 days). Mating was prevented by removing males during the evening whilst leaving the females within the male environment, including exposure to faeces, microbes and chemical cues. After 4 days, independent control males were used for mating to examine the functional effect of prior cohousing with an nABX males. (**b**) Survival of F1 offspring from mothers mated with control males but pre-exposed to control or nABX males/environments. Significance by Mantel–Cox test (log-rank). (**c**) Neonatal birth weight, *p-value* by nested two-tailed t-test and bar indicate mean with 95% C.I. (**d**) Growth curves of F1 offspring from mothers mated with control males but pre-exposed to control or nABX male environments. *ns* = non-significant by nested two-tailed t-test analysis. (**e**) Schematic of experimental strategy for in vitro fertilization using control or dysbiotic (nABX) sperm donors. Oocytes from C57BL/6J females were pooled and split evenly to be fertilized by freshly harvested sperm from CON or nABX-treated (6wk) C57BL/6J males. IVF from CON and nABX males was performed in parallel. Fertilized embryos were transferred to recipient foster dams and subsequently analysed for F1 effects postnatally. (**f**) Growth curve of F1 offspring from IVF transferred to CD1 surrogate dams shows mean birth- and postnatal- weight is reduced when fertilized by nABX donor sperm. The prevalence of the severe growth restriction phenotype, characterized by extremely low body weight (Z-score < −3) by P15, can be observed in nABX outliers. Shown right is all data points from IVF embryos at P15. *p-value* by nested two-tailed t-test. (**g**) Growth curves of independent IVF offspring transferred to C57BL/6J surrogate dams. *p-value* by nested two-tailed t-test. Note CD1 mothers foster larger pups than C57BL/6J mothers, despite offspring being genetically-identical pure C57BL/6J (compare scale in Fig B & C), owing to higher quality in utero/maternal care from CD1. Nevertheless, we still observed a recapitulation of the F1 body weight phenotypes as observed in natural C57BL/6J matings in both *in utero* backgrounds.

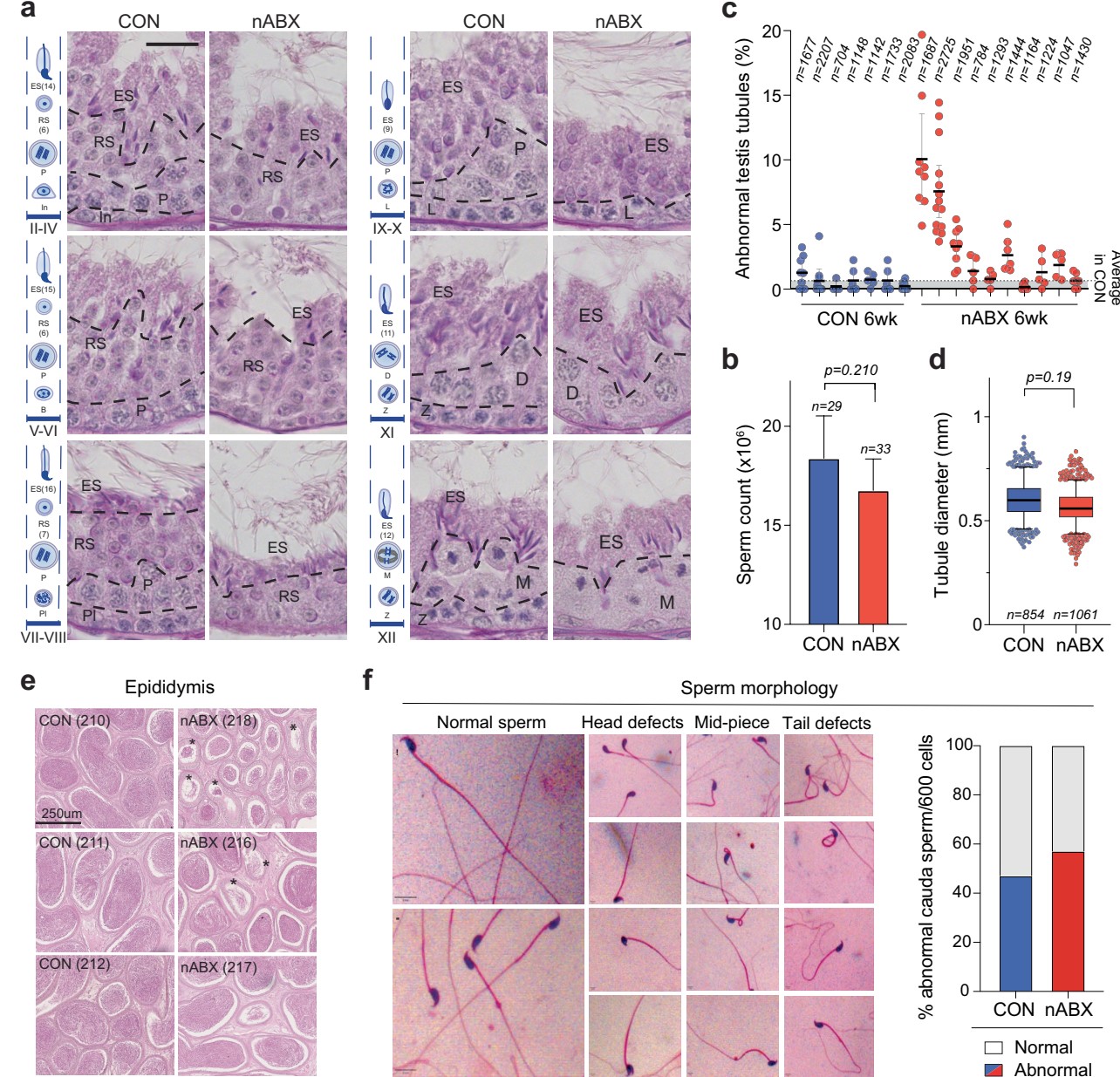

**Extended Data Fig. 7 | Changes in testes features in response to gut microbiota perturbation.** (**a**) Testes seminiferous tubule sections from control males or after 6wk nABX, stained with hematoxylin and eosin (H&E). Shown are corresponding sections of CON and nABX testis from multiple different tubule stages (labelled in roman numerals), with the expected distribution of germ cell types for each stage shown (left). nABX testis routinely exhibited abnormalities and loss of entire cell-type compartments, potentially reflective of an impacted spermatogonial stem cell pool. Round spermatids (RS), Elongated spermatids (ES), Pachytene (P), Zygotene (Z), Preleptotene (Pl), Leptotene (L), Diplotene (D), Meiotic division (M). Scale bars: 100μm. (**b**) Sperm count from indpendent control and 6wk nABX-treated males. *p-value* by unpaired two-tailed t-test (CON n = 29, nABX n = 33). Bar indicates mean with 95% C.I. (**c**) Quantification of abnormal testes stratified by effect on individual testis (CON n = 7 testis, from N = 4 males; nABX n = 10, N = 5 males). All but one nABX testis had a higher mean rate of abnormalities relative

to control average. Bar indicate mean with 95% C.I. (**d**) Quantification of total tubule diameter in control and nABX-treated (6wk) testis. *p-value* by nested unpaired two-tailed t-test (CON n = 854 tubules, nested into N = 4 males; nABX n = 1061, N = 4 males). Bar represents median and whiskers indicate 5-95th percentile. (**e**) Representative images of epididymis sections from independent nABX-treated males. Star indicates abnormality. (**f**) Assessment of sperm morphology. Shown are representative examples of normal sperm and those with abnormal head-piece, mid-piece and tail defects. Indicated right is quantification of overall level of abnormal sperm in control and nABX-treated males. All values are within the normal range, consistent with normal fecundity but altered molecular payload. Each male sperm sample was prepared in 3 slide smears at different concentrations and ~100 sperm cells were counted per slide (N = 2 males; n = 300 sperm cells counted per male). Out of 600 sperm cells counted per group, 279 from the CON group and 339 from the nABX group showed one or more morphological defects.

**Extended Data Fig. 8 | Molecular changes in testes as a response to gut microbiota perturbation.** (**a**) Representative examples of metabolites that exhibit a change in abundance specifically in testis of mice with gut dysbiosis induced by nABX. Five independently treated testis samples were analysed using untargeted metabolomics (CON = 5; nABX = 5). Shown is the effect after 6wk nABX-induced dysbiosis and the dynamic abundance of testis metabolites during microbiota recovery. The specific metabolite is shown with its class in brackets. Bar represents median and whiskers indicates data range. (**b**) Pathway analysis of differentially abundant metabolites in testes of nABX-exposed males. *p-value* adjusted for multiple testing. (**c**) Principal component analysis

(PCA) of changes on the global transcriptomes of testes from independent control (blue) or nABX-exposed males at 6wk. (**d**) PCA analysis of metabolite composition in the testes from control or dysbiotic males treated with nABX for 6wk. (**e**) Integrated joint analysis of pathway enrichments arising from both transcriptome and metabolomic changes in testes of nABX-treated males. (**f**) Enriched KEGG pathways using gene-set enrichment analysis of transcriptome in testes of nABX males relative to controls. (**g**) Collective changes in marker genes for specific cell types (Green et al, 2018) in testes after nABX. Germ cell markers are globally downregulated whilst somatic cell markers, such as for sertoli cells are increased.

## Testis cell-type clusters

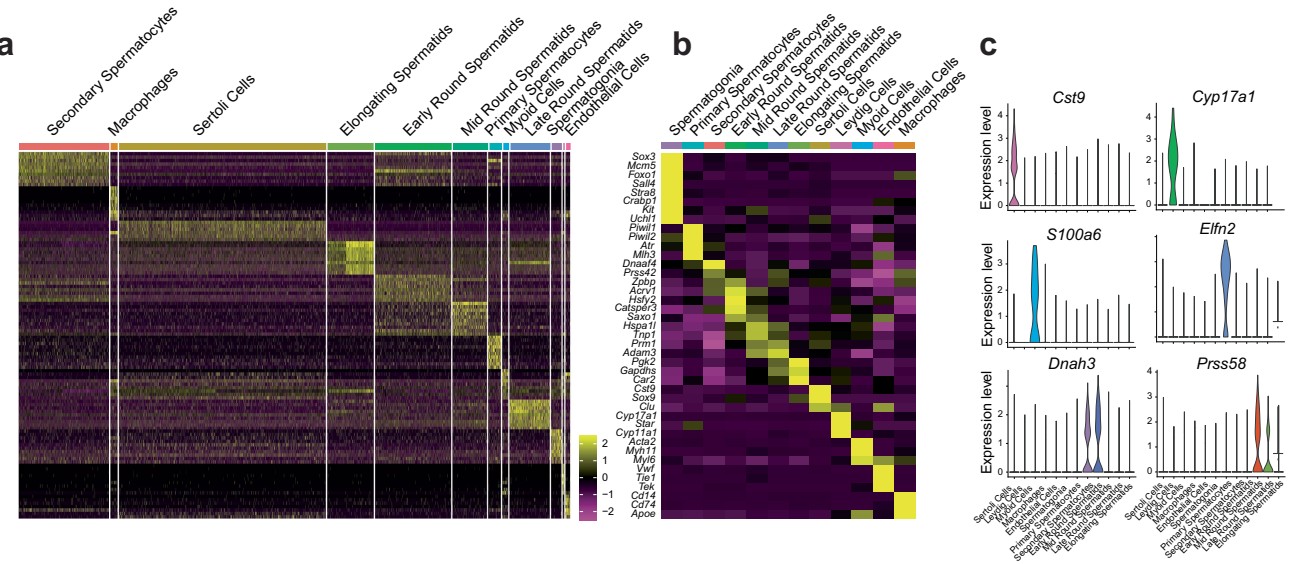

## Testes cell proportions and differential expressed genes (by cell type) in nABX males

## *In situ* sequencing of *Leptin* mRNA

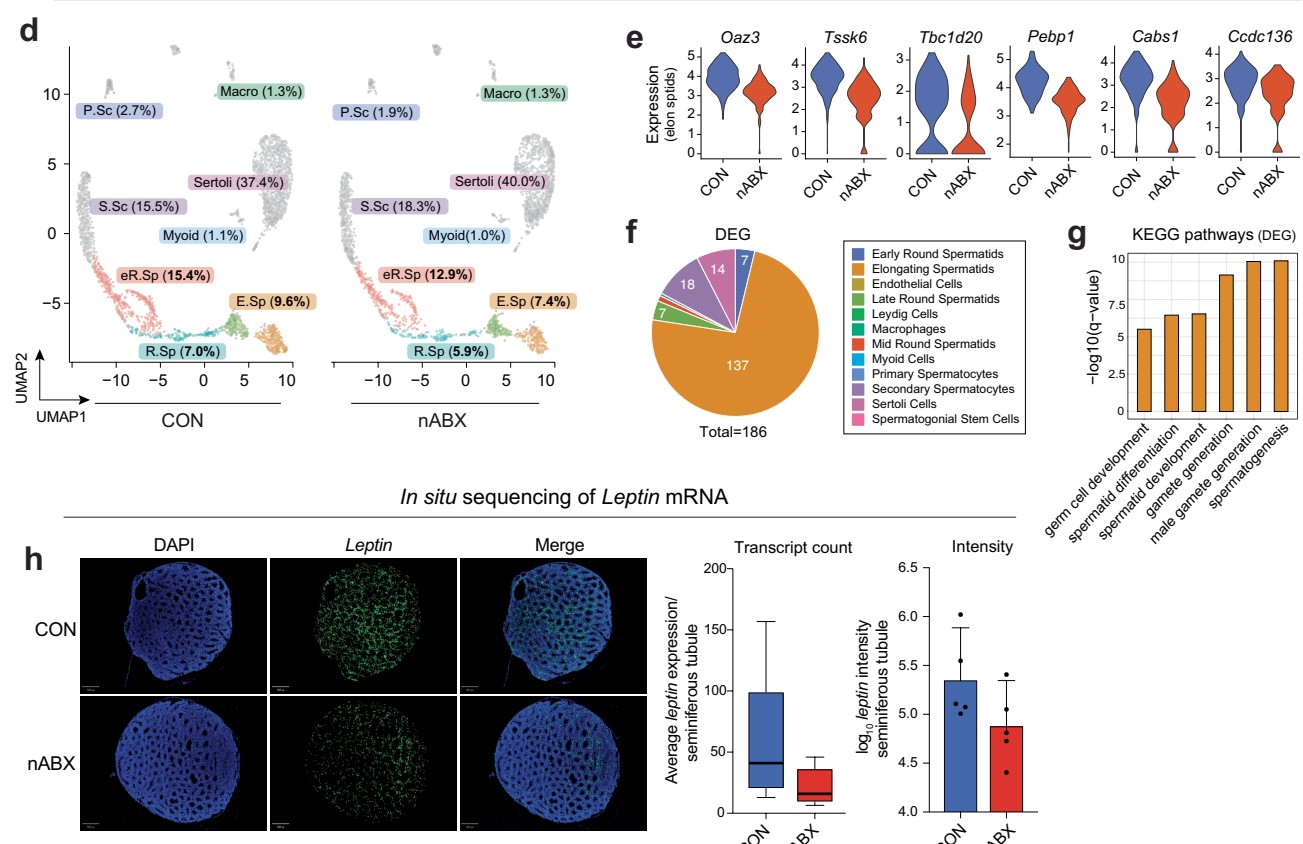

**Extended Data Fig. 9 | Single-cell RNA-seq and hybISS of testis in dysbiotic males.** (**a & b**) Heatmap showing marker gene expression for cell identity clusters identified by unbiased analysis in mouse testis from single-cell (sc)RNA-seq analysis of ~3,000 cells, pooled from n = 4 independent males for each condition. (**c**) Violin plots showing the expression profiles of selected marker genes within specific cell-type clusters in mouse testis. (**d**) UMAP projection of single-cell RNA-seq data indicating cell-type specific proportions in testes from control or nABX-treated males. Post-meiotic cell types (highlighted) are preferentially depleted in nABX males, whereas somatic sertoli cells are proportionally enriched as a consequence. Data in each condition from n = 4 independent males pooled. (**e**) Violin plots showing representative examples of single-cell expression profiles of DEGs in elongating spermatids. (**f**) Differentially expressed genes (DEG) within each testicular cell type in nABX-exposed males. (**g**). KEGG pathway analysis of differentially expressed genes from elongating spermatids. *q-value* corrected for multiple testing. (**h**) In situ sequencing in single cells confirming reduction of *leptin* expression in the seminiferous tubules of nABX-exposed males (CON = 5; nABX=5, testes). Middle panel: bar represents the median and whiskers indicate the 5-95th percentile. Right panel: bar indicate mean with 95% C.I.

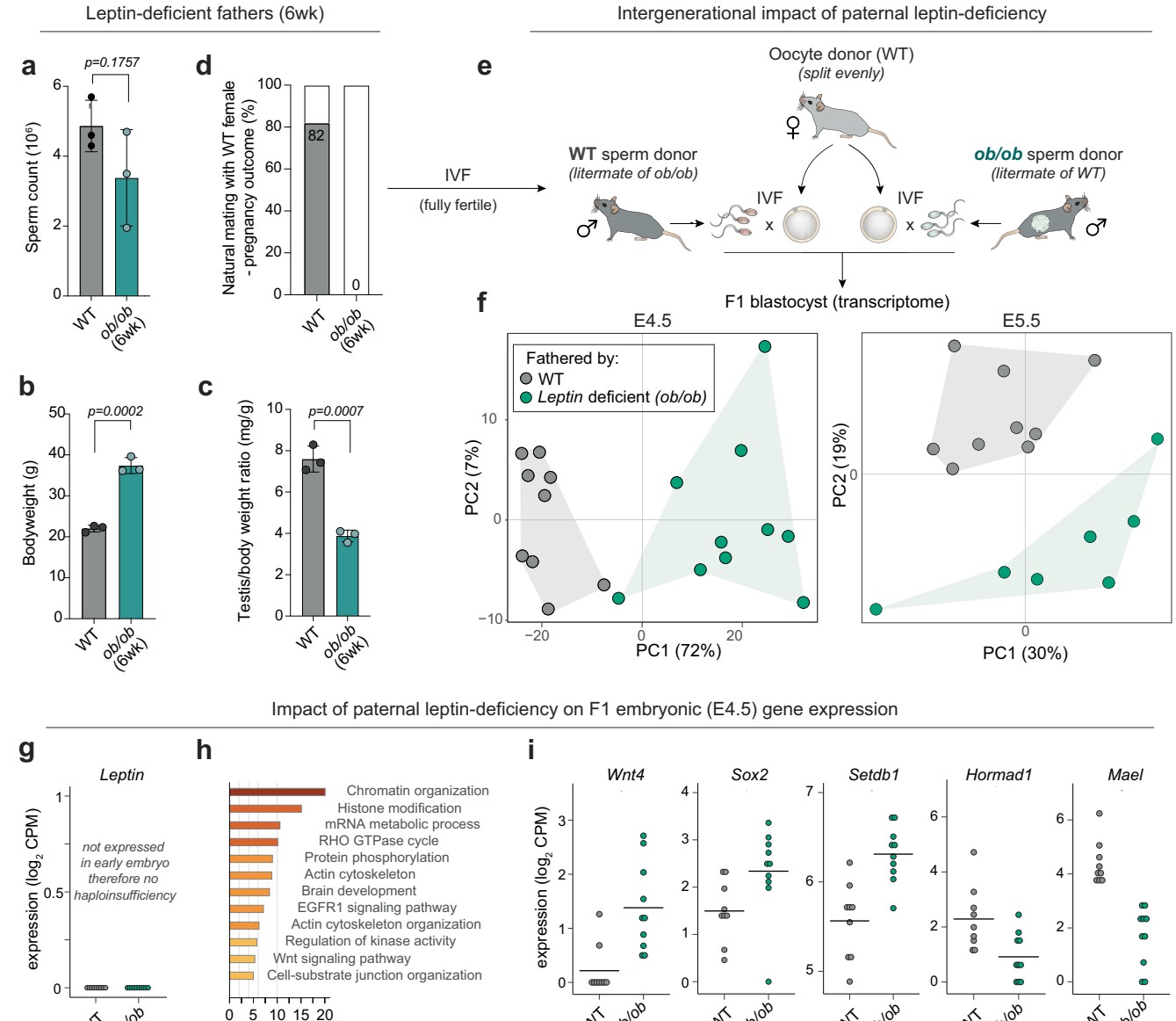

**Extended Data Fig. 10 | Paternal leptin deficiency induces testicular phenotypes and major intergenerational expression changes.** (**a**) Bar chart showing sperm count in young (6wk old) *ob/ob* males that lack leptin signalling for an equivalent time period as 6wk-treated nABX males, where leptin is also impaired (WT = 3; ob/ob = 3, sperm samples). Bars show mean with S.D and significance by two-tailed t-test. (**b-c**) Bar chart showing (**b**) body weight and (**c**) testis/body weight ratio of *ob/ob* males at 6-week-old. Note that despite increased body weight linked with elevated satiety, testes weight is still reduced in absolute terms in *ob/ob* males (see Fig. 3i) and as a ratio to body weight, indicating a reproductive response. (**d**) Percentage natural matings that led to a successful pregnancy from control or *ob/ob* males. The lack of *ob/ob* pregnancies could indicate infertility due to direct sperm defects, or to indirect effects, such as impaired behaviour or physical capacity. These can be distinguished by IVF (see panel E-F), which indicates failure of *ob/ob* natural mating at this age is due to indirect effects, as IVF from *ob/ob* sperm was equally efficient as from WT. (**e**) Schematic showing the experimental design to test potential F1 impacts of dysregulated paternal leptin signalling. *wt/ob* mice

were intercrossed to generate *ob/ob* and wt/wt males and sperm from such littermates was then isolated and used to fertilise WT oocytes, with no difference observed in the fertilization rate or development to blastocyst, indicating viable sperm from *ob/ob* and wt/wt males. (**f**) Principal component analysis of single-embryo transcripomes from E4.5 (left) and E5.5 (right) blastocysts. Each embryo was generated by parallel fertilizations using control or leptin-deficient sperm. Triplicate independent males were used per condition, across duplicate independent IVF experiments. A reproducible shift in gene expression patterns and thus phenotype, is observed depending on the father's leptin levels. (**g**) Expression of *Leptin* in early embryos is undetectable. This argues against a potential haplo-insufficient effect on gene expression in *wt/ob* embryos derived from leptin-deficient (*ob/ob*) sires. Instead it points to a paternally inherited defect in sperm. (**h**) Gene ontology pathways of differentially expressed genes in embryos from leptin-deficient fathers. The test statistic reported as *p-value*. (**i**) Representative differentially expressed genes in blastocysts, depending on the paternal leptin status (WT = 9; *ob/ob* = 10, blastocysts). Each data point indicates log expression in an individual embryo at E4.5.

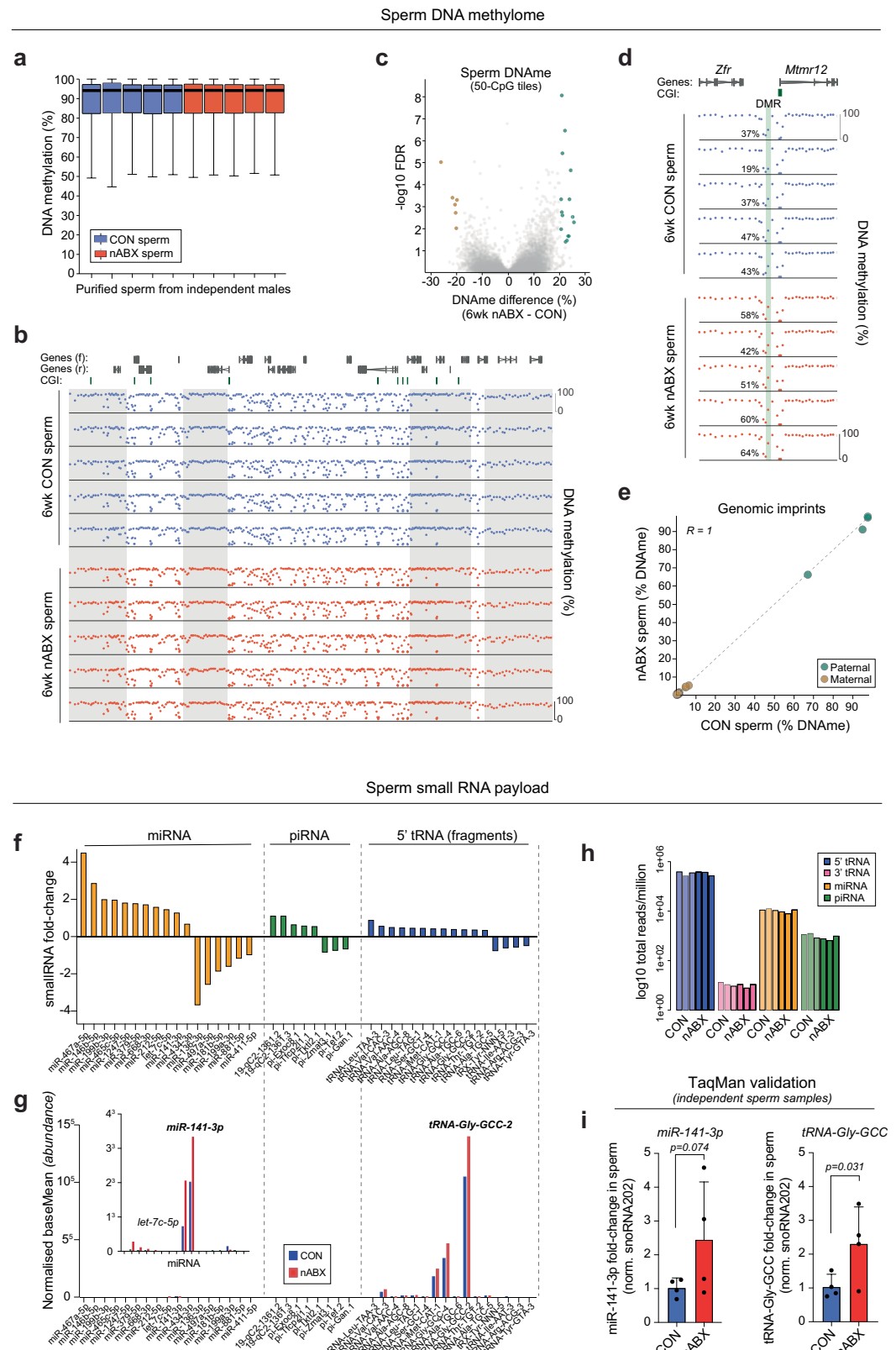

**Extended Data Fig. 11** | See next page for caption.

**Extended Data Fig. 11 | DNA methylation and small RNA profiles in purified sperm from dysbiotic males.** (**a**) Boxplot showing the global level of DNA methylation in sperm from multiple independent males, (n = 5 per condition) ascertained by whole-genome bisulfite sequencing (WGBS), quantified using 50 CpG tiles. Bar represents median and whiskers indicate median plus/minus the IQR multipled by 2. (**b**) Example genome track showing highly reproducible DNA methylation patterns in sperm irrespective of nABX treatment. Each data point represents the percentage methylation across a 50 CpG tile. Grey regions demarcate hypermethylated genomic zones. (**c**) Volcano plot highlighting significant differential methylated regions (DMR). Whilst 21 DMR loci were identified, the effect size was modest and they primarily overlapped CpG shores; loci more prone to natural biological or technical variation. (**d**) Genome track showing DNA methylation changes at a representative DMR (in green) using read-depth sensitive logistic regression ($p < 0.05$ & >20% abs. change). Percentages indicate DNA methylation level across the DMR in each sperm sample. Data points indicate methylation at each sliding 50 CpG tile. (**e**) Scatter plot showing DNA methylation at genomic imprinted regions in control and nABX-treated sperm. Paternal and maternal imprints are indicated. (**f**) Bar chart showing small RNAs that exhibit a foldchange in expression in purified sperm from nABX males relative to control sperm. (**g**) Absolute abundance of each indicated small RNA and class in control or nABX sperm. tRNA-Gly-GCC-2 (and tRNA-Gly-GCC-4) represent a major constituent of total RNA abundance in mature sperm and therefore relatively small foldchange differences in their expression constitutes a major change in absolute copy number in sperm heads. Each sample represents a total of n = 9 individual male samples pooled in to three replicates. Inset is the overall abundance (proportion of reads) of each small RNA class (**h**) Zoomed in representation of (B) showing miR-141 and Let-7 are highly abundant among microRNA and piRNA classes. (**i**) TaqMan qPCR quantification of top hits (*miR-141-3p and tRNA-Gly-GCC*) from small RNA-seq datasets using purified sperm from nABX-exposed males. Bars show mean with S.D and significance by one-tailed t-test. Sperm samples obtained in independent treatments from samples used for small RNA-seq (CON n = 4; nABX n = 4).

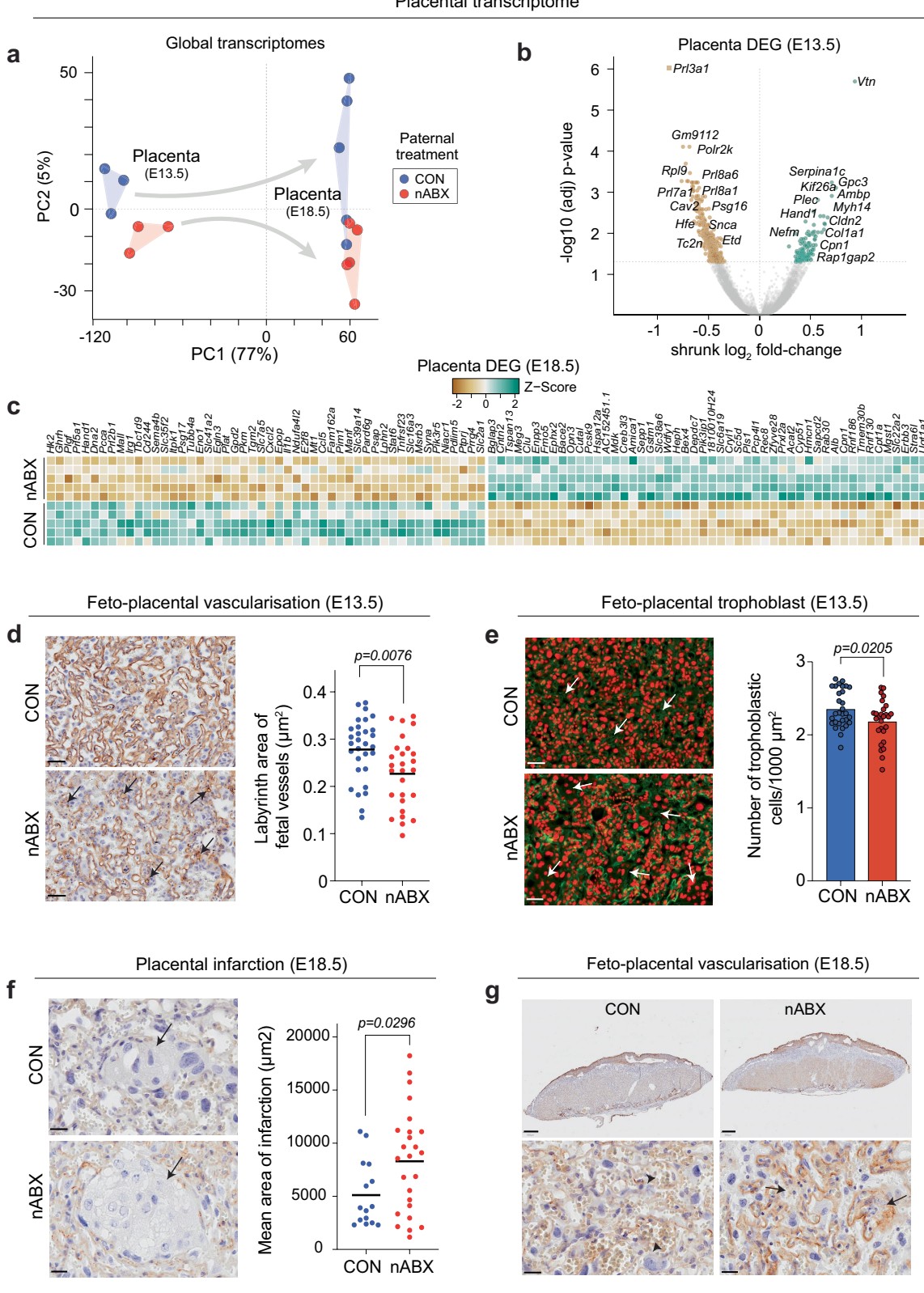

Placental transcriptome

**a** Global transcriptomes

**b** Placenta DEG (E13.5)

**c** Placenta DEG (E18.5)

**d** Feto-placental vascularisation (E13.5)

**e** Feto-placental trophoblast (E13.5)

**f** Placental infarction (E18.5)

**g** Feto-placental vascularisation (E18.5)

**Extended Data Fig. 12** | See next page for caption.

**Extended Data Fig. 12 | Paternal nABX exposure induces molecular and physiological responses in forthcoming placenta.** (**a**) Principal component analysis of placental transcriptomes from independent litters sired by control or nABX-treated fathers. (**b**) Volcano plot showing differentially expressed genes (DEG; highlighted ochre for downregulated, green for upregulated) in nascent (E13.5) placenta derived from control of nABX-exposed sires. *p-value* adjusted for multiple testing. (**c**) Heatmap showing consistent changes in gene expression between independent placenta from different litters, depending on the microbial status of the father. (**d**) E13.5 placental staining with IB4, a marker for endothelial cells to reveal fetoplacental vascularization within the labyrinth. nABX-derived placental tissue displayed abnormal vasculization (arrows) and reduced total vascularization, quantified right. (**e**) Immunofluorescent staining to determine the prevalence of trophoblast cells (arrows indicate absence) in E13.5 placenta, with quantification shown right. (**f**) Increased number and size of placental infarctions (lesions, arrow) in E18.5 placenta derived from CON or nABX males. (**g**) E18.5 placental staining with the IB4 endothelial cell marker demonstrating impaired fetoplacental vascularization within the labyrinth zone (arrowheads indicate normal blood vessels, arrows indicate abnormal). Multiple placenta from three independent litters were examined for each group, with each sired by independent fathers (CON = 3; nABX = 3). Data in panel d-f were each collected from placenta samples (CON n = 30 subregions of labyrinth, N = 5 placenta (3 litters); nABX n = 27 subregions of labyrinth, N = 5 placenta (3 litters)) and from each placenta 5 tissue sections covering different subregions of the labyrinth were analysed. It should be noted that these samples are independent litters from Fig. 4. Bar indicates mean with 95% C.I. Data analysis using unpaired student's two-tailed t-test. Scale bars: 50 µm (d,e), 20 µm (f) and 500 µm, 20 µm (g). Placental vascular network and trophoblastic cell count were analysed using QUPath-V0.2.3.

# nature research

# Reporting Summary

Nature Research wishes to improve the reproducibility of the work that we publish. This form provides structure for consistency and transparency in reporting. For further information on Nature Research policies, see our Editorial Policies and the Editorial Policy Checklist.

## Statistics

For all statistical analyses, confirm that the following items are present in the figure legend, table legend, main text, or Methods section.

| n/a | Confirmed | |
|---|---|---|
| ☐ | ☒ | The exact sample size (*n*) for each experimental group/condition, given as a discrete number and unit of measurement |
| ☐ | ☒ | A statement on whether measurements were taken from distinct samples or whether the same sample was measured repeatedly |
| ☐ | ☒ | The statistical test(s) used AND whether they are one- or two-sided *Only common tests should be described solely by name; describe more complex techniques in the Methods section.* |
| ☒ | ☐ | A description of all covariates tested |
| ☐ | ☒ | A description of any assumptions or corrections, such as tests of normality and adjustment for multiple comparisons |
| ☐ | ☒ | A full description of the statistical parameters including central tendency (e.g. means) or other basic estimates (e.g. regression coefficient) AND variation (e.g. standard deviation) or associated estimates of uncertainty (e.g. confidence intervals) |
| ☐ | ☒ | For null hypothesis testing, the test statistic (e.g. *F*, *t*, *r*) with confidence intervals, effect sizes, degrees of freedom and *P* value noted *Give P values as exact values whenever suitable.* |
| ☒ | ☐ | For Bayesian analysis, information on the choice of priors and Markov chain Monte Carlo settings |
| ☐ | ☒ | For hierarchical and complex designs, identification of the appropriate level for tests and full reporting of outcomes |
| ☐ | ☒ | Estimates of effect sizes (e.g. Cohen's *d*, Pearson's *r*), indicating how they were calculated |

*Our web collection on statistics for biologists contains articles on many of the points above.*

## Software and code

Policy information about availability of computer code

| Data collection | No custom software was used in this study. Sequencing data was collected using Illumina MiSeq, NextSeq500 or HiSeq 4000 platforms, while metabolic data was acquired using LC-MS platform. |
|---|---|
| Data analysis | Data analysis was performed using Graphpad Prism version 8.4.3 graphical software, Galaxy maintained by the EMBL Genome Biology Computational Support, R statistical software (v3.6.2) using Bioconductor packages, and with Seqmonk (v1.45.4) mapped sequence data anaylser.<br><br>Phenotypes: Significant differences in F1 offspring bodyweight between groups were determined using nested (hierarchical) t-test analysis. This compares the means of two unmatched groups (all F1 offspring from control or dysbiotic father), where there is a nested factor within those treatment groups (shared father amongst each litter). Statistical power (n number) is therefore limited to the number of litters rather than the number of offspring, which prevents spurious significance and provides robust confidence in differences. Testes to bodyweight ratio, fetoplacental ratio, and labyrinth zone were analysed by two-tailed unpaired t-test. Odds ratios (ORs) and 95% confidence intervals (CIs) were computed using with Baptista-Pike method and the statistical significance of the ORs were determined using chi-squared test. Kaplan-Meier method was applied to generate survival analysis curves compared by the log-rank (Mantel-Cox) test. ELISA calibration curves were interpolated with a Hyperbola (X is concentration) nonlinear regression model fit (R2> 0.99 was acceptable curve fit).<br><br>RNAseq: Raw reads were quality trimmed using TrimGalore (0.4.3.1, -phred33 --quality 20 --stringency 1 -e 0.1 --length 20). These were mapped to the mouse mm10 (GRCm38) genome assembly using RNA Star (2.5.2b-0, default parameters except for --outFilterMultimapNmax 1000) and reads with a MAPQ score <20 were discarded to ensure only unique-mapping high quality alignments were used for analysis of gene expression. The data was quantified using the RNA-seq quantification pipeline for directional libraries in seqmonk software to generate log2 reads per million (RPM) or gene-length-adjusted (RPKM) gene expression values. Differentially expressed genes (DEGs) were determined using the DESeq2 package (version, 1.24.0), inputting raw mapping counts, and applying a multiple-testing adjusted p-value (FDR) <0.05 significance threshold. An additional fold-change (FC) filter of >2 was applied to generate final DEGs. Principle component analysis (PCA) of transcriptomes were computed in seqmonk and R statistical software using all expressed genes as input. These were defined as having an |

April 2020

RPKM >0.1in at least 2 replicates across all assayed samples. Differentially expressed genes, or gene lists of interest, were inputed into the STRING v11.0 database, and extracting enrichment analysis related to Reactome and KEGG pathways, filtering by FDR rank.

smallRNAseq: Adapters were removed using fastx-clipper, fastq files were converted to fasta using a custom perl script and reads of 18-33 nucleotides in length were retained using a custom perl script. Reads were aligned to the mouse genome version mm10 using bowtie (v1.3.0), reporting only the best alignment and requiring 0 mismatches (parameters -v 0 -k 1 -- best --sam).  Alignment sam files were converted to bam files using samtools version 1.9 and bam files were converted to bed files using bedtools version 2.  In order to quantitate miRNAs intersectBed -c was used to count the number of reads overlapping the positions of the known Mus musculus miRNAs (from miRbase www.miRBase.org). tRNAs were obtained by using intersectBed -c to count the number of reads overlapping a bed file documenting predicted mouse tRNA coordinates downloaded from the tRNA scan database (http://gtrnadb.ucsc.edu/genomes/eukaryota/Mmusc10/). piRNA coordinates were taken from Li et al., 2013 (pmid 23523368) and converted to mm10 using liftover.  piRNAs were quantitated by selecting RNAs between 26 and 32 nucleotides long with a U as the first nucleotide and intersecting these RNAs with the piRNA coordinates using intersectBed-c. To investigate significant differences data was processed using DESeq2 and the negative binomial test used to identify significant differences after Benjamani Hochberg multiple test correction.

Single-cell RNA-seq:  Raw reads were aligned and mapped using the count module in 10x Genomics Cellranger 6.1.2 73 to the mm10 transcriptome assembly (2020-A) with default parameters. Quality control. All subsequent steps were performed in R (version 4.1.2) using the Seurat package 74. First, cells were filtered based on 3 parameters – number of UMIs (nCount_RNA; 1,000:5,000), number of unique genes detected (nFeature_RNA; 500:5,000) and mitochondrial rate (percent.mito) and ribosomal rate (percent.ribo; 0:20) – as described below. The nABX and CON samples were then clustered separately using the default Seurat clustering approach. In both conditions, clusters having no uniquely expressed genes (using the FindMarkers function with logfc.threshold=0.5 and min.pct=0.5) were discarded. Integration. The nABX and CON samples were then integrated using the Canonical Correlation Analysis (CCA) approach at default parameter settings as recommended by the Seurat package. Cell type annotation: The integrated dataset was then clustered and annotated using uniquely expressed marker genes. As above, clusters showing no uniquely expressed genes were ignored. To define more fine-grained annotations, the somatic and germ cells were split into different Seurat objects and clustered separately.Cell-type-specific differential expression: For each cell type identified in the dataset, the FindMarkers function was used to identify differentially expressed genes between nABX and CON cells using logfc.threshold=0.25 and p_val_adj<0.1.

WG Bisulfite-seq: Raw fastq sequences were quality- and adapter- trimmed using TrimGalore (0.4.3.1) and reads aligned to mm10 using Bismark (0.20.0), discarding the first 8 bp from the 5' end and the last 2 bp from the 3' of a single-end reads. Cytosine methylation status was extracted from mapped reads using the Bismark methylation extractor tool. Genome-wide methylation calls were analysed using Seqmonk software (1.44.0) with five biological independent replicate datasets for each condition. To identify differentially methylated regions (DMR) the genome was first binned into sliding tiles containing 50 consecutive CpGs and their methylation status determined using the DNA methylation pipeline. DMRs were identified by running read-depth sensitive logistic regression (p(adj)<0.05), with minimum of 10 reads, and applying a threshold for an absolute change in DNA methylation of 20%. The methylation level at specific genomic features (e.g. imprints) was calculated using the DNA methylation pipeline in Seqmonk over target features.

gDNA-seq: Alignment reads were trimmed using Trim galore (0.6.3 ), then the first ten 5' bases of both reads removed with Cutadapt (2.3). Reads were aligned to Mus Musculus refence genome (mm10) using bwa mem (BWA-0.7.17) before filtering with samtools (1.10) view with the flags '-h -F 256 -f 2 -q 30' and deduplicated with Picard toolkit (2.9.0) MarkDuplicates. SNPs and small INDELs were called using GATK (4.1.6.0) HaplotypeCaller. Variants were filtered to remove those with a PHRED-called site quality (QUAL)< 30, an allele frequency < 0.2 or low site coverage (sliding scale) in more than one individual. Variant functional region was annotated using ANNOVAR (2020Jne07) annotate_variation.pl. Structural variants were called with Delly2 call (v0.8.7).

16S rRNA-seq: Raw 16S rRNA reads were trimmed, denoised and filtered to remove chimeric PCR artefacts using DADA2. The resulting Amplicon Sequence Variants (ASVs) were then clustered into Operational Taxonomic Units (OTUs) at 98% sequence similarity using an open-reference approach: reads were first mapped to a pre-clustered reference set of full-length 16S rRNA sequences at 98% similarity using MAPseq. Reads that did not confidently map were aligned to bacterial and archaeal secondary structure-aware SSU rRNA models using Infernal and clustered into OTUs with 98% average linkage using hpc-clust. The resulting OTU count tables were noise filtered by asserting that samples retained at least 1,000 reads and taxa were prevalent in at least 2 samples; these filters removed 58% spurious OTUs, but only 0.09% of total reads from the dataset. Local sample diversities were calculated as OTU richness, exponential Shannon entropy and inverse Simpson index (corresponding to Hill diversities of order 0, 1 and 2) as average values of 100 rarefaction iterations to 5,000 reads per sample. Between-sample community diversity was calculated as Bray-Curtis dissimilarity. Trends in community composition were quantified using ordination methods (Principal Coordinate Analysis, distance-based Redundancy Analysis) and tested using permutational multivariate analysis of variance (PERMANOVA, as implemented in the R package vegan).

Metabolomics: All statistical analyses and plotting were performed in R version 3.6.2. To exclude bad sample injection from downstream analyses, the sum of all extracted metabolite features (TIC) was compared between samples (mean = 1.600e+10, range = [1.419e+10; 1.808e+10]) and samples were excluded, if their TIC was not within three standard deviations from the mean value (0 excluded samples). Missing data were imputed to a fixed threshold, set at 5000 counts. Exact duplicated features or feature falling within (i) a 0.002 amu (absolute threshold) or 20 ppm (relative threshold) window and (ii) a 0.15 mins (absolute threshold) or 2% (relative threshold) Retention Time (RT) window, were considered to be split peaks, and therefore collapsed together. Correlation between testis weight and animal body weight was computed, to verify that there was no significant correlation between the two values (cor = 0.1870, p-value = 0.2477). Therefore, testis weight z-scores were used to normalize AUC intensity values, to consider variation in signal intensity derived from testis size. Features (i) at zero variance; (ii) being singletons; (iii) present in less than 75% of the samples for each class were removed. Finally, feature tables derived from positive and negative mode were collapsed together after checking for exact duplicated features or feature falling in a 0.002 Da or 20ppm window existed; if so, the feature with higher average intensity was retained and both annotations were retained. Annotation was retrieved from the Human Metabolome Database (HMDB, https://hmdb.ca/), by searching for metabolites with the exact same mass or falling in a 0.002 amu or 20ppm window from a metabolite's monoisotopic mass. When present, multiple annotation were retrieved. Only features being annotated were retained for the downstream analysis. Moreover, for each feature, metabolite class and superclass were retrieved, when present, from the HMDB. Principal Component Analysis (PCA) was computed both for the complete dataset and after stratifying it by sampling week for all metabolic features and annotated features only. Statistical significance of the feature intensity differences was assessed using a two-sided t-test (stats::t.test function in R) of log scaled data, and P values were FDR-corrected for multiple hypotheses testing using the Benjamini–Hochberg procedure (stats::p.adjust function in R with BH parameter). Mass, retention time, HMDB annotation, Class, Superclass and composite spectrum were retrieved for all features showing significantly different intensity between the treated and the control group and an absolute fold change greater than 2. Metabolite class and superclass enrichment analysis between the treated and control group was

calculated using a Fisher's exact test. All P values were FDR-corrected for multiple hypotheses testing using the Benjamini–Hochberg procedure (stats::p.adjust function in R with BH parameter).

For manuscripts utilizing custom algorithms or software that are central to the research but not yet described in published literature, software must be made available to editors and reviewers. We strongly encourage code deposition in a community repository (e.g. GitHub). See the Nature Research guidelines for submitting code & software for further information.

## Data

Policy information about availability of data

All manuscripts must include a data availability statement. This statement should provide the following information, where applicable:
- Accession codes, unique identifiers, or web links for publicly available datasets
- A list of figures that have associated raw data
- A description of any restrictions on data availability

The authors declare that data supporting the findings of this study are available and have been deposited in ArrayExpress (RNAseq (E-MTAB-10034), Bisulfite-seq (E-MTAB-10033), gDNA-seq (E-MTAB-10273)). 16S rRNA-seq datasets are deposited in ENA (PRJEB43500); metabolomics deposited in MetaboLights (MTBLS1629). The datasets underlying figures are available as source data

# Field-specific reporting

Please select the one below that is the best fit for your research. If you are not sure, read the appropriate sections before making your selection.

☒ Life sciences ☐ Behavioural & social sciences ☐ Ecological, evolutionary & environmental sciences

For a reference copy of the document with all sections, see nature.com/documents/nr-reporting-summary-flat.pdf

# Life sciences study design

All studies must disclose on these points even when the disclosure is negative.

| | |
|---|---|
| Sample size | Information on sample size is provided within each figure legend<br>Animal phenotypic studies: both the power analysis and resource equation method were applied to determine the minimum number of mice required to enable detection and validation of a probabilistic intergenerational effect of paternal dysbiosis on offspring phenotypes.<br>RNAseq : transcriptome profile of testis was performed in 5 biological replicates (5 sires/group), while for F1 offspring or fetal transcriptome profile samples were collected from three independent mating (3 litter/group), and performed in at least 5 biological replicates samples (5 independent offspring tissues/treatment group). Sample size was determined based on prior research and to exceed field standards for the technique.<br>smallRNAseq: small RNAs profile of mouse sperm was performed in 9 biological replicates (9 males/ 3 pooled samples/treatmemnt group). Sample size was determined based on field standards for the technique in order to ensure enough power.<br>Bisulfite-seq: DNA methylation profiling of mouse sperm was performed in 5 biological replicates (5 males/treatment group). gDNA-seq: F1 offspring liver samples were profiled for genome changes in 3 biological replicates (3 offspring/phenotype). Sample size was determined based on prior research experience and to exceed field standards for the technique.<br>16S rRNA-seq: microbiome profile of sires and dams were performed at minimum in 10 biological replicates (10 sires/group & 10 dams/group), while F1 offspring microbiome profile was performed at least from 10 independent mating (offspring born from > 10 litters/dysbiotic condition/group). Sample size was determined based on prior research experience and to exceed field standards for the technique.<br>Metabolomics: metabolomics profile of testis was performed in 5 biological replicates (5 sires/group). Placenta: analysis of placental defects was performed collectively in >20 litters per paternal condition, split between time of harvest (E13.5, E18.5) and analysis technique (histology, RNA-seq etc). For each readout at least 3 independent litters/fathers were analysed. Sample size was determined based on prior research experience and to exceed field standards for the technique. |
| Data exclusions | For offspring growth phenotype analysis, outliers due to extreme litter size effects were excluded. Specifically, litters outside 2 standard deviation of the average litter size (6±2 pups/litter) were excluded in the study (i.e. litters <4 pups or >8 pups). These exclusions were comparable between control and treatment groups (~19% of litters in each). No data was excluded from RNAseq, smallRNA-seq, 16S rRNA-seq, Bisulfite-seq and metabolomics data. |
| Replication | The probabilistic impact of paternal gut dysbiosis on offspring phenotype were replicated and validated across multiple litters (n>80) using similar or different dysbiotic agents (i.e. non-absorbable antibiotics, absorbable antibiotics and osmatic laxatives), across multiple independent experiments (batches of 5 control or treated sires in parallel), and over time. Reproducibility between independent RNAseq, smallRNA-seq, 16S rRNA-seq, Bisulfite-seq and metabolomics samples were assessed on binned and library-normalised files using multiple clustering approaches including PCA, correlation assessment, and unsupervised hierarchal clustering in R and Seqmonk software, with good reproducibility observed. All replicates were biologically independent and collected in parallel to minimise batch effects. All replicates were succesful except a seque cing batch of placenta that failed standard QC pipeline and was repeated. Number of independent biological replicates is listed in each lgend and under 'sample size' above. |

| | Randomization | Throughout our experiment, before initiating the antibiotics/PEG treatment, the inbred male mice were divided into separate cages (one mouse/cage) and randomly assigned into Control group or Treated group. While randomization was not required for sample collection from offspring since they were defined based on their forefather's dysbiotic status (Control vs. Treated). |
| | Blinding | Matings were setup using a blinded code system, and F1 phenotypes were recorded by individuals and/or husbandry staff without knowledge of the paternal condition. Blinding was not relevant for RNAseq, smallRNA-seq, 16S rRNA-seq, Bisulfite-seq and metabolomics samples analysis, since the study was based on objective quantitative analysis methods. |

# Reporting for specific materials, systems and methods

We require information from authors about some types of materials, experimental systems and methods used in many studies. Here, indicate whether each material, system or method listed is relevant to your study. If you are not sure if a list item applies to your research, read the appropriate section before selecting a response.

### Materials & experimental systems

| n/a | Involved in the study |
|---|---|
| ☐ | ☒ Antibodies |
| ☒ | ☐ Eukaryotic cell lines |
| ☒ | ☐ Palaeontology and archaeology |
| ☐ | ☒ Animals and other organisms |
| ☒ | ☐ Human research participants |
| ☒ | ☐ Clinical data |
| ☒ | ☐ Dual use research of concern |

### Methods

| n/a | Involved in the study |
|---|---|
| ☒ | ☐ ChIP-seq |
| ☒ | ☐ Flow cytometry |
| ☒ | ☐ MRI-based neuroimaging |

## Antibodies

| | Antibodies used | Mouse Leptin ELISA kit MOB00 (R&D Systems)<br>Mouse PlGF-2 ELISA Kit MP200 (R&D Systems)<br>Mouse VEGFR1/Flt-1 ELISA Kit MVR100 (R&D Systems)<br>Anti-mouse VE-cadherin (Thermo Fisher Scientific; cat.14-1441-81)<br>Anti-Rat-Alexa Fluor 568 (Thermo Fisher Scientific; cat. A-11077) |
|---|---|---|
| | Validation | All used ELISA kits and antibodies are commercially available and have been validated by the manufacturer. Validations and detail product information are available on these websites:<br>Leptin ELISA kit: https://www.rndsystems.com/products/mouse-rat-leptin-quantikine-elisa-kit_mob00#product-details<br>PlGF-2 ELISA Kit: https://www.rndsystems.com/products/mouse-plgf-2-quantikine-elisa-kit_mp200<br>VEGFR1/Flt-1 ELISA Kit: https://www.rndsystems.com/products/mouse-vegfr1-flt-1-quantikine-elisa-kit_mvr100<br>Anti-mouse VE-cadherin: https://www.thermofisher.com/order/genome-database/generatePdf?productName=CD144%20(VE-cadherin)&assayType=PRANT&detailed=true&productId=14-1441-81<br>Anti-Rat-Alexa Fluor 568: https://www.thermofisher.com/antibody/product/Goat-anti-Rat-IgG-H-L-Cross-Adsorbed-Secondary-Antibody-Polyclonal/A-11077 |

## Animals and other organisms

Policy information about studies involving animals; ARRIVE guidelines recommended for reporting animal research

| | Laboratory animals | All experiments involving mice were carried out in accordance with the approved protocol and guidelines by the laboratory animal management and ethics committee of the European Molecular Biology Laboratory (EMBL) under license 20190708_JH and the Italian Ministry of Health under authorisation code 308/2021-PR. The inbred C57BL/6J strain was used as the main mouse model for this project, while CD-1 IGS or C57BL/6J dams were used as surrogate mothers for IVF. The age of the male and female mice used for this study ranges from 5-19 weeks old and 6-9 weeks old, respectively. Mice were housed under a 12-h light/dark cycle (from 7AM to 7PM), with controlled 50% humidity, at room temperature. Animals had ad libitum access to regular diet and water. Standard chow contained 18.5% protein, 5.3% fat, 4% fibre and other nutritional additives in pellet form, and is suitable for long-term maintenance, breeding, lactation and gestation periods (NFM18, Mucedola, Milan, Italy). |
|---|---|---|
| | Wild animals | The study did not involve wild animals |
| | Field-collected samples | The study did not involve samples collected from the field |
| | Ethics oversight | All experiments involving mice were carried out in accordance with the approved protocol and guidelines by the laboratory animal management and ethics committee of the European Molecular Biology Laboratory (EMBL) under license 20190708_JH and the Italian Ministry of Health under authorisation license 308/2021-PR. |

Note that full information on the approval of the study protocol must also be provided in the manuscript.

