## [Peer Review File · Nature]

Manuscript Title: Paternal microbiome perturbations impact offspring fitness

Reviewer Comments & Author Rebuttals

Reviewer Reports on the Initial Version:

Referees' comments:

Referee #1 (Remarks to the Author):

This paper tackles old questions in new ways, namely how does the microbiome affect host physiology and in turn how these physiological changes affect offspring phenotypes. The questions about the gut-germline connection are original, the approaches comprehensive, the analysis rigorous, and the implications pioneering. The work involves both a non-absorbable antibiotic as well as other means to induce dysbiosis, characterization of host molecular and physiological changes, phenotypic consequences among offspring, reversibility (too often overlooked), IVF tests for transmission through gametes, metabolic profiling, transcriptional profiling of testis, sperm and placenta, As I worked through the text, every time I had a question, wondering if it was an issue that had been overlooked, I found the question addressed with clear results. Among the many strengths of this study are the tests for reversibility. Such tests powerfully demonstrate that dynamic nature of the induced changes, but are often overlooked in studies such as this. Finally, the text is impeccably prepared.

With respect to the epigenetic aspects, the methylation profiling, the IVF experiment, and the intergenerational tests were carefully conducted, the results clear, and the interpretations appropriate. The only question involves why the random-bred CD-1 females were used in the IVF test (Materials and Methods, Animal Husbandry, line 233); could molecular and physiological variation among genetically heterogeneous females lead to phenotypic 'noise and perhaps even consistent phenotypic changes among offspring.

Several analytical issues need a comment somewhere. The p-values are relatively modest, often $0.05 > p < 0.01$, which is somewhat surprising given the usually robust sample sizes ($n > 100$). This raises questions about the magnitude of the phenotypic effects. These effects can be found as approximations in the figures; more quantitative measures should be provided in the text, - phenotypic differences in measured units would suffice. Also, the text (line 119) mentions Cohen's d , which is a commonly used measure of effect size, but its use here is unusual - I've never seen it used to assess correlation - and might deserve some explanation and perhaps reconsideration. That said, the analysis is rigorous and nicely presented - nothing superficial or casual-, and could serve as an example for how analyzes should be done and presented.

Also, a summary statement is made (main text, lines 72-73, and then lines 217-219) and then reiterated in the final paragraph of the main text that the underlying biology is probabilistic rather than deterministic'. I understand the idea, but not its relevance here. This needs some clarification.

On whom - which generation- was the metabolic profiling performed (lines 134-135)? This should be provided in the main text.

The font size in the figures is too small, e.g. Fig 1H is unreadable.

There were several questions that go beyond the scope of work needed in this manuscript, but

should be raised in the "Discussion" (last paragraph). The first concerns the mechanistic connection between changes in the gut with the germline – what are the molecules and pathways that transduce signals (and what role might this network serve under normal conditions). Then, do more subtle changes in microbiome content or functionality induce comparable phenotypic changes as those induced with antibiotics? Also, the text refers to changes in diet as well as antibiotics, but diet tests are not presented here (and are not needed) – how might diet and commonly used antibiotics impact these intergenerational effects. Finally, how might the seemingly general changes in the germline lead to fairly specific and reproducible phenotypic changes in the next generation.

Joe Nadeau
Maine Medical Center
Portland

Referee #2 (Remarks to the Author):

The authors explore the association between environmental effects acting on prospective fathers and intergenerational transmission risk of disease in mice. While focus has been on microbiome-related mother-to-offspring inheritance of both bacteria and health implications in the current literature, little attention has been paid to the potential patrilineal effects that together with matrilineal inheritance affect offspring health and fitness. The authors examine the hypothesis that the paternal microbiome plays a role in the development of both health and fitness in the offspring. This would be an important finding if correct. The authors develop lines of evidence that support this notion, but as pointed out below, there are a number of weaknesses in the methods, missing controls, and in the specificity of the conclusions reached.

The authors attempt to shed light on this by using a model in C57BL/6J mice together with non-absorbable antibiotics (and other perturbation methods) to test whether dysbiosis in the gut microbiome (using fecal samples as proxy) may affect male germline and in association with that, offspring health. They treated pre-conception males with a non-absorbable antibiotic mixture (nABx) which perturbed their gut microbiota (or not), and then mated them. They found a wide variety of abnormalities in phenotypes of the offspring. 4 weeks after the cessation of the antibiotic mixture, there is partial recovery of the paternal intestinal microbiome, and after 8 weeks, there is more complete recovery.

The authors demonstrate that nABX-treated males sire offspring that show poorer health and lower fitness than offspring sired by control males. This effect was lost after allowing the treatment group males to recover. IVF experiments examined whether the treated males suffered from reproductive complications during fertilization, mostly manifesting in changes in placental ontogeny. Some of the results are less robust than the manuscript advertises. Studying the patrilineal inheritance of microbiome-related effects is of interest and deserving of attention.

One of the weaknesses of the manuscript is the lack of a control experiment in which antibiotics are given to pre-conception females in parallel. Mating is in essence a co-housing experiment, and the effects observed from the mating may represent the transfer of a perturbed microbiota from male to female. Thus, the principal mechanism for the observed findings might be the effect of the perturbed microbiota in the dams. This possibility should be examined directly by experiment, and not by inference.

The phenotypes in the IVF experiment are borderline. Despite large numbers of mice, there are no significant effects on weight after day 15 of life, and there are no significant effects on other phenotypes (S7D).

General remarks:

The authors have done a thorough methodology overview, and present their results in a visually-pleasing manner. Readers will benefit from specifying the details of the figures, however, by indicating whether the authors use means/medians, specifying the mean/median values, indicating sample sizes per figure etc. Authors are encouraged to deposit their data and any bioinformatic/statistical code freely available online, not only upon request.

The authors indicate in their results that they collected fecal samples prior to the antibiotic treatment as well. The manuscript would benefit from presenting that data and showing that prior to the nABX intervention (and/or other interventions), the control + prospective experimental groups were similar in their fecal microbiomes.

Considering that authors collect fecal samples (not colon nor small intestine), the manuscript should indicate that the findings are based on the fecal microbiome, rather than the gut microbiome. The two may differ substantially from one another; this point should be indicated and approaches to overcome this limitation addressed.

Potential key laboratory experiments should include amplifying bacterial DNA using the extractions of total sperm DNA, to examine whether the seminal microbiomes of the two male groups also differ. This would improve Fig S8. The same extraction kit has been used in the past to sequence bacterial DNA from semen samples.

Supplementary Figure 8 shows that the effects are not directional in terms of microbiome, but rather, rely on indirect effects.

Have authors considered the potential effect of aging of the fathers that happens across the experiment where the recovery weeks accumulate. At the end of the experiment, the individual is more than twice as old than at the start of the experiment (Figure 1B)?

Why did the authors use two strains of mice in the study instead of one? The C57BL/6J strain was used as the main mouse model for this project, while CD-1 IGS dams were used as surrogate mothers for IVF]. What are the known phenotypic manifestations of the two strains that may confound the particular IVF experiment? In addition, what are the potential confounding factors that may emerge due to using the inbred C57BL/6J strain? I.e., what is known about the gut microbiome, germline, offspring health of this strain that could have affected the observed results? The survival rates (Kaplan-Meier plots) are some of the most interesting data, considering that true fitness of a male is counted in their F2 offspring number and fitness. Considering that male mice may mate during the entirety of their life, showing a lower F1 survival rate can severely interfere with generational fitness and reduce the number of F2 offspring. This is of interest particularly as there is a recovery effect for the 6week+8week recovery group. Authors should directly examine F2 fitness. The relevant plots would benefit from specifying the sample sizes for each experimental group.

Authors should consider (and discuss) that while nABX were undetectable in circulating serum and testes, it is possible that the antibiotics still affected local niches and therefore, the local microbiota. Reproductive tract and ejaculate tract microbiota + its compatibility (or lack thereof) with female reproductive tract microbiota could be a key factor in some of the results, in addition to or instead of gut microbiome dysbiosis. How do these particular antibiotics interfere with reproduction-related structures (testes weight, sperm tubule structure) and features (sperm number etc.) that are independent of the microbiome. I.e., there may be a cumulative health burden caused by the antibiotics that is directly affecting individual reproductive health, in addition to (or in place of) the indirect effects relayed by the microbiome changes. Exposing germ-free male mice to antibiotics or not will provide a comparison to assess whether the observed antibiotic effects involve the microbiota or not.

It is unlikely that the use of nABX may have long-lasting effects for the siring males (as confirmed by the recovery period effects). However, measuring the testes, sperm number/concentration, sperm cell morphology/velocity in F1 and even F2 males could shed light on whether there is also

patrilineal inheritance of poor reproductive fitness. Equivalent measurements could be carried out in F1 and F2 female offspring to test for the effect of patrilineal effects on female offspring reproductive fitness.

These ideas should be used to re-structure the final paragraph of the manuscript; whilst there are suggestions supporting the "gut-germline axis", several gaps remain and authors should point out these ideas + potential alternative interpretations of the results.

Specific, line-by-line remarks:

Lines 6-7: Authors should specify that their work is carried out with mice. This would avoid generalizing the results across species, which at this point has not been confirmed. Authors should highlight the use of mice in all text sections, as it is not immediately evident when reading the different parts of the manuscript.

Figure 1C (and for all figures in the manuscript): Please specify whether the horizontal bars represent the mean or median, and indicate the actual value of the mean/median. The difference between the experimental groups may be significant, but the actual values do not seem to differ substantially; i.e., the phenotypic manifestation is subtler than the authors indicate. The authors should soften the vocabulary they use in the entirety of their manuscript by removing subjective constructs such as "surprisingly", "most strikingly however" , etc.

Figure 1D: Please indicate the total sample size for the severe developmental delay (SDD) group in comparison to the total sample size of the experimental group. Basing it on Figure 1C, it seems ~10. This is not that unexpected to happen at random in the wild, especially considering that two of the conventional mice (Figure 1C) seem to fall under SDD as well. If the authors were to exclude the SDD mice and re-analyze the pre-waning results, would the significant weight differences persist? Also—are these mice truly representative?

Figure 1G: Where would the CON SDD mice fall in the ordination (because there seem to be few CON SDD mice based on Figure 1C as well)? Is the difference truly because of treatment (CON vs nABX) or because of SDD?

Line 120: Typo "corelate". Change to "correlate".

Line 125: Change to "we then investigated".

Figure 3A: Typos in the words "Abnormal" and "Epithelial".

Supplementary Figure 1A: This would fit well in the main text together with Figure 1B.

S Figure 10. How were the metabolites shown selected? Why are these shown? Representative?

Referee #3 (Remarks to the Author):

In this manuscript, the authors seek to link changes in the paternal microbiome, induced by ingestion of non-absorbed antibiotics mix, with substandard growth and lethality in postnatal days P3-P18. Excellent controls are provided with restoration microbiome diversity, sperm transfer and related approaches, thus establishing a solid link between the exposure and the postnatal phenotype. They subsequently determine changes in testicular structure, metabolic and transcriptomic profiles, which the authors suggest might impact placental function and thus explain the phenotype. The mechanistic thread falls short of providing evidence for the role of testicular changes in placental phenotype, and subsequently, in linking placental changes to the postnatal observations. Thus, although the initial association between paternal microbiome changes and postnatal viability and growth (male "gut-germline axis") is original and tantalizing, everything in between constitutes interesting data without a clear teleological link. The paper is generally well-written, and the statistical analyses seem most appropriate and well-presented.

Specific comments:

1. What is the cause of increased postnatal mortality? The authors should systematically assess

key organs in pups with severe developmental delay (SDD). What was the placental phenotype of these SDD pups?

2. Could it be that the entire postnatal phenotype is related to paternal-effect imprinted genes? The author should provide more data specifically on paternally-expressed imprinted genes in sperms. Is it possible that the upregulation of tRF-Gly-GCC or the 5'tRNA fragments in mature sperms is related to the postnatal phenotype?

3. Leptin was clearly reduced in the nABX male testis, but what evidence ties it to the transgenerational effect?

4. The structural, epigenetic, transcriptional, and metabolic changes in the testis are fascinating. Which one of them explains the placental changes and the postnatal phenomenon?

5. The phenotype seems to localize primarily to a postnatal effect (P3 to P15-18). The authors indicate that nABX fathers had normal fertility and normal litter size. These highly suggest intact placental function. To establish a pregnancy/placental phenotype, irrespective of a mechanism, evidence of fetal weight restriction or death must be provided during the fetal period (E11-E18). Increased fetal/placental ratio (Fig 4G), if anything, suggests a more efficient placenta, which was able to support normal fetal development despite its smaller size. Lastly, invoking preeclampsia on the basis of gene expression changes as the mechanism for the postnatal growth changes seems somewhat superficial. The placental might respond to some paternally induced effects, but how does that explain the mostly postnatal phenotype?

Minor Comments:

1. Figure S3D title - should be "distribution". The legend to S5B - should be "frequency". Lines 119/120 should be "correlate"

2. Line 65; should mention brains of SDD offspring. Are figures 1H and S4D the same? Needed twice?

3. Line 168 refers to Fig S11-E-F, but there is no S11F. Line 189 and line 200 refer to Fig S13C, not S12C, as in the text.

4. Sometimes the authors use PIGF and sometimes PGF

Referee #4 (Remarks to the Author):

In this study, Argaw-Denboba et al. report that disruption of the microbiome of male mice has intriguing effects on their offspring. First, using different strategies, including ad lib non-absorbable antibiotics (nABX), a combination of other antibiotics (avaABX), and osmotic laxatives, the authors found that microbiota perturbation of male mice causes growth retardation and increased postnatal mortality of their F1 offspring. These phenotypic defects were rescued when the microbiota perturbation was removed for 8 weeks, indicating that it is a reversible defect. RNA-seq analysis revealed that expression of genes associated with metabolic processes are misregulated by nABX. Through performing in vitro fertilization using sperm from nABX-treated mice, the authors obtained evidence that the F1 phenotypes are transmitted via the sperm. nABX treatment led to smaller testes, lower sperm count, as well as morphological changes in testes seminiferous tubules, indicative of defects in spermatogenesis, not just sperm. The authors also performed unbiased metabolomic profiling and found that nABX treatment caused significant alterations in the levels of some metabolites in testes, raising the possibility that one or more of these changes is responsible for the spermatogenic and sperm defects. To begin to understand mechanism, the authors performed RNA-seq analysis of whole testes from the nABX-treated fathers. Among the most downregulated genes (~2-fold) was Leptin, a gene encoding a hormone associated with germ cell development and androgen signaling. The authors also performed small RNA-seq analysis of sperm, which revealed that multiple microRNAs are dysregulated by nABX. Finally, to further investigate potential mechanism, they performed RNA-seq analysis on the brain of the offspring (at E13.5) and placenta of the mothers pregnant with the F1 offspring. While they found little effects on gene expression in the embryonic brain, they identified many differentially expressed genes in the placenta. Interestingly, they observed downregulation of multiple Prolactin

genes and other genes essential for placental development (e.g., Hand1 and Syna). This raised the possibility that nABX treatment causes placental defects, which they directly demonstrated (altered F1 fetus/placenta weight ratio and reduced labyrinth zone).

This study provides compelling evidence for the origins of health and disease hypothesis: paternal stresses influence germline and the offspring. Thus, this work joins other key studies showing that offspring are influenced by paternal stresses, including low-protein diet (PMID: 30150380) and disruption of circadian rhythm (PMID: 34039610). Overall, this study is well designed and makes some interesting observations regarding how the paternal microbiome influences the male germline and offspring. However, there are several concerns that need to be addressed.

MAJOR CONCERNS

- 1) Statistics. While most panels are adequate in this regard, in some panels, the number of samples and/or statistical testing are not described clearly; e.g., Fig. 1B, 1E, 2E, 2I, S1B, S1E, S1F, S6B, S7D, S9B, S10A, and others.
- 2) Cell subset changes vs. gene expression changes in the testes. The authors overinterpret their RNA-seq data from whole testes. The testis contains both somatic and germ cells, both of which can be further divided into many different cell populations. Thus, when the authors observed changes in the "expression" of genes, this will often not be due to an alteration in expression per cell, but rather a change in the proportion of cells that express that gene. Considering the authors' finding that nABX influences the cell subset composition in the testis, it seems likely that many of the differentially expressed genes they identified are merely reflecting changes in the proportion of cell subsets. For example, Leptin was found to be downregulated ~2-fold after nABX treatment. However, the 2-fold reduction in its level may not be because of reduced expression per cell but rather because of reduced number of leptin-producing cells (spermatogonia and/or spermatocytes). To address this and other claims by the authors with regard to key RNA-seq results, they should perform: (i) FACS analysis of the relevant germ cell population (e.g., spermatogonia or spermatocytes) to test if there are changes in their frequency after nABX treatment, (ii) FACS-purify the relevant cell population and examine the expression of the gene in question (e.g., Leptin) by qPCR or similar method.
- 3) Cell subset changes vs. gene expression differences in the placenta. The same concept described above is relevant to the authors conclusions about gene expression in placenta. Thus, they should take the same steps as above to distinguish between gene expression vs. cell subset alterations. Given that the authors show nABX-treated mice have reduced labyrinth zone area, an obvious possibility is that many of the differentially expressed genes they identified results from this, not altered expression per cell.
- 4) Abstract. It is claimed that the authors uncovered the underlying mechanism for why paternal microbiome impairment causes defects in the offspring (e.g., through reduced leptin signaling), but actually only correlative (no rescue) data was provided. At the least, the Abstract should be revised with this in mind. Optimally, it would be desirable if the authors could perform experiments to directly address mechanism; e.g., rescue experiments. For example, the authors could ask whether Leptin depletion causes some of the phenotypic defects caused by microbiome disruption.

MINOR CONCERNS

- 1) RNA-seq validation. The authors should validate key dysregulated genes/microRNAs by RNA scope and/or TaqMan analysis. This also applies to RNA-seq analysis of placenta.
- 2) Some detailed experimental procedures need to be provided. For example, for the mouse mating. Are these mice mated within one day time window? Details about samples harvest, such as testes, placenta, brain, and brown adipose tissue.
- 3) Fig. S1D. Do all mice survive nABX treatment?
- 4) Fig. 2G. The PCA assay shows that samples from nABX 6wk+4rec are more distinct with Controls than those with nABX 6wk mice. Any explanations why 4wk recovery make them "worse"?
- 5) Line 38. The format of ref 25 is wrong.
- 6) Line 65. The authors performed "transcriptome profiling" (a preferable term is RNA-seq analysis), not "transcriptional profiling." The latter implies that transcription rate is measured, which is not correct. RNA-seq analysis measures steady-state RNA levels, which is a reflection of both RNA synthesis (transcription) and RNA decay (turnover).

- 7) Line 128. It is claimed that the vacuoles observed in the testes of mABX-treated males is due to germ cell loss, but no evidence is provided for this.
- 8) Line 130. Related to Major Concern point 2, above, it is critical to perform detailed histological analysis to determine the nature of the abnormalities in the testes of mABX-treated males; e.g., quantitative analysis of SPG, SPC, and STs.
- 9) Line 149. The reduction in SPC markers and increase in somatic cell markers is consistent with fewer SPCs and subsequent germ cell stages, leading to a relative enrichment in testicular somatic cells. This should be tested directly, as indicated in Major Concern, point 2, above.
- 10) Fig. S10B. The used marker lists of different cell types need to be provided.
- 11) Line 153. Leptin is not only expressed in spermatogonia, but spermatocytes (PMID: 17935159).

Response to Reviewer comments

Argaw-Denboba et al, Paternal microbiome perturbations impact offspring fitness

We would first like to thank the reviewers for their thoughtful and valuable feedback, that also considered our “*approaches comprehensive, the analysis rigorous, and the implications pioneering*”. Over the past eighteen months we have undertaken an extensive range of new experiments that address the comments raised, and serve to strengthen the manuscript further. We first briefly summarise the major lines of experimental revision below, and then respond (R) specifically to each comment (C) in full. Overall, we believe our improved manuscript provides compelling evidence that paternal gut dysbiosis impacts offspring fitness through influencing placental ontogeny.

The major experimental additions can be summarised as:

1. **Animal co-housing.** We have undertaken a major series of animal co-housing experiments to comprehensively investigate modes of phenotype and (putatively) microbiota transmission. These experiments first rule out the potential confounder of transmission of an altered microbiome from father to the mother. Secondly, by undertaking extensive breeding strategies following female co-housing with dysbiotic or control males, we empirically test how the intergenerational phenotype is transmitted.
2. **In vitro fertilisation.** We have undertaken a significant new line of *in vitro* fertilisation (IVF) experiments using BL/6 surrogate dams. These experiments independently confirm previous IVF with CD-1 dams (and natural matings), and thus provide strong evidence of paternal transmission of phenotype risk that can be transmitted specifically via sperm.
3. **Single-cell RNA sequencing.** To examine both cell-type specific responses and to further phenotype tissues, we have performed scRNA-seq on testes of dysbiotic or control fathers. This confirms the principle that gut microbiome perturbation leads to gene expression changes in multiple cell types within testes. It moreover improves resolution by identifying the cell-type proportionality changes, particularly in elongating spermatids. Finally, it further validates the importance of the ‘gut-germline axis’ concept we propose.
4. **Characterisation of microbiomes.** We have undertaken a careful analysis of the seminal fluid microbiome composition, which indicates no change, as predicted for exposure to non-absorbable antibiotics restricted to the gut. This argues against this potential modality of inheritance. We have additionally generated many new 16S datasets from mothers and fathers at dense timepoints around mating and following co-housing.
5. **Leptin characterisation and perturbations.** We have examined mechanistic questions by exploiting leptin-deficient mouse models. In response to direct *Leptin* perturbation we find phenotypic changes in testes physiology that are comparable to changes following microbiome perturbation on the same timescale. More remarkably, we also demonstrate a clear intergenerational impact of altering paternal leptin levels that manifests as early as the blastocyst stage.
6. **Placenta.** We have performed an in-depth additional histological phenotyping of the placenta, depending on the male status at the time of conception. This confirms previous observations, and reveals vascularisation defects associated with placental insufficiency and F1 phenotype.
7. **F2 transmission.** We have thoroughly tested whether the inter-generational effect (F1) also propagates trans-generationally to F2. Careful analysis has revealed no overt phenotype is transmitted to F2 progeny. This is consistent with a direct influence of male sperm status on placenta, which in turn impacts the developing fetus via placenta insufficiency. This has important implications for research into human paternal pre-conception health.

Referee #1 (Remarks to the Author):

C1: This paper tackles old questions in new ways, namely how does the microbiome affect host physiology and in turn how these physiological changes affect offspring phenotypes. The questions about the gut-germline connection are original, the approaches comprehensive, the analysis rigorous, and the implications pioneering. The work involves both a non-absorbable antibiotic as well as other means to induce dysbiosis, characterization of host molecular and physiological changes, phenotypic consequences among offspring, reversibility (too often overlooked), IVF tests for transmission through gametes, metabolic profiling, transcriptional profiling of testis, sperm and placenta, As I worked through the text, every time I had a question, wondering if it was an issue that had been overlooked, I found the question addressed with clear results. Among the many strengths of this study are the tests for reversibility. Such tests powerfully demonstrate that dynamic nature of the induced changes, but are often overlooked in studies such as this. Finally, the text is impeccably prepared.

With respect to the epigenetic aspects, the methylation profiling, the IVF experiment, and the intergenerational tests were carefully conducted, the results clear, and the interpretations appropriate.

R1: We are thankful for the supportive review of our study concluding that “*the intergenerational tests were carefully conducted, the results clear, and the interpretations appropriate*”. Below we respond to comments in full.

C2: The only question involves why the random-bred CD-1 females were used in the IVF test (Materials and Methods, Animal Husbandry, line 233); could molecular and physiological variation among genetically heterogeneous females lead to phenotypic ‘noise and perhaps even consistent phenotypic changes among offspring.

R2: This is an important question that we can address fully. First, CD-1 mice are the chosen model in the field as surrogate mothers for embryo-transfer after IVF. This is because they are known to be maternally competent, with a higher likelihood of conceiving after embryo transfer, and a higher rate of offspring survival and development (ref. *Manipulating mouse embryo book, 4th ed, page 93*). In other words, they increase the chance of IVF success, as they are efficient surrogate mothers. In contrast C57BL/6J (BL6) are known to be poor surrogate mothers, with low IVF success rates. Note that in the manuscript BL6 oocytes and sperm (latter derived from either dysbiotic or control father) were used, but that the surrogate mother receiving the fertilised embryo was CD-1 to maximise experimental success. This is important especially for experiments that require a long time to complete, such as ours, which takes 3-4 months per round.

Nevertheless, to empirically address the reviewer comment, and to provide further validation of our core result, we performed a series of additional IVF experiments, except now using inbred BL6 surrogate mothers. This resulted in a recapitulation of the effect; offspring derived from dysbiotic male sperm were of significantly lower weight than control (*see new Fig 2I*, and appended data below, right panel). Indeed, the effect size was actually larger with BL6 surrogate mothers than previous (significant) CD-1 data (*compare with new Fig 2H*, and left panel below). This is important for two reasons. *First*, it indicates the core effect is robust across different genetic backgrounds of gestating mothers, and also rules out ‘heterogenous female leading to phenotypic noise’. *Second*, it suggests that the previous high quality *in utero* environment from CD-1 mothers may have partially masked the paternal impact, and that the true background matched

offspring effect is actually larger. Consistently, it is noteworthy that control isogenic BL6 offspring born to CD-1 mothers are larger on average than the same isogenic BL6 offspring born to BL6 mothers, implying the CD-1 environment boosts offspring growth (albeit still resulting in a significant difference depending on paternal microbiome status). In summary, we observe phenotypic differences in offspring of dysbiotic males irrespective of the foster mother. This has proved to be an important experiment that provides strong additional lines of evidence to support our conclusions.

C3: Several analytical issues need a comment somewhere. The p-values are relatively modest, often $0.05 > p < 0.01$, which is somewhat surprising given the usually robust sample sizes ($n > 100$). This raises questions about the magnitude of the phenotypic effects. These effects can be found as approximations in the figures; more quantitative measures should be provided in the text, - phenotypic differences in measured units would suffice.

R3: The statistical framework is a critical point, that we are delighted to clarify both here, and now more clearly in the manuscript as well, since it's appropriate application is central to ensure robust conclusions in intergenerational studies. The *P-values* reported in the manuscript are based on a rigorous 'nested' analysis, whereby the replicate number is dictated by the number of uniquely exposed fathers (N), not by the number of offspring (n), which is greater. Thus, whilst we typically generate $n > 150$ offspring to robustly capture effect-size and partially penetrant phenotypes, statistically speaking each individual derived from the same father is hierarchically nested into a single N value. In other words, the phenotypes of offspring are analysed per exposed father rather than per individual offspring. This of course greatly reduces power to obtain significance. For example, in Fig 1C (also shown right), the indicated *p-value* is 0.023, based on a nested t-test. However, use of a standard t-test here, as is often deployed in intergenerational studies, derives a *p-value* of < 0.0001 , which is more in line with the apparent replicate number. This is however inappropriate as using individual offspring as independent variables will lead to inflated alpha error rates.

It is important to note that whilst we apply rigorous statistics, we still report significance across multiple replicate cohorts and orthogonal assays. Given that statistical significance is digital (yes or no) and does not indicate effect size, once the threshold is reached we can conclude a difference exists between the F1 populations based on paternal exposures. Effect size is captured in the forest plots that indicate 'risk' of F1 phenotypes emerging and in *Cohen's d* values. (e.g. see Fig 1E). In summary, the *p-values* we report reflect the nested statistical strategy that empowers confidence in conclusions. Indeed, going forward it is crucial that intergenerational studies apply nested analysis. We have now included the following sentence in the manuscript in order to further clarify our approach, line 47: "...applying a rigorous nested t-test", whilst also describing the statistics in the methods.

C4: Also, the text (line 119) mentions *Cohen's d*, which is a commonly used measure of effect size, but its use here is unusual – I've never seen it used to assess correlation - and might deserve some explanation and perhaps reconsideration. That said, the analysis is rigorous and nicely presented – nothing superficial or casual-, and could serve as an example for how analyzes should be done and presented.

R4: We realise that the phrasing was not entirely clear in this instance. In the analysis referred to by the reviewer, *Cohen's d* was to indicate phenotype strength in the F1 offspring, not a strength of association with microbiome richness. The corresponding Fig 2G shows that phenotype strength (as measured by *Cohen's d*) correlates with gut microbiome richness in the father, but not with microbiome richness in the F1 offspring themselves. We have modified the text to indicate this.

C5: Also, a summary statement is made (main text, lines 72-73, and then lines 217-219) and then reiterated in the final paragraph of the main text that the underlying biology is probabilistic rather than deterministic'. I understand the idea, but not its relevance here. This needs some clarification.

R5: There is a general perception that putative intergenerational effects may involve relatively linear relationships. That is, environment X promotes change Y and as a consequence phenotype Z. As such, experiments are typically designed with this in mind, for example, with relatively low replicate (n) numbers. One key conclusion of our data is that F1 outcomes are probabilistic in nature, or 'partially-penetrant', and require high powered datasets to capture. This has a significant implication for intergenerational inheritance effects more generally, including potentially in human. Specifically, it suggests that exposure to a specific context likely mediates complex responses that ultimately may affect offspring risk, but do not unequivocally lead to phenotypes. Indeed, in most cases no overt phenotype emerges. This shift to probabilistic thinking is, we think, an important outcome to emphasise.

C6: On whom – which generation- was the metabolic profiling performed (lines 134-135)? This should be provided in the main text.

R6: We harvested testes from the fathers (F0). We modified the following sentences to clarify the source of the tissue: Line 143 "*This prompted us to investigate the physiological changes in the father's reproductive system that are induced by acute microbiome dysbiosis.*". Likewise, from line 155 "*To characterize the responses to gut dysbiosis at the molecular level, we performed untargeted metabolomic profiling of father's testis,*"

C7: The font size in the figures is too small, e.g. Fig 1H is unreadable.

R7: We thank the reviewer for pointing this out, and agree the font is simply too small. This figure is intended to illustrate the principle that the molecular changes in independent offspring from different dysbiotic fathers (each column) are highly comparable. For clarity, we have now reduced the number of plotted genes in Fig 1H such that we can increase the font size of each for readability in the available space. A figure that contains all gene names in a legible format across different tissues is also available in Extended data 4.

C8: There were several questions that go beyond the scope of work needed in this manuscript, but should be raised in the "Discussion" (last paragraph).

C8.1: The first concerns the mechanistic connection between changes in the gut with the germline – what are the molecules and pathways that transduce signals (and what role might this network serve under normal conditions).

C8.2: Then, do more subtle changes in microbiome content or functionality induce comparable phenotypic changes as those induced with antibiotics?

C8.3: Also, the text refers to changes in diet as well as antibiotics, but diet tests are not presented here (and are not needed) – how might diet and commonly used antibiotics impact these intergenerational effects.

R8.1: These are important and intriguing questions that indeed, go beyond this work, and that we are actively pursuing over the coming research cycle. Here we have added analysis of the effects of systemic leptin changes in response to microbiome perturbation and also shown altered systemic metabolites, which along with other factors, could underlie communication between the gut and testis.

Future work will identify the specific microbial parameters involved. In the current manuscript, we are tightly limited by space parameters, but have nonetheless highlighted these questions more prominently in the discussion, and pointed out that further research is necessary.

C8.4: Finally, how might the seemingly general changes in the germline lead to fairly specific and reproducible phenotypic changes in the next generation.

R8.2: It is an intriguing question, since it was one of the main premises for initiating this project. Our hypothesis was that because gut microbiota are lifetime partners of the host, dysbiosis (perturbation of gut microbiota) could trigger reproducible phenotypic changes in subsequent generations, regardless of the cause of dysbiosis. There have been many extrinsic factors implicated as potential antecedents of paternally induced inter-/trans-generational effects, including hazardous chemicals, drugs, nutritional deficiencies, lifestyle changes, etc. However, because male germ cells interact with multiple risk factors at different stages of spermatogenesis throughout life, it is more complex to find the underlying causes of relatively specific environmental factors that drive reproducible phenotypic changes in subsequent generations. Throughout life, the gut microbiota inhabits and evolve with the host, and dysbiosis poses a risk at the interface of host-environment interactions, naturally serving as a hub for multiple environmental factors. Hence, to reduce these multidimensional environmental factors into one-dimensional triggering factors, we developed an antibiotic-inducible dysbiosis model. To this end, this manuscript aimed at examining the impact of preconception paternal gut dysbiosis on the male reproductive system and its implications for their offspring. Indeed, regardless of the dysbiotic agents we used (i.e. non-absorbable antibiotics, absorbable antibiotics, or laxatives like PEG), we observed similar phenotypic consequences in the offspring. Moreover, placental insufficiency to various degrees during pregnancy is likely to represent a common cause of postnatal offspring phenotypes. Thus, dysbiosis may trigger predictable changes in male reproductive systems, leading to reproducible phenotypic changes across generations through placental insufficiency.

Referee #2 (Remarks to the Author):

C1: The authors explore the association between environmental effects acting on prospective fathers and intergenerational transmission risk of disease in mice. While focus has been on microbiome-related mother-to-offspring inheritance of both bacteria and health implications in the current literature, little attention has been paid to the potential patrilineal effects that together with matrilineal inheritance affect offspring health and fitness. The authors examine the hypothesis that the paternal microbiome plays a role in the development of both health and fitness in the offspring. This would be an important finding if correct. The authors develop lines of evidence that support this notion, but as pointed out below, there are a number of weaknesses in the methods, missing controls, and in the specificity of the conclusions reached.

The authors attempt to shed light on this by using a model in C57BL/6J mice together with non-absorbable antibiotics (and other perturbation methods) to test whether dysbiosis in the gut microbiome (using fecal samples as proxy) may affect male germline and in association with that, offspring health. They treated pre-conception males with a non-absorbable antibiotic mixture (nABx) which perturbed their gut microbiota (or not), and then mated them. They found a wide variety of abnormalities in phenotypes of the offspring. 4 weeks after the cessation of the antibiotic mixture, there is partial recovery of the paternal intestinal microbiome, and after 8 weeks, there is more complete recovery.

The authors demonstrate that nABX-treated males sire offspring that show poorer health and lower fitness than offspring sired by control males. This effect was lost after allowing the treatment group males to recover. IVF experiments examined whether the treated males suffered from reproductive complications during fertilization, mostly manifesting in changes in placental ontogeny. Some of the

results are less robust than the manuscript advertises. Studying the patrilineal inheritance of microbiome-related effects is of interest and deserving of attention.

R1: Thank you for the thoughtful feedback. Your comprehensive review of the manuscript and the questions you raised were extremely helpful in improving the manuscript and fully addressing its potential limitations. Moreover, we agree that “*studying the patrilineal inheritance of microbiome-related effects is of interest and deserving of attention*”. Given the implications of this, we produced datasets built on exceptional numbers of replicates and orthogonal strategies to have confidence in our conclusions. Specifically, the results presented come from three independent dysbiotic agents in fathers, each with natural matings generating unprecedented offspring sample sizes (by order of magnitude), with a longitudinal rescue paradigm built in, supported by independent IVF, and applying a rigorous nested statistical analysis. Thus, we believe the intergenerational phenomenology reported is founded on robust experimental designs. Nevertheless, in recognition of the concerns raised, the revised version now includes extensive new datasets that address potential confounders and also support and extend our earlier findings. A point-by-point response is below.

C2: One of the weaknesses of the manuscript is the lack of a control experiment in which antibiotics are given to pre-conception females in parallel. Mating is in essence a co-housing experiment, and the effects observed from the mating may represent the transfer of a perturbed microbiota from male to female. Thus, the principal mechanism for the observed findings might be the effect of the perturbed microbiota in the dams. This possibility should be examined directly by experiment, and not by inference.

R2: We have addressed this point in three independent (experimental) ways.

(1) First, we co-housed nABX or control males with females and examined the microbiota composition following 4 days co-housing (and mating) by 16S sequencing to determine if there is cross-transfer of a perturbed microbiome. We found that **no significant microbial transfer from dysbiotic males to the dam during the mating process** (*see new Extended data Fig S8D-E*). This is consistent with the father carrying a depleted microbial load. Moreover, this data supports our previously presented analysis that revealed no change in the composition or richness of the postpartum maternal or offspring microbiome, irrespective of mating with control or nABX-exposed male (*see Extended data Fig 8A-B*).

(2) Second, given mice are coprophagous, we also asked whether trace antibiotic residue in the male faeces may affect the mother. To test the functional significance of this, or other indirect paternal effects on mothers, we undertook a major co-housing *and* mating strategy. Here, female mice were again co-housed for 4 days with either nABX or control male mice (only at daytime to prevent mating). Females were subsequently mated with only untreated male mice (independent) and offspring examined. **We found no significant change in offspring survival or birth weight, irrespective of whether females had previously been co-housed with an nABX or a control male** (*see new Extended data Fig 8G-H, also summarised below*). Taken together, this strongly suggests that neither female ingestion of antibiotic residues nor direct microbial transfer from males underlies the intergenerational phenotype.

(3) Finally, we performed *in vitro* fertilisation assays, wherein there is no interaction between a male and female, and only purified gametes are utilised. This reproduced significant offspring weight reduction in the offspring of nABX treated fathers (*see new Fig 2H-I*). Considering this with the new co-housing experiments that do not elicit an effect, and additional 16S data showing no significant maternal changes postpartum or after mating, we can conclude the phenotypic effect is primarily transmitted via modified germ cells rather than microbial changes in the dam.

In response to the reviewer's comment regarding the inclusion of pre-conception antibiotic-treated females as a control. An antibiotic perturbation during pre-conception period will also extend into pregnancy in female mice, since microbiome recovery takes 6-8 weeks, which is longer than the duration of pregnancy. Experimentally this approach would therefore be difficult to consider as a control but rather a maternal *in utero* perturbation. Since our study is limited to understanding the intergenerational effects of gut dysbiosis that occurs prior to conception, male mice are an ideal model, with female mice naturally carrying forward pre-conception effects into pregnancy. Accordingly, while we agree with the reviewer that a female study will be interesting, including a female model is beyond the scope of this manuscript.

C3: The phenotypes in the IVF experiment are borderline. Despite large numbers of mice, there are no significant effects on weight after day 15 of life, and there are no significant effects on other phenotypes (S7D).

R3: As noted by the reviewer, IVF-produced offspring from nABX-treated males recapitulates much but not all of the growth trajectory phenotype of their naturally born counterparts. This is partially expected, since IVF and embryo transfer to CD1 surrogates are more likely to produce experimental and biological selection bias than natural pregnancy, potentially resulting in fewer pups with mild/severe phenotypes. To briefly summarize these technical and biological biases; (1) *In vitro* fertilization of oocytes is carried out only with those 10% high quality epididymal sperm cells isolated by the swim up assay, eliminating the 90% of poorer quality sperm; (2) Embryo transfer is performed at 2-cell stage, and only those embryos that appear healthy with a good quality score are transferred to CD1 surrogate mothers; (3) It is known that CD1 mice are excellent surrogate mothers, yet they have lower rates of pregnancy and live birth (~60-70%) compared with the number embryos transferred at the 2-cell stage (i.e. 20 embryos/mouse), indicating the possibility of losing those embryos with severe phenotypes; (4) Whilst IVF offspring *do* recapitulate significantly lower birth weight and delayed growth, we noticed that IVF offspring (from nABX *and* control fathers) were heavier than their naturally conceived counterparts (*see Fig 2H*), which is explained by known CD1 high-quality *in utero* and maternal nursing. This could mask the severe growth phenotypes. Overall such technical and biological factors may reduce/obscure the natural phenotypic differences.

It is however critical to note the *p*-values are significant for birthweight by IVF. This confirms the fundamental message that gametes from paternal nABX males can induce an intergenerational phenotype. Moreover, all *p*-values are based on a rigorous "nested t-test" framework, in which offspring traits are examined per father rather than per individual (*see R3* on page 3 for full explanation and implication). For this reason, statistically speaking we do not have "large numbers of mice" but rather nest all statistics into the number of unique fathers (typically N=8-12 for IVF). Indeed, if we use a standardised t-test, as is usually applied, we find all IVF weight phenotypes reach strong significance, that is $p < 0.001$.

To add more experimental weight to the IVF data, the revised manuscript now includes an entirely new set of IVF experiments, in which we used a C57BL/6 (BL6) foster mother. Whilst these match the genetic background of the embryo, they are rarely used as surrogate mothers since they are known for poor pregnancy rates after embryo transfer (*refer to manipulating mouse embryo book, 4th ed, page 93*). Despite several pregnancy failures following embryo transfer, we managed to obtain sufficient data to support our conclusions. Specifically, even with relatively low numbers we observe a significant reduction in postnatal weight of offspring from nABX fathers. Indeed, the effect size is actually

considerably greater than with CD1 foster mothers (*see new Fig 2I*), consistent with the idea that the high-quality maternal care from CD1 surrogates may compensate for F1 phenotypic effects. In summary, **two independent lines of IVF demonstrate significant F1 neonatal weight phenotypes, using robust statistics, and despite the selection bias and additional experimental noise intrinsic to IVF**. In response to the reviewer comments we have however also modified our conclusion to point out that IVF may not capture all the transmission by stating that paternally-induced F1 phenotypes only occur “*primarily via the gametes*”, line 135.

C5: The authors have done a thorough methodology overview, and present their results in a visually-pleasing manner. Readers will benefit from specifying the details of the figures, however, by indicating whether the authors use means/medians, specifying the mean/median values, indicating sample sizes per figure etc. Authors are encouraged to deposit their data and any bioinformatic/statistical code freely available online, not only upon request.

R5: We went through the entire manuscript to confirm we have indicated the specific statistical parameters and sample sizes in figure legends and/or directly within figure panels, as well as in text. We try to be open with all data, which is already deposited in repositories.

C6: The authors indicate in their results that they collected fecal samples prior to the antibiotic treatment as well. The manuscript would benefit from presenting that data and showing that prior to the nABX intervention (and/or other interventions), the control + prospective experimental groups were similar in their fecal microbiomes.

R6: This information is shown in Fig 1B and Extended data Fig 1A-B. The 0wk timepoint represents the taxa richness in the randomly allocated male groups assayed just prior to nABX treatment initiation (CON vs. prospective nABX). As expected, this demonstrates no differences between the groups. We have amended the nomenclature of the figure/legend to make this clearer, for example, specifying “t0” rather than “wk0”.

C7: Considering that authors collect fecal samples (not colon nor small intestine), the manuscript should indicate that the findings are based on the fecal microbiome, rather than the gut microbiome. The two may differ substantially from one another; this point should be indicated and approaches to overcome this limitation addressed.

R7: We have revised the manuscript and changed the phrasing to "fecal microbiome" at key points where we refer to analysis of fecal material. It should be noted however that the fecal microbiome profile was used as the proxy for gut microbiota - an established method for studying gut microbiota - to demonstrate differences between the groups. Indeed, the phenotype characteristics reported in the manuscript are the result of specifically perturbing resident microbiota in the mouse gut, thereby the findings of this study should be interpreted with regard to gut microbiota. Further, due to the nature of the study (a longitudinal analysis of individual males at different times of mating and a microbiome analysis), it is not possible to obtain a cecum sample for analysis.

C8: Potential key laboratory experiments should include amplifying bacterial DNA using the extractions of total sperm DNA, to examine whether the seminal microbiomes of the two male groups also differ. This would improve Fig S8. The same extraction kit has been used in the past to sequence bacterial DNA from semen samples. Supplementary Figure 8 shows that the effects are not directional in terms of microbiome, but rather, rely on indirect effects.

R8: Thank you for proposing this experiment. We have now generated a complete set of new experimental 16S sequencing data from multiple microbiome sources. As suggested by the reviewer,

this includes seminal fluid, as well as copulatory plugs (coagulated semen deposited in the female genital tract after mating). To guard against ambient or non-specific contamination, we enacted rigorous sterilization protocols and incorporated multiple controls, for example negative controls include swabs from the sample processing environment (i.e. working area + dissecting tools). Within these highly rigorous conditions, we were unable to detect a microbiome specific to seminal fluid (it was comparable to negative controls signals) (*see new Extended data Fig 8F*), and further found no differences between control or nABX males from the signal we acquired. We robustly detected a copulatory plug microbiome but again there was no difference between the control and nABX groups. Finally, throughout our study we deployed distinct gut microbiome perturbations, that include oral laxatives and non-absorbable antibiotics, which do not reach a systematic level (as confirmed by mass spectrometry in testes) (*see Extended data Fig 2*). They therefore would not be expected to have an impact on distal microbiomes, which is consistent with our additional data here.

C9: Have authors considered the potential effect of aging of the fathers that happens across the experiment where the recovery weeks accumulate. At the end of the experiment, the individual is more than twice as old than at the start of the experiment (Figure 1B)?

R9: The design of the central experiment is to initiate microbiome perturbation in 5wk old mice for a period of 6 weeks. We then mate and compare the F1 impact between age-matched parallel-cohort control and treated animals, which are all at 11wks old at first mating. The vast majority of data in the manuscript, relating to three independent microbiome perturbations and IVF, is therefore derived from 11wk old males and age-matched controls and therefore ageing is not a factor. In addition, as pointed out we build in a rescue experiment that includes up to eight weeks microbiome recovery (6wk+8rec), where we again compare between age-matched and parallel control vs treated mice. Thus, all mice in our model mated between 11-19 weeks of age, which is under the age at which mice are affected by senescence, and corresponds to less than twice as old. Generally, mature adult mice range in age from 3-6 months, which is the reference group used as a control group in aging studies (JAX® website, Guide: Aged C57BL/6J Mice for Research Studies). As the reviewer points out, age-related impact is an important biological question that should be explored in the future, but not in the age range we worked with, rather in mice older than 10 months.

C10: Why did the authors use two strains of mice in the study instead of one? The C57BL/6J strain was used as the main mouse model for this project, while CD-1 IGS dams were used as surrogate mothers for IVF]. What are the known phenotypic manifestations of the two strains that may confound the particular IVF experiment? In addition, what are the potential confounding factors that may emerge due to using the inbred C57BL/6J strain? I.e., what is known about the gut microbiome, germline, offspring health of this strain that could have affected the observed results?

R10: In our previous answer to comment 3 (**R3**) above, we covered the biological and experimental aspects related to this as well as additional data. Summarizing: CD1 mice are preferred strains to be used as surrogate mothers, not only for their high pregnancy efficiency and live birth rate, but also for their quality of maternal care and nursing ability. In contrast, C57BL/6J surrogate mothers have low pregnancy efficiency from IVF surgeries, and are therefore rarely used. Thus, whilst all assays/perturbations for both natural matings and IVF were performed exclusively using C57BL/6J fathers and mothers (genetically), specifically the surrogate mother was CD-1 for IVF embryo transfers. However, the revised manuscript now includes new IVF data also using C57BL/6J surrogate mothers, which despite the technical challenges in obtaining sufficient replicates, reproduces the significant intergenerational effect, actually with a greater effect size (*see new Fig 2I*). This further supports our conclusions. To our knowledge, this is the first study in mammals that links gut microbiome-germline and offspring health. Whilst it is possible that different genetic backgrounds will introduce complex interactions between the genome, microbiome and environment, we have used the extremely well characterized and inbred C57BL/6J strain (near isogenic). This largely reduces the impact of genetic variation, whilst a whole genome sequence analysis found no evidence for a genetic basis of the

phenotypes observed (*see Extended data Fig 6*). Moreover, our study has used an unprecedented number of replicates (by order of magnitude) within one strain precisely to enable robust conclusions on the phenomenology, which has been a challenge in the field.

C11: The survival rates (Kaplan-Meier plots) are some of the most interesting data, considering that true fitness of a male is counted in their F2 offspring number and fitness. Considering that male mice may mate during the entirety of their life, showing a lower F1 survival rate can severely interfere with generational fitness and reduce the number of F2 offspring. This is of interest particularly as there is a recovery effect for the 6week+8week recovery group. Authors should directly examine F2 fitness. The relevant plots would benefit from specifying the sample sizes for each experimental group.

R11: We fully agree that the survival plots are amongst the most interesting data, in part because mortality is an unambiguous and highly quantitative phenotype, thus enabling robust conclusions about intergenerational effects. Thanks for bringing up this intriguing question about transgenerational effects, which we answer both conceptually and experimentally as follows. (1) First, we want to emphasize that this study is not designed to evaluate the *trans*generational impact on F2 descendants, rather it is designed to examine the *inter*generational impact on direct F1 offspring health. Nevertheless, irrespective of transgenerational (F2) phenotype, there is still an F2 fitness disadvantage to the original dysbiotic male (F0), since the F1 offspring are at increased risk of premature mortality. The father therefore produces less reproductively active offspring, which would propagate to a decreased number of F2 offspring on average. (2) Nevertheless, in light of the reviewer's suggestion, we chose to empirically test whether a phenotype is detectable in the F2 descendants from dysbiotic fathers, by undertaking a major line of experiments at considerable effort. This data revealed no significant differences between F2 animals born to F1 siblings derived from nABX treated fathers relative to controls, implying the primary effect is *inter*generational in nature and consistent with a paternal impact on *in utero* placenta. This data further support the conclusion that our intergenerational inheritance phenotype is non-genetic in nature, since genetic changes should propagate to F2 effects (*see new Extended data Fig 7*, and appended right). It is also worth pointing out that the F1 phenotype is probabilistic, and to some extent segregates into severely impacted or overtly normal offspring. Because the severely impacted offspring typically suffer premature mortality prior to sexual maturity, we are forced to investigate transgenerational effects using the overtly normal F1 from dysbiotic fathers. Therefore, we cannot rule out an F2 effect from the subset of unfit F1. This new data is now included in the manuscript.

C12: Authors should consider (and discuss) that while nABX were undetectable in circulating serum and testes, it is possible that the antibiotics still affected local niches and therefore, the local microbiota. Reproductive tract and ejaculate tract microbiota + its compatibility (or lack thereof) with female reproductive tract microbiota could be a key factor in some of the results, in addition to or instead of gut microbiome dysbiosis. How do these particular antibiotics interfere with reproduction-related structures (testes weight, sperm tubule structure) and features (sperm number etc.) that are independent of the microbiome. I.e., there may be a cumulative health burden caused by the antibiotics that is directly affecting individual reproductive health, in addition to (or in place of) the indirect effects relayed by the

microbiome changes. Exposing germ-free male mice to antibiotics or not will provide a comparison to assess whether the observed antibiotic effects involve the microbiota or not.

R12: Thank you, we appreciate the reviewer interest in understanding the effects of antibiotics that go beyond their direct gut microbiome perturbation. As noted by the reviewer we used a low concentration of non-absorbable antibiotics that are largely restricted from entering systemic circulation, and that could not be detected in testis using sensitive mass spectrometry approaches (*see Extended data Fig 2A-C*). Using a functional test for antibiotics we were also unable to detect residual anti-microbial activity in distal tissues (*see Extended data Fig 2D*). We have now further directly assayed distal microbiomes, including careful background controls, and we were unable to detect changes. Thus, the effect of these antibiotics outside of the gut is likely to be negligible or tolerable. Our study nevertheless also attempted to address the question of indirect effects of the nABX cocktail empirically by using orthogonal gut microbiome perturbation strategies: (1) a laxative (PEG), which cannot directly impact distal microbiomes (*see Fig 1J*), and (2) terminating the nABX antibiotic treatment for 4 weeks, which maintains a gut microbiota dysbiosis but provides ample time to flush any putative antibiotics from the system (*see Fig 2A, 2C, 2E*). In each case, phenotypic effects were observed in the offspring. Hence, the phenotype is independent of dysbiotic agents (nABX, avaABX, or PEG), but rather is the consequence of processes that are linked by converging specifically on inducing dysbiosis of the gut microbiota.

Nonetheless, we agree with the reviewer that a cumulative physiological effect may occur, which makes sense given that gut microbiota dysbiosis has been shown to affect distal organs of the body. Future studies should investigate the correlated systemic effect in collaboration with groups studying the interaction of microbiota with the brain, endocrine system, liver, bile...etc, to better understand the non-direct mechanisms that may influence the gut-testis axis. As an example, our study found that leptin is directly responsive to acute gut microbiota perturbation by nABX (*see Fig 3H-I*), and that direct leptin perturbation induces testicular and offspring phenotypes (*see new Fig 3J & Extended data 13*). This effect could potentially involve multiple pleiotropic actions of leptin, including those on the brain.

C13: It is unlikely that the use of nABX may have long-lasting effects for the siring males (as confirmed by the recovery period effects). However, measuring the testes, sperm number/concentration, sperm cell morphology/velocity in F1 and even F2 males could shed light on whether there is also patrilineal inheritance of poor reproductive fitness. Equivalent measurements could be carried out in F1 and F2 female offspring to test for the effect of patrilineal effects on female offspring reproductive fitness.

R13: Thank you, indeed as indicated by the reviewer we do not detect any long-term reproductive impact on nABX exposed males. We have now also completed an analysis of the reproductive fitness and fecundity potential of F1 siblings (F1-male x F1-female) born to the nABX sire and the control group, as discussed in R11. There were no significant differences (*see new Extended data Fig 7A-D*), indicating that the F1 generation had comparable reproductive fitness when not exhibiting the (S)GR growth phenotype. Regarding reproductive impact on F2 offspring (which is detected as an effect on F3), in line with the explanation provided above, this falls beyond the scope of this manuscript, which has focused considerable effort on obtaining high-quality robust data on F1 effects

C14: These ideas should be used to re-structure the final paragraph of the manuscript; whilst there are suggestions supporting the “gut-germline axis”, several gaps remain and authors should point out these ideas + potential alternative interpretations of the results.

R14: Taking the reviewer's comments into consideration, we have revised the conclusion to include more nuance, and highlight there is more to be determined. Further, in addition to the comments above, we have generated single-cell RNA-seq data from testicular tissues in nABX-treated males, which further confirms a cross-talk between gut and germline in male reproductive system (*see new Extended data Fig 11*) thus supporting a ‘gut-germline axis’.

Specific, line-by-line remarks:

C15: Lines 6-7: Authors should specify that their work is carried out with mice. This would avoid generalizing the results across species, which at this point has not been confirmed. Authors should highlight the use of mice in all text sections, as it is not immediately evident when reading the different parts of the manuscript.

R15: Thanks for your comment, we have gone through and modified the manuscript to ensure that it is clear we use a mouse model, including mentioning it earlier in the abstract. Further, the use of mice was mentioned in the first line of the results section “*To investigate this, we established an inducible model of gut microbiota dysbiosis in isogenic male mice using ad lib non-absorbable antibiotics (nABX)*”. In addition, mouse schematics are deployed in the first figure (*see Fig 1A*), and throughout other figures. Finally, we have now modified the conclusion to highlight our results are in murine models and cannot be transposed “*...future work is necessary to deconvolve the phenotypically-relevant modalities and their applicability beyond murine models.*”

C16: Figure 1C (and for all figures in the manuscript): Please specify whether the horizontal bars represent the mean or median, and indicate the actual value of the mean/median.

R16: The bar in *Fig 1C* indicates the mean value, which was indicated in the figure legend. We have checked through to ensure the specific metric displayed in each figure is clearly marked.

C17: The difference between the experimental groups may be significant, but the actual values do not seem to differ substantially; i.e., the phenotypic manifestation is subtler than the authors indicate. The authors should soften the vocabulary they use in the entirety of their manuscript by removing subjective constructs such as “surprisingly”, “most strikingly however”, etc.

R17: We have revised the manuscript accordingly and removed some of the subjective phrases we use. Conversely, our principal phenotype is increased risk of lethality. We believe this to be a significant phenotypic manifestation that was indeed surprising for an intergenerational impact.

C18: Figure 1D: Please indicate the total sample size for the severe developmental delay (SDD) group in comparison to the total sample size of the experimental group. Basing it on Figure 1C, it seems ~10. This is not that unexpected to happen at random in the wild, especially considering that two of the conventional mice (Figure 1C) seem to fall under SDD as well.

R18: Thank you for the comment, which is important to expand upon. (1) First, the SDD (now renamed severe growth restriction (SGR) for accuracy) shown in *Fig 1C* are only offspring that survived up to their 15th postnatal day (P15). Importantly, the majority of growth restricted offspring die before reaching this age. Moreover, the SGR phenotypic outcome was consistent across all dysbiotic agents we used, including the 4-week recovery period (*see Fig 1F, 1I, 1J, 2C*), where many of the pups with severe growth problems also do not reach P15. As such, the prevalence captured at P15 does not represent all SGR offspring born to dysbiotic fathers, rather those who survived to that age. For this reason, we underestimate the impact using weight metrics whilst mortality plots capture the full impact (see below). (2) The two specific mice showing lower growth in the control group of *Fig 1C* were caused by sporadic anomalies - malocclusion and hydrocephalus - respectively. This was clearly a distinct phenotype from the severe growth phenotype reproducibly observed nABX-derived offspring. We of course did not exclude them from the dataset, but the underlying cause is different. Indeed, in all independent control F1 dataset generated in this study (corresponding to >400 offspring), we only observed the SGR phenotype once e.g. *see large control dataset in Extended data Fig 5A*. In contrast

we observed over 35 cases of SGR offspring at P15 from dysbiotic fathers. (3) Importantly, even with these control offspring included in Fig 1C, the prevalence of SGRs was considerably higher in nABX than in the cohort-matched control group, and reached strong statistical significance. Taken together, if anything we believe we have taken a stringent approach to ensure our conclusions are robust, but nevertheless reproducibly observe statistical significance considerably above background (control) levels. Finally, mortality is an unambiguous phenotype. Across paternal perturbations we find highly significant increased rates of F1 premature mortality (appended below).

C19: If the authors were to exclude the SDD mice and re-analyze the pre-weaning results, would the significant weight differences persist? Also—are these mice truly ‘representative [Figure 1D]?’

R19: As can be seen in the reanalyzed result (appended right), the significant F1 weight difference persists when excluding SGR mice, indicating a population level impact. Moreover, we report *p-values* in the manuscript based on a rigorous "nested-t-test" instead of the standard t-test, thus quantifying phenotype significance at the level of the father rather than per offspring. This minimizes the influence of SGR outliers, that may for example arise from a single litter. The principle that there is a change in the distribution is shown in the QQ plots and kurtosis values (see Extended data Fig 5B).

The images shown in Fig 1D are affected individuals from two independent litters born on the same day and are truly representative of the severity of the partially-penetrant SGR phenotype. Moreover, it is also important to note that the SGRs shown in our data are only those that survived beyond the 15th day. We reached the conclusion that SGR mice represent a true phenotype on the basis of the empirical datasets presented in the manuscript, displaying a consistent phenotype across different dysbiotic agents (*i.e.* nABX Fig 1C, avaABX Fig 1I, and PEG Fig 1J), including the 4-week recovery period when dysbiosis persists (see Fig 2E).

C20: Figure 1G: Where would the CON SDD mice fall in the ordination (because there seem to be few CON SDD mice based on Figure 1C as well)? Is the difference truly because of treatment (CON vs nABX) or because of SDD?

R20: Taking the above explanation into account, the two offspring showing developmental delay in the control group were caused by sporadic anomalies in the mice colony, malocclusion and hydrocephalus, respectively. Whilst it likely occurs at low frequency we did not capture any instances of SGR amongst all controls datasets ($n > 400$ offspring). If these phenotypes recurred, we could have performed RNA-seq, but their rarity means we couldn't test this question. From a pathophysiological perspective, the two CON pups suffer from two different anomalies, malocclusion and hydrocephalus, and thus their molecular phenotype results are unlikely to cluster with the SGRs from the nABX.

C21: Line 120: Typo “corelate”. Change to “correlate”..

Line 125: Change to “we then investigated”.

Figure 3A: Typos in the words “Abnormal” and “Epithelial”.

R21: Thank you for spotting these. All corrected.

C22: Supplementary Figure 1A: This would fit well in the main text together with Figure 1B.

R21: Given the quantity of data we have generated we are severely restricted in space. Whilst we agree that Supplementary Figure 1B would complement the main figure, it is not possible to fit it in without detriment to figure readability or removal of other important panels. It remains accessible in the extended data

C23: S Figure 10. How were the metabolites shown selected? Why are these shown? Representative?

R23: They were selected primarily because they are representative of the overall metabolite pathway enrichments and/or because of the role they play in the reproductive system, either directly or indirectly. We can not call out all significant metabolites given their number, so chose to highlight those that may be most representative.

Referee #3 (Remarks to the Author):

C1: In this manuscript, the authors seek to link changes in the paternal microbiome, induced by ingestion of non-absorbed antibiotics mix, with substandard growth and lethality in postnatal days P3-P18. Excellent controls are provided with restoration microbiome diversity, sperm transfer and related approaches, thus establishing a solid link between the exposure and the postnatal phenotype. They subsequently determine changes in testicular structure, metabolic and transcriptomic profiles, which the authors suggest might impact placental function and thus explain the phenotype. The mechanistic thread falls short of providing evidence for the role of testicular changes in placental phenotype, and subsequently, in linking placental changes to the postnatal observations. Thus, although the initial association between paternal microbiome changes and postnatal viability and growth (male “gut-germline axis”) is original and tantalizing, everything in between constitutes interesting data without a clear teleological link. The paper is generally well-written, and the statistical analyses seem most appropriate and well-presented.

R1: We thank the reviewer for their careful analysis of our study and insightful comments throughout. In particular, we agree with the summary that the core phenomenology is “*original and tantalizing*” and that the statistical analysis establishes a “*solid link between the exposure and the postnatal phenotype*”, since this is the central message of our manuscript. The underlying mechanisms of this important observation are likely complex and pleiotropic, and we do not claim to have discerned them fully. The questions you raised about mechanisms however have proved extremely helpful in improving the study. In particular, we gained new insight into how paternal leptin levels may influence preimplantation embryos; a new intergenerational effect previously unexplored. Please find below detailed response to all your comments.

Specific comments:

C2: What is the cause of increased postnatal mortality? The authors should systematically assess key organs in pups with severe developmental delay (SDD). What was the placental phenotype of these SDD pups?

R2: In spite of our efforts to find the organs that contribute to SDD lethality (now renamed ‘severe growth restriction’ (SGR) for accuracy), pinpointing the cause has been, and remains, challenging. Particularly, growth retardation makes offspring more susceptible to multiple compound physiological complications (*see Extended data Fig 2*, which depicts normal, mild, and severe growth restricted offspring). Identifying the exact organ responsible for death is therefore somewhat intractable since the causes and consequences are interconnected, and multiple biological failures can influence the development process leading to death. Indeed, we have performed transcriptome analysis of F1 liver, brain and brown adipose tissue, all of which exhibit widespread gene mis-expression prior to death. Thus overall, death is associated with multiple interconnected organ complications that stem from the growth phenotype that we trace back to sub-optimal *in utero* development. With respect to understanding the placental phenotype of SGR offspring, this is indeed an intriguing question. However, because dams eat their offspring’s placentas immediately after birth, it is impossible to correlate SGR (which progressively emerges) with placenta phenotypic characteristics.

C3: Could it be that the entire postnatal phenotype is related to paternal-effect imprinted genes? The author should provide more data specifically on paternally-expressed imprinted genes in sperms.

R3: This is an important question, and indeed postnatal growth phenotypes are linked with genomic imprinting abnormalities. Sperm are transcriptionally quiescent but carry imprinted marks to offspring in the form of DNA methylation at gametic DMRs. To test these, we performed quintuplicate independent replicates of whole genome bisulfite sequencing (WGBS) on purified male sperm. Analysis showed that the correct DNA methylation imprints were established at all paternal and maternal gametic imprinted loci in nABX sperm (*see Extended data Fig 14E*). This suggests no epigenetic defects in transmitted imprinted genes. We have now further also analyzed imprinted gene expression in testis and mature sperm from nABX exposed fathers, finding no preferential changes in this gene class. We do however observe significant mis-expression of imprinted genes such as *Meg3*, and imprinted regulatory such as *Zfp57*, in placentae derived from nABX fathers (*e.g. see Fig 4E*). This suggests changes in imprinted gene networks could ultimately contribute to the F1 phenotype by affecting placental development. However, given widespread molecular and physiological changes in placenta, and that we do not find imprinted defects in sperm, it is not possible to distinguish whether imprinted mis-expression in placenta is a cause or consequence of the wider defects.

C4: Is it possible that the upregulation of tRF-Gly-GCC or the 5’ tRNA fragments in mature sperms is related to the postnatal phenotype?

R4: The change in tRF-Gly-GCC abundance could indeed contribute to the postnatal phenotype, and it has been previously implicated in intergenerational effects. For example, introduction of tRF-Gly-GCC into embryos caused changes in genes and transposable elements expressed at the 2-cell stage of development (Sharma et al., 2016). However, it is important to note that we observed complex responses across molecular layers in the testes/sperm, which suggests tRF-Gly-GCC may not be the sole factor influencing the postnatal phenotype. To this end we know that hormones and metabolic profiles were significantly altered in the testis (*see Fig 3E-F*), leptin level was markedly reduced (*see Fig 3H*), genes were mis-expressed throughout the spermatogenic process (*see new Extended data Fig 11 – scRNAseq in testis – & Fig 3G*), while many other tRNAs and miRNAs were also dysregulated in mature sperm (*see Fig 4B*). We anticipate that several other molecular changes hitherto not assayed also occur, for example in protein and RNA modifications, in chromatin, or on the sperm cell surface. Because regulatory layers interact, we suggest that a single paternal factor change may not account for

F1 phenotypes, but rather that compound changes may be more relevant. We now explicitly point out in the conclusion that the full mechanistic basis of gametic transmission requires further study. Taken together, we report complex reproductive molecular responses to gut microbiome perturbation, which complicates the causal relationship between paternal factors and postnatal effects.

C5: Leptin was clearly reduced in the nABX male testis, but what evidence ties it to the transgenerational effect?

R5: Thanks for asking this question, it has helped us to uncover a significant and previously unknown role of paternal leptin in preimplantation embryo development (*i.e.* an intergenerational effect). As noted by the reviewer, we intriguingly observed that nABX-mediated dysbiosis induced a reduction in leptin levels. This is true in both general circulation and specifically in testis. Leptin has a well-established role in energy balance and reproductive function. Indeed, *Leptin* knockout mice (*ob/ob*) are reported to be infertile and lack germ cells in mature males. To understand the role of the severely reduced leptin in our nABX system, we utilised *ob/ob* males at a young age (6 weeks), which mimics the time duration in which leptin is perturbed upon nABX exposure. Whilst, we did not obtain any progeny through natural mating, analysis of testis revealed the presence of mature germ cells at this age. Indeed, the testis architecture of *ob/ob* mice at 6 weeks was highly comparable to that after 6wk nABX treatment (*see new Fig 3I*). Moreover, the reduction in testis weight was strikingly similar between young *ob/ob* mice and nABX treatment (*see new Extended data Fig 13A-D*), thereby suggesting short-term perturbation to leptin closely phenocopies some of the testicular responses to gut microbial dysbiosis (which induces leptin reduction).

Given mature sperm were present in young *ob/ob* males, we therefore asked whether an intergenerational phenotype was induced through leptin deficiency. Natural mating failed to produce progeny, possibly owing to behavioral abnormalities in *ob/ob* males or inviable sperm. We reasoned that IVF would test the viability of sperm *per se*, and thus distinguish these. We used fresh sperm collected in parallel from wild-type and *ob/ob* males (littermates from heterozygous crosses), to generate embryos using isogenic oocytes obtained from wild-type females. The rate of IVF was comparable between *ob/ob* and WT males, supporting the conclusion that short-term loss of leptin does not affect fertility potential of sperm (similarly to nABX). To understand whether such sperm carry a signal that impacts offspring however, we profiled blastocysts generated from leptin-deficient or control fathers (*see new Fig 3J & Extended data Fig 13* – partly shown right). Strikingly, embryos from fathers with leptin perturbation clearly clustered away from control based on transcriptome at E4.5. Pathway analyses showed that most DE genes are involved in chromatin organization, histone modification, mRNA metabolism, and other pathways related to embryonic development. Note that whilst offspring from *ob/ob* fathers are genetically heterozygous, leptin is not expressed at any stage during early development thereby arguing against any potential hypomorphic impact of heterozygosity. This data indicate that a legacy of paternal leptin deficiency manifests in offspring development as early as the blastocyst stage. Taken together, we find that leptin levels in the father influence

offspring gene expression phenotype, and that the short-term absence of leptin phenocopies several aspects of the reproductive response to acute gut microbiome perturbation.

C6: The structural, epigenetic, transcriptional, and metabolic changes in the testis are fascinating. Which one of them explains the placental changes and the postnatal phenomenon?

R6: As noted, we report that testis exhibit transcriptional, epigenetic, and particularly metabolic changes in response to induced dysbiosis. This observation underpins one central conclusion that a ‘gut-germline’ axis operates in mammals, and influences testicular parameters. Further, it is an important and highly valid question to ask which change explains the downstream postnatal phenomenon. However, because paternal driving factors come from diverse sources (metabolites, leptin, misexpression of germline genes and small RNAs) originating both from the testis and matured sperm, essentially one or more of these combinatorial changes may contribute to the spermatogenic and sperm defects, and may interact with one another. Hence, to dissect mechanisms and enable rigorous conclusions of such complex responses is likely to require at least 3-5 years of *in vivo* experiments. As such, we have actively avoided making claims about precise mechanisms. With this in mind, it should be noted that one of the most compelling findings of this study is that pre-conception paternal dysbiosis triggers cumulative paternal risk factors that induce placental insufficiency, a single pathophysiological readout. Here we presented comprehensive lines of evidence linking poor pregnancy outcomes through the gut-germline-placenta axis, contributing new insights into reproductive health, antibiotic use, and diseases of the placenta, as well as emphasizing the importance of non-genetic intergenerational inheritance more generally.

C7: The phenotype seems to localize primarily to a postnatal effect (P3 to P15-18). The authors indicate that nABX fathers had normal fertility and normal litter size. These highly suggest intact placental function.

R7: This is not always the case, having normal fertility (i.e. being pregnant) and normal litter size (i.e. number of pups born), does not mean that no anomalies exist, but rather is fully compatible with stillbirth, low birth weight, neonatal death, growth retardation, and premature death amongst other adverse outcomes. In numerous epidemiological studies and a report by the World Health Organization, placental insufficiency has been linked to increased neonatal mortality, morbidity, and an altered postnatal growth trajectory. Hence, the postnatal effects (neonates to P18) reported in our findings are indeed expected outcomes of placental insufficiency, and interestingly, we captured many of the phenotypic anomalies reported in previous studies within our datasets. Furthermore, offspring that survive after P18 often undergo catch-up growth phenotypes, a phenomenon also observed in a number of animal and human epidemiological studies related to placental insufficiency. Thus, the assumption that normal fertility and normal litter size are indicators of intact placental function is at odds with what is known in the field related to placental insufficiency and adverse pregnancy outcomes. It is important to note that our findings were not confined to postnatal phenotypic anomalies, as we demonstrated that placental insufficiency is manifested at molecular, histological, and phenotypic levels of conceptus development from midgestation (E13.5) to late pregnancy (E18.5). The revised manuscript includes extensive new histological data further quantifying this pre-natal effect (see new Fig S17A-D & Fig 4I).

C8: To establish a pregnancy/placental phenotype, irrespective of a mechanism, evidence of fetal weight restriction or death must be provided during the fetal period (E11-E18).

R8: We have partially addressed this comment in the above response. Intrauterine growth restriction or death of the fetus are two of the possible adverse pregnancy outcomes that happen due to placental dysfunction, mainly due to *severe* placental insufficiency, where a near-complete or complete failure to deliver oxygen and nutrients occurs. In general, the adverse consequences of placental insufficiency on

fetal and offspring morbidity and mortality depend on the time of onset, severity, and gestational age at birth. In addition, it is important to emphasize that we did not claim *in utero* fetal weight restriction or death; rather, our conclusions are based on physiological defects in placenta (size, development), extensive transcriptomic dysregulation, and placental growth factor (PGF) secretion, which indicate impaired placental function both at mid and late gestations. It should also be noted that PGF is a well-established biomarker of placental dysfunction, a key player in angiogenesis and vasculogenesis during pregnancy, and is significantly downregulated irrespective of the antibiotic cocktails (nABX or avaABX) we used to induce paternal gut dysbiosis. To further investigate the placental phenotype we have now performed additional analyses on new independent placental cohorts from dysbiotic or control fathers. We find significantly decreased vascularisation and increased placental infarctions at both E13.5 and E18.5 (see new Fig S17A-D). Each of these phenotypes is consistent with increased risk of placental insufficiency induced by paternal microbiome condition at the time of conception.

C8: Increased fetal/placental ratio (Fig 4G), if anything, suggests a more efficient placenta, which was able to support normal fetal development despite its smaller size.

R8: It is not always the case, both increased and decreased (i.e. disproportionate) fetal-placental weight ratios are associated with risks of adverse pregnancy outcomes. There is however substantial variation in the clinical implications of fetoplacental ratios (some authors use placenta-fetal ratios). However, one can generally conclude from these that marginally disproportionate ratios correlate strongly with various fetomaternal complications, such as preeclampsia, intrauterine growth restriction, fetal loss, preterm delivery, low birth weight, neonatal mortality, postnatal developmental delay, gestational diabetes mellitus, maternal hypertension and mortality (Matsuda et al., 2018). A large retrospective study (n = 534 892) from the Medical Birth Registry of Norway found that both increased and decreased placenta-fetal ratios were associated with adverse pregnancy outcomes (Haavaldsen et al., 2012). While other studies reported that a high fetoplacental ratio (probably due to a small placenta) is associated with maternal hypertension, whereas a low fetoplacental ratio (probably due to a large placenta) may indicate underlying chronic conditions, such as anemia or malnutrition, that require placental overgrowth (Naeye RL., 1987; Molteni et al., 1978; Robertson et al., 2002; Fox et al., 1991.; Eriksson et al., 2000., Panti et al., 2012; Matsuda et al., 2018). Other studies also reported higher fetoplacental ratios in preeclamptic/eclamptic and gestational diabetes mellitus cases, whilst suggesting a smaller placenta is strongly associated with preeclampsia (Kim et al., 2014; Sharma et al., 2016; Ezeigwe et al., 2018; Phadungkiatwattana and Puttasiri, 2019). On the basis of these various reports, we can therefore argue that adverse pregnancy outcomes could occur when the fetal-placental weight ratios are disproportionate; however, the clinical implications should be cautiously interpreted, as placental insufficiency outcomes may vary depending on several risk factors.

C9: Lastly, invoking preeclampsia on the basis of gene expression changes as the mechanism for the postnatal growth changes seems somewhat superficial. The placenta might respond to some paternally induced effects, but how does that explain the mostly postnatal phenotype?

R9: We fully take on board the reviewer's concern, and have toned down the implication of preeclampsia, instead stating that our placental insufficiency phenotype exhibits hallmarks of preeclampsia. Indeed, preeclampsia is one possible adverse consequence (symptom) of placental inefficiency, but not the primary cause. Thus, the major mechanism driving adverse pregnancy outcomes, including the expression of preeclampsia-associated genes, is placental insufficiency, in which oxygen and nutrients are not adequately supplied to the fetus, thereby causing pregnancy complications including preeclampsia. The nABX F1 offspring are therefore subject to serious prenatal and postnatal morbidities and mortality, likely due to placental insufficiency. Preeclampsia is one possible manifestation of this, which we raise. Moreover, our conclusion that the phenotype shows 'hallmarks of preeclampsia' was based not solely on gene expression, but also on the physiology and direct secretion of PLGF (also known as PGF), the prime diagnostic marker of preeclampsia (Chau et al., 2017; Duhig et al., 2019).

Minor Comments:

C10: Figure S3D title - should be “distribution”. The legend to S5B – should be ”frequency”. Lines 119/120 should be “correlate”

Line 65; should mention brains of SDD offspring. Are figures 1H and S4D the same? Needed twice?

Line 168 refers to Fig S11-E-F, but there is no S11F. Line 189 and line 200 refer to Fig S13C, not S12C, as in the text.

Sometimes the authors use PIGF and sometimes PGF

R10: Thanks for pointing out these errors, we have now corrected them or made consistent as suggested, for example referring to ‘PIGF’.

Referee #4 (Remarks to the Author):

C1: In this study, Argaw-Denboba et al. report that disruption of the microbiome of male mice has intriguing effects on their offspring. First, using different strategies, including ad lib non-absorbable antibiotics (nABX), a combination of other antibiotics (avaABX), and osmotic laxatives, the authors found that microbiota perturbation of male mice causes growth retardation and increased postnatal mortality of their F1 offspring. These phenotypic defects were rescued when the microbiota perturbation was removed for 8 weeks, indicating that it is a reversible defect. RNA-seq analysis revealed that expression of genes associated with metabolic processes are misregulated by nABX. Through performing in vitro fertilization using sperm from nABX-treated mice, the authors obtained evidence that the F1 phenotypes are transmitted via the sperm. nABX treatment led to smaller testes, lower sperm count, as well as morphological changes in testes seminiferous tubules, indicative of defects in spermatogenesis, not just sperm. The authors also performed unbiased metabolomic profiling and found that nABX treatment caused significant alterations in the levels of some metabolites in testes, raising the possibility that one or more of these changes is responsible for the spermatogenic and sperm defects. To begin to understand mechanism, the authors performed RNA-seq analysis of whole testes from the nABX-treated fathers. Among the most downregulated genes (~2-fold) was Leptin, a gene encoding a hormone associated with germ cell development and androgen signaling. The authors also performed small RNA-seq analysis of sperm, which revealed that multiple microRNAs are dysregulated by nABX. Finally, to further investigate potential mechanism, they performed RNA-seq analysis on the brain of the offspring (at E13.5) and placenta of the mothers pregnant with the F1 offspring. While they found little effects on gene expression in the embryonic brain, they identified many differentially expressed genes in the placenta. Interestingly, they observed downregulation of multiple Prolactin genes and other genes essential for placental development (e.g., Hand1 and Syna). This raised the possibility that nABX treatment causes placental defects, which they directly demonstrated (altered F1 fetus/placenta weight ratio and reduced labyrinth zone).

This study provides compelling evidence for the origins of health and disease hypothesis: paternal stresses influence germline and the offspring. Thus, this work joins other key studies showing that offspring are influenced by paternal stresses, including low-protein diet (PMID: 30150380) and disruption of circadian rhythm (PMID: 34039610). Overall, this study is well designed and makes some interesting observations regarding how the paternal microbiome influences the male germline and offspring. However, there are several concerns that need to be addressed.

R1: We greatly appreciate the reviewer’s time and careful assessment of our study, which “*provides compelling evidence for the origins of health and disease hypothesis*”. Your comments have helped us to understand more about the cell-type specific responses of the male reproductive system to gut dysbiosis, amongst other areas. Below we provide responses that fully address the outstanding concerns.

Major Concerns

C2: Statistics. While most panels are adequate in this regard, in some panels, the number of samples and/or statistical testing are not described clearly; e.g., Fig. 1B, 1E, 2E, 2I, S1B, S1E, S1F, S6B, S7D, S9B, S10A, and others.

R2: Thanks for pointing out these oversights. We have reviewed the entire manuscript and figures, sample sizes, and statistical tests where applied are described in the revised figure legends or the figure panel itself, in addition to text.

C3: Cell subset changes vs. gene expression changes in the testes. The authors overinterpret their RNA-seq data from whole testes. The testis contains both somatic and germ cells, both of which can be further divided into many different cell populations. Thus, when the authors observed changes in the “expression” of genes, this will often not be due to an alteration in expression per cell, but rather a change in the proportion of cells that express that gene. Considering the authors’ finding that nABX influences the cell subset composition in the testis, it seems likely that many of the differentially expressed genes they identified are merely reflecting changes in the proportion of cell subsets. For example, *Leptin* was found to be downregulated ~2-fold after nABX treatment. However, the 2-fold reduction in its level may not be because of reduced expression per cell but rather because of reduced number of leptin-producing cells (spermatogonia and/or spermatocytes). To address this and other claims by the authors with regard to key RNA-seq results, they should perform: FACS analysis of the relevant germ cell population (e.g., spermatogonia or spermatocytes) to test if there are changes in their frequency after nABX treatment. FACS-purify the relevant cell population and examine the expression of the gene in question (e.g., *Leptin*) by qPCR or similar method.

R3: We agree fully with the reviewer that changes in gene expression in testes may reflect either transcriptional changes, cell composition changes or a combination of both. To distinguish these questions, we have now performed an entirely new set of 10X single cell RNA-seq experiments (scRNAseq) from testes of control and nABX exposed males (pooled n=4 from each; ~3,000 cells after QC). This novel scRNAseq dataset indicates that dysbiosis in the gut contributes both to gene mis-expression (>100 DEGs identified) and a slight proportional reduction of post-meiotic germ cells (colored clusters shown right), particularly at the elongating spermatids stage (*see new Extended data Fig 11A-F – partly shown right*). This suggests both a molecular *and* cell composition response, further supporting a cross-talk between gut microbiota and germline homeostasis/development.

Specifically for the expression of *Leptin*, please first note that it is downregulated by a \log_2 value >-3 (i.e. a >8-fold reduction) in our bulk transcriptomics data from nABX testes (*see Fig 3G*). This trend was confirmed by independently assessing the level of leptin hormone present specifically in testis, as well as systemically in plasma, in response to nABX (*see Fig 3H*).

However, due to the limitations of single-cell transcriptomics, which cannot detect lowly expressed genes such as *Leptin*, we could not capture its expression at single-cell resolution in testes from 10X data (as expected). Hence, to determine the specific cells that express *Leptin*, we have now used *in situ* sequencing (ISS) that detects spatial expression at single-molecule resolution (shown right). This demonstrated that *Leptin* is expressed predominantly in early stage germ cells, as previously published. Given the downregulation in leptin levels in testes are far greater at both RNA and protein level than any cell proportion changes detected following nABX, this collectively indicates *leptin* is transcriptionally downregulated in germ cells.

C4: Cell subset changes vs. gene expression differences in the placenta. The same concept described above is relevant to the authors conclusions about gene expression in placenta. Thus, they should take the same steps as above to distinguish between gene expression vs. cell subset alterations. Given that the authors show nABX-treated mice have reduced labyrinth zone area, an obvious possibility is that many of the differentially expressed genes they identified results from this, not altered expression per cell.

R4: The objective of RNAseq analysis in placenta is to act as a phenotyping strategy to detect whether there are reproducible differences between placentae derived from control of nABX-exposed fathers. With this in mind, whether the effect stems from transcriptional changes or from an altered subset of cells (or most likely both) therefore does not impact the phenotypic conclusions we draw. Specifically, that gene expression differences are consistent with a paternally-induced placental insufficiency effect, leading to pre/postnatal complications. A comprehensive analysis of the aetiology of the placental defect(s) in our model, which would include subset changes vs. gene expression differences, requires time-resolved multi-parametric readouts that are beyond the scope of the current manuscript. However, we now have produced an extensive independent and additional analysis of placenta from nABX fathers in the revised manuscript, identifying decreased vascularization and increased placental infarction (*see new Extended data Fig 17*). These data further support our histological evidence (*Fig 4I*), clinical marker ELISA assays (*Fig 4J*), and RNAseq results at mid- and late gestation (*Fig 4C-D*), that the source of postnatal fetal effects can be traced back to a paternal influence on placenta development.

C5: It is claimed that the authors uncovered the underlying mechanism for why paternal microbiome impairment causes defects in the offspring (e.g., through reduced leptin signaling), but actually only correlative (no rescue) data was provided. At the least, the Abstract should be revised with this in mind. Optimally, it would be desirable if the authors could perform experiments to directly address mechanism; e.g., rescue experiments. For example, the authors could ask whether Leptin depletion causes some of the phenotypic defects caused by microbiome disruption.

R5: We fully agree that this is important and requires revision. We have taken several courses of action. (1) First, taking on board the reviewer's useful suggestion to examine leptin's functional role, we investigated "*whether Leptin depletion causes some of the phenotypic defects caused by microbiome disruption*", starting with the direct effects on the father. We utilised Leptin-deficient *ob/ob* males at a young age (6 weeks), which mimics the time duration in which *Leptin* is perturbed upon nABX exposure in our model (i.e. 6 weeks). Of note, as *ob/ob* mature, they become progressively sterile and lose germ cells, establishing a direct link between leptin and reproductive fitness (Moschos et al., 2002). However, after only 6 weeks of leptin depletion, histological analysis of testis revealed the presence of mature germ cells. Indeed, the testis architecture of *ob/ob* mice at 6 weeks of age was highly comparable to that after 6wk nABX treatment, with formation of vacuoles and partial loss of germ cells (*see new Fig 3I & Extended data 13A-D*). Moreover, we observed a reduction in testis weight in young *ob/ob* mice that was similar to weight reduction after 6 week nABX treatment (*see Fig 3I*). This suggests that short-term (6wk) absence of leptin phenocopies several aspects of the testicular responses as gut microbial dysbiosis (which itself induces highly significant leptin reduction).

(2) Second, despite infertility at later (older) stages, we found apparently normal mature sperm in young (6wk) *ob/ob* males, and therefore could ask whether an intergenerational phenotype is induced through acute paternal leptin deficiency. Natural mating failed to produce progeny, possibly owing to behavior abnormalities in *ob/ob* males or to embryonic lethality of offspring. However, we reasoned that IVF would directly test the viability of sperm *per se*. We used fresh sperm collected in parallel from wild-type and leptin-deficient *ob/ob* males (same genetic background; littermates), and used these to fertilise isogenic oocytes obtained from wild-type females. The rate of IVF was comparable between *ob/ob* and WT males, supporting the conclusion that short-term loss of leptin does not affect fertility potential of sperm (similarly to nABX). To understand whether such sperm carry a signal that impacts offspring, we profiled IVF blastocysts (see also R5 to reviewer #3). Strikingly, embryos from fathers with leptin perturbation clearly clustered away from control based on transcriptome at E4.5 and E5.5 (see new Fig 3J & Extended data Fig 13E-F). Pathway analyses showed that most significant differentially expressed genes are involved in chromatin organization, histone modification, mRNA metabolism, and other pathways related to embryonic development (see new Extended data Fig 13H-J). Note that whilst offspring from *ob/ob* fathers are genetically heterozygous, *Leptin* is not expressed at any stage during early development (see new Extended data Fig 13G), thereby arguing against any potential hypomorphic impact of heterozygosity. These data indicate that paternal leptin deficiency manifests as changes in offspring development/transcriptome as early as the blastocyst stage.

Taken together, we find that leptin levels in the father influence offspring gene expression programmes, and that a short-term absence of leptin at least partially recapitulates the testicular response in males induced by gut microbiome perturbation. These new data are included in the revised manuscript, wherein we have also revised the abstract to better reflect the role of leptin. As leptin plays a pleiotropic role in several pathophysiological conditions, we believe our new data on paternal effects of leptin deficiency open up a new line of research, particularly one aimed at identifying the developmental origins of chronic diseases such as obesity and diabetes.

Minor Concerns

C6: RNA-seq validation. The authors should validate key dysregulated genes/microRNAs by RNA scope and/or TaqMan analysis. This also applies to RNA-seq analysis of placenta.

R6: We validated key RNA-seq results such as *Leptin* using ELISA, as these orthogonal strategies provide a more reliable biological context for the effect, and indeed confirmed our RNA-seq. Moreover, we designed each experiment to utilise sufficient independent replicates ($n > 5$), from independent litters and cages, to draw robust conclusions from the data. Note that RNA-seq based analyses are a mature technology routinely considered as technically equivalent to qPCR/TaqMan based approaches for obtaining statistical differences in gene expression. It is evident that our next step is to analyse and dissect the molecular role played by mis-expressed genes/microRNAs in the developmental process. As part of our ongoing research, we aim to dissect paternal driving factors (metabolites, leptin, mis-expression of germline genes and small RNAs) and validate the phenotypic consequences at the molecular level.

C7: Some detailed experimental procedures need to be provided. For example, for the mouse mating. Are these mice mated within one day time window? Details about samples harvest, such as testes, placenta, brain, and brown adipose tissue.

R7: We have reviewed the methods and believe the information requested has to a large extent been described. For example, the following paragraph relates to mouse mating: “*To generate F1 offspring, each CON or nABX a male mouse was placed with a single 6-week old treatment-naïve virgin female mouse. During the mating period (~1-4 days), mice were housed under standard optimised environmental conditions with ad libitum access to regular diet and water. Each morning females were checked for vaginal status, all females with positive signs of mating (vaginal plugs) were removed from*

the male cage and housed into new cage, while all females that were not vaginal plug positive were returned to the male cages in the afternoon and examined in each successive morning. To minimize microbial transfers between the pairs housed together, the mating hours of the pairs was restricted from late afternoon to early morning (3PM-9AM) throughout the mating period.” To directly answer the reviewer question, the majority of matings (>50%) occurred within 1 day, but some took as long as 4 days (evening co-housing only). In further response, we have attempted to improve the methods for clarity by adding many additional descriptions of sample harvesting (e.g. for placenta and testes) and re-considered all entries.

C8: Fig. S1D. Do all mice survive nABX treatment?

R8: Fig S1D (now renamed Extended data Fig 1D) shows that only three deaths occurred out of 72 treated male mice, while one death occurred out of 70 control mice. Hence, the treatment does not pose a significant death risk ($p=0.33$), as expected for FDA-approved antibiotics administered at a human-equivalent dose.

C9: Fig. 2G. The PCA assay shows that samples from nABX 6wk+4rec are more distinct with Controls than those with nABX 6wk mice. Any explanations why 4wk recovery make them “worse”?

R9: Thank you for raising this question. Spermatogenesis is a continuous multi-stage process, which in mice takes ~ 35 days to produce spermatozoa from spermatogonial stem cells, and spreads asynchronously along the seminiferous tubules. As evidenced by the 16S rRNAseq result in Fig 1B, nABX 6wk+4rec males are still dysbiotic and have only partially recovered. The 6wk+4rec mice therefore have a continued impact on their germ cells for a longer period (~10 weeks) despite the discontinuation of treatment. Consequently, nABX 6wk+4rec males accumulate more affected sperm than nABX 6wk sires due to continuous and longer cycles of sperm production and only a partial recovery of dysbiosis. Thus, one can expect a ‘worse’ effect in nABX 6wk+4rec offspring than nABX 6wk offspring. Indeed, analysis of offspring phenotypes indicate a higher risk (odds-ratio) for severe F1 phenotypes derived from 6wk+4rec fathers relative to 6wk fathers (*see* Fig 2E).

C10: Line 38. The format of ref 25 is wrong.

R10: Thank you for pointing this out, we have corrected it.

C11: Line 65. The authors performed “transcriptome profiling” (a preferable term is RNA-seq analysis), not “transcriptional profiling.” The latter implies that transcription rate is measured, which is not correct. RNA-seq analysis measures steady-state RNA levels, which is a reflection of both RNA synthesis (transcription) and RNA decay (turnover).

R11: Thanks for pointing this out. The reviewer is absolutely correct and "transcriptome profiling" is now used in place of "transcriptional profiling".

C12: Line 128. It is claimed that the vacuoles observed in the testes of nABX-treated males is due to germ cell loss, but no evidence is provided for this.

R12: The evidence for this is based on the histological analysis, illustrated in Fig 3B (and Extended data Fig 10), in the middle panel (nABX) where vacuoles are seen to form in the testis. Upon horizontally sectioning the testis, spermatogenic stages are clearly visible along the seminiferous tubule. In the basal compartment, spermatogonia and preleptotene spermatocytes reside, while in the adluminal compartment, round spermatids and elongating/elongated spermatids reside. On the basis of these

testicular features, germ cells appear to be the most adversely affected, since the vacuoles form where they should have been present. Absence of specific germ cell compartments is further shown in Extended data Fig 10. Note, in young males such vacuoles are *extremely* rare, since spermatogenesis is a highly regulated stereotypical process. Moreover, we have also performed scRNAseq analysis in the revised manuscript. This analysis confirmed our previous bulk RNAseq analysis that dysbiosis affected the spermatogenesis process, resulting in a decrease in late stage germ cells (see new Extended data Fig 11).

C13: Line 130. Related to Major Concern point 2, above, it is critical to perform detailed histological analysis to determine the nature of the abnormalities in the testes of nABX-treated males; e.g., quantitative analysis of SPG, SPC, and STs.

R13: This question is addressed in R3, where single-cell RNA analysis was carried out using the 10x Chromium Single Cell 3' Reagent Kits. In agreement with the results of the histological and RNAseq analyses, our scRNAseq analysis confirmed paternal dysbiosis contributes to both misexpression of stage specific marker genes and proportionate reductions of germ cells, notably at the stage of elongated spermatids.

C14: Line 149. The reduction in SPC markers and increase in somatic cell markers is consistent with fewer SPCs and subsequent germ cell stages, leading to a relative enrichment in testicular somatic cells. This should be tested directly, as indicated in Major Concern, point 2, above.

R14: We believe that the question is adequately addressed in R3. As we stated, using scRNAseq we were able to capture the gradual reduction of germ cells that is evident at the elongated spermatid stage, and a corresponding proportionate increase in somatic cells.

C15: The used marker lists of different cell types need to be provided.

R15: Following the suggestion, the markers are derived from unbiased cluster analysis published in 'A Comprehensive Roadmap of Murine Spermatogenesis Defined by Single-Cell RNA-Seq' (Green et al., 2018). This is indicated in the methods.

C16: Line 153. Leptin is not only expressed in spermatogonia, but spermatocytes (PMID: 17935159).

R16: Thank you for bringing this to our attention. We have updated the manuscript accordingly to include this reference.

References

- Chau, K., Hennessy, A., and Makris, A. (2017). Placental growth factor and pre-eclampsia. *J Hum Hypertens* 31, 782-786. 10.1038/jhh.2017.61.
- Duhig, K.E., Myers, J., Seed, P.T., Sparkes, J., Lowe, J., Hunter, R.M., Shennan, A.H., Chappell, L.C., and group, P.t. (2019). Placental growth factor testing to assess women with suspected pre-eclampsia: a multicentre, pragmatic, stepped-wedge cluster-randomised controlled trial. *Lancet* 393, 1807-1818. 10.1016/S0140-6736(18)33212-4.
- Green, C.D., Ma, Q., Manske, G.L., Shami, A.N., Zheng, X., Marini, S., Moritz, L., Sultan, C., Gurczynski, S.J., Moore, B.B., et al. (2018). A Comprehensive Roadmap of Murine

Spermatogenesis Defined by Single-Cell RNA-Seq. *Dev Cell* 46, 651-667 e610. 10.1016/j.devcel.2018.07.025.

Moschos, S., Chan, J.L., and Mantzoros, C.S. (2002). Leptin and reproduction: a review. *Fertil Steril* 77, 433-444. 10.1016/s0015-0282(01)03010-2.

Sharma, U., Conine, C.C., Shea, J.M., Boskovic, A., Derr, A.G., Bing, X.Y., Belleannee, C., Kucukural, A., Serra, R.W., Sun, F., et al. (2016). Biogenesis and function of tRNA fragments during sperm maturation and fertilization in mammals. *Science* 351, 391-396. 10.1126/science.aad6780.

Reviewer Reports on the First Revision:

Referees' comments:

Referee #1 (Remarks to the Author):

The work reported here linking the gut microbiome with offspring phenotypes through the paternal germline is a discovery of the first-order. The interplay between host physiology with gut microbiome is emerging as recently published work documents (e.g. exercise, dopamine metabolism, and the microbiome). Similarly, work published over the last several years showing transmission of non-genetic environmentally-induced information across generations is revolutionizing the ways we understand inheritance. The work reported in this manuscript connects these fundamental discoveries and extends the work in exciting and unexpected directions, identifying a novel dimension of information transfer at least from one generation to the next.

An already exceptional manuscript has been significantly enhanced with substantial new evidence. In particular, the response to Reviewers Comments was outstanding with many substantial experiments conducted to address questions that were raised – seven major tests listed on page 1 of the Response. The breadth of these experiments and the rigor of the work nicely validate previous Results, break new ground (intergenerational leptin effect for example), and clarify observations and proposed mechanisms. While one might quibble about additional studies and controls that could be pursued, the evidence in hand tells a compelling story about intergenerational phenotypic effects that gut microbiota trigger in the paternal germline. The evidence is as complete and convincing as any body of work I've seen in a long time. Rigorous work, thorough analysis, and thoughtful interpretation of results and limitations pervades every aspect of the manuscript. The statistical analysis and presentation is about as good as I've seen in any publication, e.g. attention to skew of data, use of nested t-tests. Use of recent technical developments is impressive, e.g. in situ single-molecule sequencing.

Two comments and a question/comment:

1. Comment - Review #4. Cause of death. Remarkably, this is rarely known, even in humans. The condition at death can be documented, but which if any recognized pathology 'caused' death is usually unclear. The authors are correct that the pathological state can be documented, but the cause of death rarely identified.
2. Comment in several places in the Review Response - vacuoles in the testis with germ cell deficiency. This histology is very common in mouse mutants and environmental exposures. It seems to be a general feature rather than diagnostic for a particular defect or exposure.
3. Question/comment. The tests for transgenerational effects to the F2 generation assume that the genetic background is permissive, i.e. if there is transgenerational inheritance, the background would show the epigenetically inherited phenotype. Interestingly, some genetic backgrounds are permissive and some not (see some early papers by Mike Skinner). The only way to know in this model system is to test several genetic backgrounds, a question and task that goes far beyond the scope of the present work. All that needs to be done here is to insert a phrase at the appropriately place saying that the test assumes that the genetic background is permissive. The observed phenotypes found in F1 mice could simply reflect direct paternal effects rather than true epigenetic inheritance.

Referee #2 (Remarks to the Author):

The authors of the manuscript have re-submitted a revised version of their study, investigating the interdependence between paternal pre-mating environment, father's gut microbiota, and intergenerational health using mice as a model system. The authors propose that non-absorbable

antibiotics (and now osmotic laxatives) are capable of perturbing the paternal gut microbiota, leading to dysbiosis that increases the risk of placental insufficiency and low birth weight of offspring among other detrimental effects that affect the next generation's fitness. Thus, authors suggest that the environment-dependent "gut-germline axis" in males has the capacity to determine the health of F1 offspring via altered sperm quality of the father.

Authors have responded to each of the reviewers' comments extensively, carrying out additional experiments to clarify their findings. Goals were to further investigate co-housing and its potential effects on microbiota transmission, testing the direct and indirect systemic effects of nABX, assaying different forms of gut microbiota perturbations and recovery effect, and assessing whether the father's "gut-germline axis" also affect F2 offspring (which it does not). The amount of new work done is very impressive! However, the manuscript still falls short in addressing the fundamental questions, including some weaknesses not discovered previously.

Specific comments:

The authors have clearly shown that exposure of paternal mice to nABX prior to mating affects:

1. The intestinal microbiome.
2. Important phenotypes in the male reproductive tract (However, the nABX were given to the males at 5 weeks of age, prior to sexual maturation, and this clearly had multiple effects on sexual development)
3. A number of phenotypes in the next generation offspring, but not in the F2 generation.

The mechanisms between 1 and 2 were still not completely defined, but there was progress. This work involved administering non-absorbable low-dose antibiotics to 5-week-old male mice and assessing the effect of the potentially perturbed microbiome on host molecular and physiological changes and offspring fitness. One important point that needs to be addressed is that age 5-weeks is the timepoint at which male mice are undergoing sexual maturity, particularly spermatid elongation (seen in Extended data Fig 11 D, elongated spermatids as the population most affected by nABX). The authors should include quantitation of viable sperm as a male fertility parameter in their natural mating and IVF experiments.

However, while 2 seems to lead to 3, the mechanism underlying this effect is still not fully resolved.

There are still 2 remaining possible explanations for the intergenerational phenomenon that have not been fully addressed.

First, could an altered sperm microbiome be carrying the phenotype? qPCR quantitates the total load only, not the composition. Was compositional analysis done? For specimens at which timing? There was discussion of this point, but more clarity is needed.

The IVF experiments are very convincing—that might be the simplest and clearest experimental setting to determine if sperm microbes have an effect, first by in-depth microbial sequence analysis, comparing sperm from nABX vs control. A second approach would be to treat the nABX (and Control) sperm with a bactericidal antibiotic that does not penetrate intact mammalian cells—e.g. gentamicin.

If there is no microbiome, the results should be unaffected, but the presence of a differential effect of the treatment would provide direct experimental evidence for a microbiota effect.

Second, is the effect occurring through the maternal microbiome after all? Specifically, the co-housing was for 7 hours a day only for 4 days, during a period of the daily cycle (9a-4p) when mice are not very active. This is not an adequate test of co-housing. In addition, after the 4-day co-housing (mating), at which time points were the mothers tested? Testing immediately after

may have been too soon to detect the effect of invasion by the dysbiotic male microbiota, if such occurred. Generally, the flow of microbes is asymmetric in co-housing with stronger flow from the dysbiotic to the normal than vice versa. It appears that the next testing time for the mothers was P15—that also could be too late to detect a transient (but crucial) effect. Such analyses could be carried out in the short-term without a long multi-generational effect.

Progressive loss after resting for 4 weeks, and more so after 8 weeks does not really address whether maternal or sperm transmission is important, only that there is recovery in the fathers over time.

In Extended Figure 8A are the top and bottom labels reversed? The Results should focus on the 6 +0 experiment only, with sufficient numbers to rule out subtle signals. What was the time point for this study? If there were multiple time points (6 = 0, 6 + 4, 6 + 8), they should each be shown separately.

When proposing a new mechanism, the burden of the proof is for the proposer to consider all possible ways in which the conventional explanation can be ruled out.

Extended Figure 8B includes the 6+4 and the 6+ 8 offspring, which might dilute out any signal in the 6 + 0 offspring, and those should be eliminated from the Figure. Similar to 8A, what was the time point for this study? If there were multiple time points, they should each be shown separately.

Extended Figure 8C does not rule out sperm microbe transmission.

Considering Authors' Response to Comment 2:

Subresponse 1: See above the limitations of the author's response 1.

Subresponse 2: The short co-housing might not have been sufficient to change the maternal microbiome, and thus the lack of a change in phenotype would not be surprising. This is circumstantial evidence.

Subresponse 3: See the above comments about the limitations of interpreting the IVF results.

In addition, the avaABX cocktail includes ampicillin, which is well-absorbed—this does not meet the definition of nABX. Not clear why studies with this regimen should be included at all.

Considering Authors' Response to Comment 3:

Please see above comments about IVF.

Also, what was the percent of high quality sperm in the nABX and control groups?

Considering Authors' Response to Comment 5:

Appropriate

Considering Authors' Response to Comment 6:

Appropriate

Considering Authors' Response to Comment 7:

Appropriate

Considering Authors' Response to Comment 8:

Greater clarification and extension of the experimental results needed, as indicated above.

Considering Authors' Response to Comment 9:

Appropriate

Considering Authors' Response to Comment 10:

Appropriate

Considering Authors' Response to Comment 11:

Appropriate, agree that phenotypes are non-genetic.

Considering Authors' Response to Comment 12:

The suggestion to give the antibiotic courses for 6 weeks to germ-free mice, still stands. If there is no effect on the reproductive phenotypes, it can be concluded that a (perturbed) microbiome is required.

In addition, MS was done for the nABX regimen but not for the avaABX regimen that used ampicillin, which is the agent that is absorbable.

Considering Authors' Response to Comments 13-23:

All appropriate

Other Specific comments:

The authors have used a nested statistical analysis to compare the litters sired by individual fathers rather than individual pups to one another, which is the correct approach. However, for example, in Figure 1C, the data are presented as individual pups. Authors should consider adding a plot where the data is presented as litter averages, add S.E.M. to their figures, and specify whether the "severe growth restriction" individuals are from the same father. This applies to all figures where data are analysed in a nested way but presented as individual pup data points. Importantly, figure 1C appears to combine male and female pups—if this is correct, they should be separated and sex-specific analyses done. Same for 2H and 2I. Extended 3C, 5A, 7B, 8I, 9B,C—why combined?—there is plenty of room in Extended to show both sexes individually to determine if the effects were parallel, and significant

Authors discuss the role of leptin in intragenerational health and signalling in the "gut-germline axis". Did the authors test/consult previous literature for hypotheses about how exactly leptin is affected by gut dysbiosis and then consequently affects the germline? Moreover, which recovered first, father's gut microbiota or leptin levels?

In the transcriptomic profile of dysbiotic male testes, the authors found that the Leptin gene was strongly affected by nABX. Then they included ob/ob mice to study the functional role of Leptin in

their system. However, this experiment fails to have an appropriate control. In the IVF experiment, they should include sperm from 6wk ob/ob or WT that received nABX or not. If Leptin plays a central role, then the F1 offspring sired by ob/ob-nAbx father will show no phenotypic change compared to the ob/ob-control offspring. Additionally, the ob/ob mice the authors used in this study have germline mutation; as such, it's not surprising to see segregation in the PCoA plot of the F1 blastocyst transcriptome. To better understand the role of reduced levels of leptin in testes, the author should consider using a testes-specific ob/ob knockout.

Did the authors test whether the experimental mice were drinking equal amounts of nABX water? Microbiota perturbations generally are dosage dependent.

Authors have used rarefaction to analyse their 16S rRNA gene data. They should specify the average number of reads +/- SEM per sample in their data set prior to rarefying.

From the aspect of microbiota analysis, it would be interesting to see a PCA plot with all the father-mother-offspring data combined to see how they cluster, considering the control vs. nABX males and pre- and post-mating females. Authors should also consider showing which specific bacterial taxa (e.g., relative abundance plot) are present in control vs. nABX males.

Referee #3 (Remarks to the Author):

I thank the authors for thoroughly addressing my comments: The added data and figures clearly strengthen the most exciting male gut-germline association and the parental contributions to intrauterine and postnatal development. The mechanisms underlying the convincingly presented associations remain enigmatic. There are data that implicate the placental in the post-natal outcomes, again, without a mechanistic link. Implicating pre-eclampsia in the pathophysiological process, merely because of some shared placental analyte patterns, seems unjustified.

Referee #4 (Remarks to the Author):

In the revised manuscript, the authors addressed most of my concerns. However, there are some remaining issues/new ones, as summarized below.

1) The newly implemented scRNAseq analysis of mouse testes is critical for distinguishing cellular and molecular changes treated with nABX. However, many details of scRNAseq analysis are missing. The authors should provide more details, including: i) representativeness of their 4 samples from each group. For example, the authors should provide tsne plots for individual samples; also, the authors should provide the cell percentage of each annotated cell subsets from individuals. ii) more specific and more markers should be used to define these cell populations. The authors only used a single marker per each cell subset to define them (Extended data Fig 11B), and some of them are inappropriate. For example, *Dazl* is a pan spermatogonial marker, not specific for spermatogonial stem cells. Likewise, markers used for identifying spermatocytes and spermatids are not specific. iii) the authors used "logfc.threshold=0.25 and p_val_adj<0.1" as cut-off for identifying DEGs from scRNAseq. This is too flexible, especially considering the shallow sequence depth of 10X.

2) A following-up concern of scRNAseq analysis: is the post-meiotic germ cell number significantly reduced in the nABX group, as compared to the control group?

3) In the revised manuscript, the authors used in situ sequencing (ISS) to show the expression of Leptin in the wild-type testis. The authors could use ISS to compare Leptin expression between Control vs nABX at single cell level. This is useful for validating their bulk RNA-seq.

4) The reviewer does not agree with the author's argument (R6 to C6). While, yes, RNA-seq/small RNA-seq are mature techniques that are "technically equivalent" to qPCR/TaqMan, it IS still critical to validate key findings using independent techniques, especially considering that these authors

only provided correlative evidence.

5) The format of refs for M&M is wrong.

Author Rebuttals to First Revision:

Response to Reviewer's second round comments

Paternal microbiome perturbations impact offspring fitness

Argaw-Denboba et al

We would like to thank the reviewers for again contributing their time and thoughtful feedback on our revised manuscript. We are pleased that it was considered that “*an already exceptional manuscript has been significantly enhanced with substantial new evidence*”. Over the past 10 months we have undertaken several further lines of experiments that address the new comments raised and any outstanding ambiguities. We note that given the challenges of intergenerational scientific research with animal models, each experimental revision takes considerable time and resources to complete. Nevertheless, as a consequence of the reviewer feedback, the conclusions are strengthened further and we believe the study provides exceptional levels of replication and statistical rigour to support its core conclusions. As before, here we briefly summarise the major lines of revision in this version, and then respond (R) specifically to each comment (C) in full.

The major experimental additions can be summarised as:

1. **Microbiome characterisation.** We have generated new 16S profiling datasets of microbiomes from carefully collected ejaculated copulatory plugs (sperm plugs) to investigate the precise nature of microbial transmission to dams. We have also generated new supplementary figures using 16S data from hundreds of extant maternal and offspring samples to further rule out lingering confounders.
2. **Validation of small RNA.** We have performed new and independent dysbiosis experiments to validate the quantitative small RNA changes in sperm via orthogonal strategies.
3. **Leptin and nABX interaction.** We have undertaken an entirely new round of IVF experiments to test the interaction between paternal nABX and leptin perturbation(s), whilst further validating previous results.
4. **Germ cell metrics.** We have carefully scored physiological and viability features of mature sperm from dysbiotic males.
5. **Single-cell analysis.** We updated the analysis parameters of single cell data and performed in situ sequencing (ISS) for spatially-resolved *leptin* expression on new nABX testis.

Referee #1

C1: The work reported here linking the gut microbiome with offspring phenotypes through the paternal germline is a discovery of the first-order. The interplay between host physiology with gut microbiome is emerging as recently published work documents (e.g. exercise, dopamine metabolism, and the microbiome). Similarly, work published over the last several years showing transmission of non-genetic environmentally-induced information across generations is revolutionizing the ways we understand inheritance. The work reported in this manuscript connects these fundamental discoveries and extends the work in exciting and unexpected directions, identifying a novel dimension of information transfer at least from one generation to the next.

An already exceptional manuscript has been significantly enhanced with substantial new evidence. In particular, the response to Reviewers Comments was outstanding with many substantial experiments conducted to address questions that were raised – seven major tests listed on page 1 of the Response. The breadth of these experiments and the rigor of the work nicely validate previous Results, break new ground (intergenerational leptin effect for example), and clarify observations and proposed mechanisms. While one might quibble about additional studies and controls that could be pursued, the evidence in hand tells a compelling story about intergenerational phenotypic effects that gut microbiota trigger in the paternal germline. The evidence is as complete and convincing as any body of work I've seen in a long time. Rigorous work, thorough analysis, and thoughtful interpretation of results and limitations pervades every aspect of the manuscript. The statistical analysis and presentation is about as good as I've seen in any publication, e.g. attention to skew of data, use of nested t-tests. Use of recent technical developments is impressive, e.g. in situ single-molecule sequencing.

R1: We thank the reviewer for a constructive (re-)review of our study, for acknowledging the progress in improving the manuscript, and for supporting some of our previous responses to reviewer comments. Below are our responses to the current comments.

C2: Comment - Review #4. Cause of death. Remarkably, this is rarely known, even in humans. The condition at death can be documented, but which if any recognized pathology 'caused' death is usually unclear. The authors are correct that the pathological state can be documented, but the cause of death rarely identified.

R2: We thank the reviewer for reinforcing some of the challenges associated with cause of death, especially amongst compound physiological complications.

C3: Comment in several places in the Review Response - vacuoles in the testis with germ cell deficiency. This histology is very common in mouse mutants and environmental exposures. It seems to be a general feature rather than diagnostic for a particular defect or exposure.

R3: We agree and had not intended for this observation to have been characterised as a specific response to dysbiosis. The text notes the phenotype but does not suggest specificity.

C4: Question/comment. The tests for transgenerational effects to the F2 generation assume that the genetic background is permissive, i.e. if there is transgenerational inheritance, the background would show the epigenetically inherited phenotype. Interestingly, some genetic backgrounds are permissive and some not (see some early papers by Mike Skinner). The only way to know in this model system is to test several genetic backgrounds, a question and task that goes far beyond the scope of the present work. All that needs to be done here is to insert a phrase at the appropriately place saying that the test assumes that the genetic background is permissive. The observed phenotypes found in F1 mice could simply reflect direct paternal effects rather than true epigenetic inheritance.

R4: We *fully* agree that the background likely has major influence on the nature and extent of the effects we report. Thus, we have now articulated the importance of underlying genetics for a permissive model: "*...future work is necessary to deconvolve the phenotypically-relevant modalities of inheritance, and their applicability beyond murine models and across genetic backgrounds*". Exploring GxE effects within appropriate experimental designs represents an important frontier for future studies.

Referee #2:

The authors of the manuscript have re-submitted a revised version of their study, investigating the interdependence between paternal pre-mating environment, father’s gut microbiota, and intergenerational health using mice as a model system. The authors propose that non-absorbable antibiotics (and now osmotic laxatives) are capable of perturbing the paternal gut microbiota, leading to dysbiosis that increases the risk of placental insufficiency and low birth weight of offspring among other detrimental effects that affect the next generation’s fitness. Thus, authors suggest that the environment-dependent “gut-germline axis” in males has the capacity to determine the health of F1 offspring via altered sperm quality of the father.

C1: Authors have responded to each of the reviewers’ comments extensively, carrying out additional experiments to clarify their findings. Goals were to further investigate co-housing and its potential effects on microbiota transmission, testing the direct and indirect systemic effects of nABX, assaying different forms of gut microbiota perturbations and recovery effect, and assessing whether the father’s “gut-germline axis” also affect F2 offspring (which it does not). The amount of new work done is very impressive! However, the manuscript still falls short in addressing the fundamental questions, including some weaknesses not discovered previously.

R1: We thank the review for reconsidering our study and acknowledging that the “*amount of new work done is very impressive*”. We have spent considerable effort on validating our findings and ensuring the conclusions are supported. Below we respond to each comment, including the newly raised feedback, and articulate why the core conclusions fully stand.

Specific comments:

C2: The authors have clearly shown that exposure of paternal mice to nABX prior to mating affects: 1. The intestinal microbiome. 2. Important phenotypes in the male reproductive tract (however, the nABX were given to the males at 5 weeks of age, prior to sexual maturation, and this clearly had multiple effects on sexual development) 3. A number of phenotypes in the next generation offspring, but not in the F2 generation. The mechanisms between 1 and 2 were still not completely defined, but there was progress. This work involved administering non-absorbable low-dose antibiotics to 5-week-old male mice and assessing the effect of the potentially perturbed microbiome on host molecular and physiological changes and offspring fitness. One important point that needs to be addressed is that age 5-weeks is the timepoint at which male mice are undergoing sexual maturity, particularly spermatid elongation (seen in Extended data Fig 11 D, elongated spermatids as the population most affected by nABX). The authors should include quantitation of viable sperm as a male fertility parameter in their natural mating and IVF experiments.

R2: To address the reviewer suggestion to score sperm viability parameters, we have now performed extensive *new* viability experiments in a double-blind study and quantitated the average of two independent experiments, each with three biological replicate males. Because both natural mating and IVF throughout the study were conducted after 6-weeks induced dysbiosis (11wk old males), we characterised sperm parameters at this stage. First, we found both groups (nABX vs CON) demonstrate comparable sperm viability (> 85% motility) (*see new Extended data Fig 1H*, appended right). We further conducted extensive morphological analysis of head, mid-piece and/or tail defects, finding a slight overall increase in abnormalities in nABX males (47% vs 57%), albeit both values are within the normal range (*see new Extended data Fig 11G*). These data suggests no overt sperm viability issues. Importantly, this is consistent with fertility data already in the

manuscript showing dysbiotic males sire equivalent sized litters and have normal fecundity relative to control males (*see Extended Data Fig 1F-G*). Indeed, we previously noted reduced but non-significant sperm counts (*see Extended Data Fig 11F*). Thus, whilst nABX-induced dysbiosis impacts molecular features of sperm-borne inheritance (e.g. small RNA payload), it does not significantly affect the higher level parameters of sperm viability

and male fertility. In other words, dysbiosis does not impact fertility itself, but does have a more subtle yet critical impact on the nature of *what* is inherited.

A second issue raised by the reviewer regards the timing of sexual maturation. The first matured sperm appear in the cauda epididymis and vas deferens between 30 and 36 days postpartum, i.e. around 5 weeks of age (Janca et al., 1986). We have generated *new* data to verify this (*see Figure*, appended left), which shows mature sperm in the cauda epididymis by P35, implying the epididymis can release mature sperm by 5 weeks of age. Therefore, P35 is an appropriate time to begin nABX treatment. Indeed, since spermatogenesis is a continuous process, once the first wave of matured sperm is produced, gut dysbiosis can theoretically impact all stages of spermatogenesis during the treatment period. Thus, we began treatment once the male reproductive system became physiologically mature enough to release spermatozoa, thereby minimizing direct effects of organ development on spermatogenesis.

C3: There are still 2 remaining possible explanations for the intergenerational phenomenon that have not been fully addressed. First, could an altered sperm microbiome be carrying the phenotype? qPCR quantitates the total load only, not the composition. Was compositional analysis done? For specimens at which timing? There was discussion of this point, but more clarity is needed. The IVF experiments are very convincing—that might be the simplest and clearest experimental setting to determine if sperm microbes have an effect, first by in-depth microbial sequence analysis, comparing sperm from nABX vs control. A second approach would be to treat the nABX (and Control) sperm with a bactericidal antibiotic that does not penetrate intact mammalian cells—e.g. gentamicin. If there is no microbiome, the results should be unaffected, but the presence of a differential effect of the treatment would provide direct experimental evidence for a microbiota effect.

Second, is the effect occurring through the maternal microbiome after all? Specifically, the co-housing was for 7 hours a day only for 4 days, during a period of the daily cycle (9a-4p) when mice are not very active. This is not an adequate test of co-housing. In addition, after the 4-day co-housing (mating), at which time points were the mothers tested? Testing immediately after may have been too soon to detect the effect of invasion by the dysbiotic male microbiota, if such occurred. Generally, the flow of microbes is asymmetric in co-housing with stronger flow from the dysbiotic to the normal than vice versa. It appears that the next testing time for the mothers was P15—that also could be too late to detect a transient (but crucial) effect. Such analyses could be carried out in the short-term without a long multi-generational effect.

R3: The reviewer comment is centered on two alternative possibilities to explain the intergenerational phenomenon we report. Either (1) that a “sperm microbiome” is being impacted by nABX and transmitted, thereby influencing progeny or, (2) that the maternal microbiome is being impacted by paternal nABX regimes. These are critical questions that we believe our study has thoroughly addressed

experimentally. Below we discuss the *existing* lines of data detailing why these two possibilities are beyond reasonable likelihood, and then introduce *new* data that further reinforces this.

Firstly, could an “altered sperm microbiome be carrying the phenotype?”.

To date we have performed the following experiments that suggest strongly against this (1) The main antibiotics deployed to induce gut dysbiosis, are non-absorbable and therefore do not reach the systemic level nor the reproductive system. We confirmed absence of nABX in reproductive system with mass spectrometry and functional assays (*see Extended Data Fig 2A-D*). Therefore, any putative sperm microbiome is not impacted by this perturbation, which has specific action on gut microbes. (2) We also induced gut dysbiosis with an osmotic laxative that is inherently restricted to the gut (*see Fig 1I-J*). This again would not directly affect any putative sperm microbiome – only gut microbes - yet reproduces an intergenerational effect. (3) We carefully and extensively profiled the seminal fluid microbiome. Consistent with assertions #1 & #2, we do not find any significant differences in composition following nABX treatment (see also *new* analysis in subsequent paragraph). Indeed, microbial levels are so low as to be beyond robust detection (*see Extended Data Fig 8F*). (4) We recapitulated paternal intergenerational effects with *in vitro* fertilisation (IVF) (in two strains) (*see Fig 2H-I*). Here we used mature cauda-derived sperm. Anatomically, these reside before mixing with seminal microbes and are thus sterile (see below for explanation). This means that IVF-effects occur through sperm-borne inheritance, since there are no microbes to transfer. Taken together, multiple lines of evidence suggest with high confidence that there is **no altered seminal microbiome in our model(s)** and indeed, that such a ‘sperm’ microbiome **does not functionally carry the F1 phenotype**.

To support understanding this conclusion further, and prior to additional data (below), it is helpful to describe the anatomy of the male reproductive system, the ejaculatory process and where microbiota reside (*see Figure*, appended right). Functionally, mature sperm are stored in the cauda epididymis, which is a *bona fide* microbial-free part of a healthy male reproductive system. When mating occurs, mature sperm leave the cauda epididymis via the vas deferens and mix with seminal secretions in the ampulla. This mix is propelled toward the urethra, passing first by the prostate gland, where fluid is added to form semen. Hence, the term sperm microbiome (commonly called the seminal microbiome), does not refer to the fact that sperm themselves contain or carry microbes, as no evidence has yet been found to support this claim. The evidence thus far shows that it is the seminal fluid (possibly originating from seminal vesicles) that may carry the microbiome. To date, the strongest evidence for this has been reported from human semen samples (i.e. mature sperm and seminal fluid released by ejaculation) and three studies have reported microbiome from seminal vesicles in mice (*Javurek et al, 2016; Javurek et al 2017; Rosenfeld et al 2018*), but no evidence supports that matured sperm stored in a healthy cauda epididymis contain a microbiome. As noted above in #4, all the IVF experiments in our study were done with mature sperm harvested from cauda epididymis, which is prior to mixing with seminal fluid, ruling out a seminal microbiome effect. Thus, we appreciate the reviewer's suggestion of a possible IVF+nABX approach to clarify whether sperm microbes affect the outcome. However, since all IVF experiments were performed using matured sperm from a microbial-free source (i.e. cauda epididymis), it is not necessary to perform the suggested experiment.

Nonetheless, to address the experimental questions raised by the reviewer beyond the cumulative evidence in previous submissions, including seminal microbiome analysis, non-absorbable ABX/laxatives, IVF and co-housing experiments, we have now performed *new* 16S RNA-seq *and* quantitation via qPCR from ejaculated sperm (i.e. copulatory plug) and screened the entire male reproductive organ with staining to detect *living* bacteria. It is technically challenging to collect ejaculated sperm (copulatory plug) from mice, but with

continuous effort, copulatory plugs were collected and 16S rRNAseq was performed. These capture the composition of microbes actually transferred to females as a result of mating. Whilst sufficient signal after DNA amplification was low, we obtained six clear samples. No significant differences were found between the nABX and control group microbes in copulatory plugs (e.g. *see* Figure, right) further supporting that the phenotype is unlikely to originate with changes in transferred microbes. In addition, the entire male reproductive system was screened using Gram stain; bacteria were found to be absent from the cauda epididymis to the proximal region urethra (e.g. *see* Figure, overleaf). Moreover, we performed a highly-controlled sterile sample collection of seminal vesicles and microbiota DNA extraction, with extensive procedures in place to prevent contamination. 16S rRNA-seq of the seminal fluid did not detect differences between CON and nABX samples, whilst the quantitative levels were actually below threshold detection relative to the background environment (*see* Extended Data Fig 8F). These results represent further lines of evidence that the seminal microbiome does not transfer the phenotype since it is unaltered by perturbations either in copulatory plugs and upon direct measurement, and indeed occurs at extremely low levels *per se*.

We conclude that the cumulative data present a compelling case that non-absorbable ABX and osmotic laxatives are not mediating F1 effects via impacting a seminal microbiome.

Secondly, the reviewer asks “is the effect occurring through the maternal microbiome after all?”

As above we believe we have addressed this question fully with the controls and experiments in previous submissions. To clarify the extant data: (1) IVF experiments performed with two different strains of foster mother (CD1 or C57BL/6J), and with many replicates, strongly support that this is not primarily a maternal effect. In these cases, the mother has absolutely no exposure to the father, only to purified cauda-sperm, meaning that the transmitted effect must pass through sperm (i.e. not via an indirect maternal microbiome change) (*see* Fig 2H-I). These IVF experiments specifically test the potential for maternal effects. (2) Corroborating the lack of a functional maternal effect, profiling the maternal microbiome revealed no significant changes after mating with nABX vs CON males. Note that we profiled the mother directly after mating *and* post-partum, thereby covering all possibilities for a putative microbiome effect to manifest (*see* Extended Data Fig 8A-H). (3) Extensive co-housing experiments reveal that prior exposure to an nABX male (and faeces) does not impact subsequent offspring fitness – only siring suffices - providing another line of functional evidence for a germline-transmission (*see* Extended Data Fig 9A-D). Ruling out these confounders are rarely undertaken in research studies, and has entailed considerable effort to reach this point.

We believe that the independent functional experiments (co-housing and IVF) *and* extensive microbiome profiling conducted at different stages adequately answers the reviewer's questions. To provide further clarity, here we highlight some critical points related to the reviewer's comments about the co-housing experiment. During natural mating paradigms that formed the bulk of experiments, females were housed for a maximum of 4 days (typically 1-2 days) with nABX or control males, leading

to an intergenerational effect. Therefore, we designed co-housing experiments (without mating) to proceed for 4 days to recapitulate this timeframe precisely. Since microbiota are shared primarily through coprophagy, whether the male mice remain in their cages or not, the female mice continue to consume the male faeces. Hence, it is important to note the reviewer's comment "*co-housing was for 7 hours a day for only 4 days, during a period of the daily cycle (9a-4p)*" refers to the time period in which male mice were housed with female mice, not their exposure to the male microbiome. As mice are coprophagous, if kept in the male cage, the female mice are exposed to male mice's microbiome continuously for 4 consecutive days.

Irrespective, to assess the maternal microbiome contribution to the offspring phenotype, we carried out the co-housing experiment in two ways: (1) Continuous co-housing (night and day, regardless of plug positivity) followed by comparing the microbiome profiles of mating pairs at pre- and post-mating (day 0 and day 4) , as well as postnatal day 21: no significant maternal changes were found (see Extended Data Fig 8D-H). (2) Co-housing the mating pairs during the day in the nABX male cage, leaving female mice alone in the same cage overnight to prevent mating, and then after 4 days using plug-negative females (i.e. un-sired by nABX males) to mate with CON male mice kept in the same experimental room. This second co-housing experiment examined whether changes in the microbiome composition of the mating female during the 4-day exposure affected offspring phenotype outcome or not. **We found no significant change** (see Extended Data Fig 9A-D).

In summary, we can make strongly supported conclusions regarding both possibilities raised by the reviewer: paternal transmission of an altered seminal microbiome, or functionally-relevant maternal changes in microbiome in response to nABX males, owing to our step-by-step experimental approach, namely:

- (1) IVF-mediated transmission predominantly rules out maternal effects of paternal microbe transmission, and implicates sperm-mediated inheritance;
- (2) Analysis of seminal vesicles indicates no seminal microbiome (change) in dysbiotic males;
- (3) Ejaculated semen microbiome profiling (copulatory plug) further reveals no change in transmitted microbes from male>female;
- (4) Co-housing experiments using two approaches show no functional effect of exposure to dysbiotic males on offspring phenotype;
- (5) Pre- and post-mating microbiota profiles of breeding pairs show no microbiome difference in mother mated with nABX males and;
- (6) Pre- and post-birth changes of maternal microbiome at P21 are undetectable arguing against maternal-effect.

The cumulative evidence supports that microbial transmission is not the primary mechanism at play but rather that transmission occurs preferentially via the paternal germline. We thank the reviewer for ensuring that these potential confounders are thoroughly investigated, since the experiments conducted as a result have enabled clear conclusions.

C4: Progressive loss after resting for 4 weeks, and more so after 8 weeks does not really address whether maternal or sperm transmission is important, only that there is recovery in the fathers over time. In Extended Figure 8A are the top and bottom labels reversed? The Results should focus on the 6 +0 experiment only, with sufficient numbers to rule out subtle signals. What was the time point for this study? If there were multiple time points (6 = 0, 6 + 4, 6 + 8), they should each be shown separately. When proposing a new mechanism, the burden of the proof is for the proposer to consider all possible ways in which the conventional explanation can be ruled out. Extended Figure 8B includes the 6+4 and the 6+ 8 offspring, which might dilute out any signal in the 6 + 0 offspring, and those should be eliminated from the Figure. Similar to 8A, what was the time point for this study? If there were multiple time points, they should each be shown separately.

R4: We first confirm to the reviewer that Extended Figure 8A is not mislabelled: it showed the faecal microbiome composition of the mother (upper) and offspring (lower) following mating with either a control or nABX male (now reconfigured to be shown separately). However, the figure legend was inverted and we apologise for this oversight. We have been through carefully to check other legends and find all to be correct. Regarding the reviewer suggestion to stratify mother’s microbiome according to the treatment regime of the male she mated with, this is appended right, and is already included as Extended Data Fig 8A. In all cases, no difference was found in mother (or offspring) following mating with nABX males, irrespective of the stage of male perturbation (i.e 6wk, 6+4wk, 6+8wk). Furthermore, alpha diversity in mothers stratified by male she mated with showed no significant effect between nABX and CON father treatment (6wk $p=0.77$; 6+4wk $p=0.31$; 6+8wk $p=0.38$) This is indicative of transmission independent of offspring/mother microbiome, since there is no significant microbial change in mothers. This is also shown in via complementary analyses in Extended Data Fig 8D, 9A-D). Of note, in all experiments each female was only used once for mating.

Regarding the remaining point about diluting the signal, it is important to state that 6+4 males (dysbiotic) transmit the phenotype to F1, therefore the profile of females mated with them does not “dilute out any signal” but is an integral part of it. In any case, stratified profiles are shown above. Moreover, we have addressed this question of whether an altered maternal microbiome underpins the effect comprehensively and functionally in R3.

C5: Extended Figure 8C does not rule out sperm microbe transmission.

R5: Extended Data Figure 8C shows the correlation between offspring phenotype severity and microbial taxa changes in either its own microbiome or that of its father. The reviewer is correct that this does not rule out sperm microbe transmission, but that is not the objective of the figure. Rather the figure demonstrates that taxa changes in the father exhibit some positive or negative correlations with the offspring phenotype, but that the offspring’s own microbiome is not correlated with phenotype. This begins to explore the question, what is the relevant change in the paternal gut microbiome composition that leads to offspring effects, and following this up in future studies is warranted. Nonetheless, with reference to the reviewer comment and as detailed fully above (see R2-R3), we have generated many independent lines of experimental evidence that effectively rules out the seminal microbiome as a credible source of phenotypic transmission.

C5: Considering Authors’ Response to Comment 2: Subresponse 1: See above the limitations of the author’s response 1.

R5: Please see R3 above, which addresses this fully.

C6: Subresponse 2: The short co-housing might not have been sufficient to change the maternal microbiome, and thus the lack of a change in phenotype would not be surprising. This is circumstantial evidence.

R6: To extend our response from R3, all natural matings with dysbiotic males were conducted for a maximum period of 4 days, with plugs typically after 1-2 days. Thus, all our experiments that report a robust probabilistic F1 phenotype are derived from a maximum parental co-housing of 4 days. To test whether the co-housing itself, rather than sperm, was responsible for F1 phenotype transmission, we precisely recapitulated the 4 days in the co-housing design. In other words, if 4 days “co-housing might not have been sufficient to change the maternal microbiome” than it cannot be responsible for

phenotype transmission in natural mating either, since this followed the same or shorter co-housing period. Consistently, we do not detect a change in the maternal microbiome after the appropriate time 4 days.

C7: Subresponse 3: In addition, the avaABX cocktail includes ampicillin, which is well-absorbed—this does not meet the definition of nABX. Not clear why studies with this regimen should be included at all.

R7: We agree that ampicillin is well-absorbed. Following on from our extensive experiments with non-absorbable antibiotics, we wished to recapitulate paternal gut microbiome dysbiosis via orthogonal strategies, to examine whether the F1 effect is robust to the mode of induced dysbiosis. We chose osmotic laxatives and an alternative antibiotic cocktail (avaABX). It is important to note that we did not describe this as non-absorbable, and it is labelled as ‘general antibiotics’ in Fig 11. However, since it recapitulated the F1 effects of both nABX and osmotic laxatives, which *are* restricted to the gut, this provides another line evidence linking gut dysbiosis to reduced F1 fitness, which supports the main nABX model. We have now further modified the experimental description to point out specifically that avaABX is absorbable: “*ad libitum administration of a cocktail of absorbable antibiotics*”

C8: Considering Authors’ Response to Comment 3: Please see above comments about IVF. Also, what was the percent of high quality sperm in the nABX and control groups?

R8: We have addressed this fully in R2 and R3. Besides a slight reduction in germ cells, no significant differences have been observed in sperm quality between the two groups. Both groups (nABX vs CON) demonstrate comparable sperm viability (> 85% motility) and morphology (see Extended Data Fig 1G, 11F-G).

C9: Considering Authors’ Response to Comment 5: Appropriate
Considering Authors’ Response to Comment 6: Appropriate
Considering Authors’ Response to Comment 7: Appropriate

R9: Thank you for the acknowledgments.

C10: Considering Authors’ Response to Comment 8: Greater clarification and extension of the experimental results needed, as indicated above.

R10: We have described and extended this in the methods sections. Due to word restrictions, and considering the vast amount of data in the manuscript, there is limited possibility clarify this further in the main text.

C11: Considering Authors’ Response to Comment 9: Appropriate
Considering Authors’ Response to Comment 10: Appropriate
Considering Authors’ Response to Comment 11: Appropriate, agree that phenotypes are non-genetic.

R11: Thank you

C12: Considering Authors’ Response to Comment 12: The suggestion to give the antibiotic courses for 6 weeks to germ-free mice, still stands. If there is no effect on the reproductive phenotypes, it can be

concluded that a (perturbed) microbiome is required. In addition, MS was done for the nABX regimen but not for the avaABX regimen that used ampicillin, which is the agent that is absorbable.

R12: Thank you for suggesting this control experiment, which we have now performed generating an entirely *new* dataset. We subjected germ-free (GF) males to nABX treatment for 6 successive weeks, exactly as for conventional mice, and assayed for transcriptional responses. We found that GF testis of males exposed to nABX showed no difference from those treated with sterile water, implying that nABX *per se* has no direct impact on the male reproductive system in the absence of microbiota changes (see Figure, appended right). This in agreement with multiple lines of evidence that nABX do not reach the testis and have specific action on gut microbes (e.g see Extended Data Fig 2). Moreover, since orthogonal strategies that perturb the gut microbiome elicit comparable F1 effects, this further suggests specific perturbation of the gut microbiome underlies effects (e.g see Fig 1I-J). Note also that GF mice are widely used to validate FMT or monocolonization experiments, but they differ greatly in physiological and organ development from the nABX model (Kennedy et al, 2018). For example, the testis of GF mice suffers both developmental and meiotic defects from birth due to a lack of gut microbiota during development. In contrast, the nABX model reached sexual maturity (P35) in the presence of microbiota and thus correspond to a more physiologically relevant perturbation model. Regarding ampicillin, as discussed in R7, we do not suggest that it is non-absorbable but rather it is deployed as one of several orthogonal perturbation strategies to validate nABX effects. MS is therefore not necessary.

C13: Considering Authors' Response to Comments 13-23: All appropriate

R13: We appreciate the discussion on these points

C14: Other Specific comments: The authors have used a nested statistical analysis to compare the litters sired by individual fathers rather than individual pups to one another, which is the correct approach. However, for example, in Figure 1C, the data are presented as individual pups. Authors should consider adding a plot where the data is presented as litter averages, add S.E.M. to their figures, and specify whether the “severe growth restriction” individuals are from the same father. This applies to all figures where data are analysed in a nested way but presented as individual pup data points. Importantly, figure 1C appears to combine male and female pups—if this is correct, they should be separated and sex-specific analyses done. Same for 2H and 2I. Extended 3C, 5A, 7B, 8I, 9B,C—why combined?—there is plenty of room in Extended to show both sexes individually to determine if the effects were parallel, and significant

R14: Thank you for your comments and for acknowledging that we used the appropriate statistical methods. In this study, we aimed to overcome two major limitations of previous intergenerational studies: 1) Insufficient replicate size and 2) Absence of nested statistical analysis. The data presented in this paper shows individual pup datapoints while displaying the statistical value of the nested t-test, which addresses both gaps, and is therefore the most appropriate way to convey our message to scientists and readers. Regarding sex specific effects – we observed statistically significant impacts on growth trajectories and risk of severe growth restriction for both male and female F1 offspring sired from nABX-treated fathers. These data and hierarchical (nested) *p-values* are already shown in Extended Data Fig 3A-B. Because we observe and present evidence that the central F1 effect is independent of sex, we chose to combine data to increase statistical power and maximise figure space, given the comprehensive datasets already shown in figures. To address the reviewer point, the

manuscript includes the line: “Both female ($p=0.017$) and male ($p=0.029$) offspring were affected, whilst litter size was constant”.

C15: Authors discuss the role of leptin in intragenerational health and signalling in the “gut-germline axis”. Did the authors test/consult previous literature for hypotheses about how exactly leptin is affected by gut dysbiosis and then consequently affects the germline? Moreover, which recovered first, father’s gut microbiota or leptin levels? In the transcriptomic profile of dysbiotic male testes, the authors found that the Leptin gene was strongly affected by nABX. Then they included ob/ob mice to study the functional role of Leptin in their system. However, this experiment fails to have an appropriate control. In the IVF experiment, they should include sperm from 6wk ob/ob or WT that received nABX or not. If Leptin plays a central role, then the F1 offspring sired by ob/ob-nAbx father will show no phenotypic change compared to the ob/ob-control offspring.

R15: The potential role of leptin as a signalling mechanism of dysbiosis to germline is an extremely interesting question to consider and discuss. Following the reviewer suggestion, we have now performed an entirely *new* set of IVF experiments using sperm from *Ob/Ob* fathers with or without 6wk nABX treatment, followed by single-embryo transcriptomics. First, we independently confirm that the offspring of leptin-deficient fathers exhibit a clear separation from control fathers as early as the blastocyst stage. Secondly and importantly, we find that offspring from *Ob/Ob* +nABX fathers cluster closely with those from *Ob/Ob* in PCA. This supports our conclusions, and as the reviewer anticipated, suggests that leptin deficiency plays an important role in triggering intergenerational phenotypic effects (see Figure, appended right). It is important to note however that future study is warranted to dissect the precise relationship between leptin deficiency and paternal gut dysbiosis and intergenerational phenotypes. To this end, this study lays the groundwork but rigorously dissecting the inheritance mechanism in detail is beyond the scope of this study and could take >5 years to complete.

C16: Additionally, the ob/ob mice the authors used in this study have germline mutation; as such, it's not surprising to see segregation in the PCoA plot of the F1 blastocyst transcriptome. To better understand the role of reduced levels of leptin in testes, the author should consider using a testes-specific ob/ob knockout.

R16: We demonstrated that leptin perturbation in fathers impacts offspring by using 6wk-old *Ob/Ob* fathers. We observe a striking transcriptome/phenotypic difference in the offspring of leptin-perturbed *Ob/Ob* fathers, which we detect clearly and at least as early as the blastocyst stage. The reviewer suggests this is because offspring are heterozygous (*WT/Ob*). This however cannot be the explanation for the following reason: Crucially, leptin is not expressed at any stage during early embryonic development, and is only expressed after gastrulation in adipose tissue and germ cells, which have not been specified yet. Indeed, we confirmed *leptin* is completely undetectable in blastocysts in Extended Data Fig 14G (appended right) to make this point. This is critical because whether blastocysts are heterozygous or wildtype for *leptin* at this stage is therefore not relevant, since no expression is observed in any case. In other words, there is no possibility of haploinsufficiency to manifest prior to the developmental onset of leptin expression, so the fact that they are heterozygous does not explain their phenotype. Thus, **the major transcriptome differences in blastocysts derived from leptin-deficient fathers can only be linked with a paternal effect** (i.e. intergenerational

inheritance of perturbed sperm) rather than the offspring (heterozygous) genotype. This supports the conclusion that a father's leptin levels can impact offspring.

Furthermore, as the reviewer noted, conditional knockout mice (i.e. a testis-specific *Ob/Ob* knockout) might provide additional insight into the effects of testicular leptin deficiency on offspring phenotypes. However, conditional knockout mice cannot confirm the intergenerational effects of leptin, for two reasons: (1) Dysbiotic gut microbiota affect both systemic (serum) and local (testis) leptin levels, making full knockout mice an ideal model to investigate intergenerational effects. (2) Despite conditional knockout of leptin in the testis, circulating leptin can still cross the blood-testis barrier as it has local and systemic effect. For these reasons, using conditional knockout mouse is not appropriate for this study.

C16: Did the authors test whether the experimental mice were drinking equal amounts of nABX water? Microbiota perturbations generally are dosage dependent.

R16: We have tested this during initial optimisations and the amount of water consumption among nABX mice is comparable across the 6-week period (see right Figure). We observed a slight difference each week and as they get older, otherwise the differences are insignificant among each animal. For a 6-week treatment, the average intake per mouse for 5 days is about 25-30ml. The plot shows the average volume of water consumed by six independently tested male mice in each group (CON vs nABX).

C17: Authors have used rarefaction to analyse their 16S rRNA gene data. They should specify the average number of reads +/- SEM per sample in their data set prior to rarefying.

R17: Thanks for this insightful question, the revised manuscript includes the following figure (Extended Data Fig 8D, also shown right). We have also computed the mean and SD of the per-sample read depths after initial quality filtering, but prior to rarefying:

Run 1 (primary): 13685 ± 3187; Run 2 feaces: 4969 ± 2065; Run 2 sperm plug (6 samples): 2374 ± 1301.

It is also worth pointing out that rarefied data was used exclusively for the alpha diversity analyses; all other analyses were run based on relative abundances of taxa, with samples failing to meet QC for sequencing depth discarded as described in the methods.

C18: From the aspect of microbiota analysis, it would be interesting to see a PCA plot with all the father-mother-offspring data combined to see how they cluster, considering the control vs. nABX males and pre- and post-mating females. Authors should also consider showing which specific bacterial taxa (e.g., relative abundance plot) are present in control vs. nABX males).

R18: Thanks for your question. Appended overleaf is a relative abundance plot illustrating the relative changes in taxa upon paternal nABX. Also appended is the combined PCA requested.

Referee #3

C1: I thank the authors for thoroughly addressing my comments: The added data and figures clearly strengthen the most exciting male gut-germline association and the parental contributions to intrauterine and postnatal development. The mechanisms underlying the convincingly presented associations remain enigmatic. There are data that implicate the placental in the post-natal outcomes, again, without a mechanistic link. Implicating pre-eclampsia in the pathophysiologic process, merely because of some shared placental analyte patterns, seems unjustified.

R1: We thank the reviewer for the constructive and balanced feedback on our manuscript submissions. We believe our study uncovers a phenomenon that has broad implications for our understanding of inheritance, environmental signalling to organisms and the potential origins of disease risk. Moreover, the scale of replications, controls and orthogonal approaches provide, we believe, the most compelling support for transmission of phenotypically-relevant non-genetic information between mammalian generations yet. However, the underlying set of mechanisms that propagate information from a dysbiotic microbiome, to a systemic signal, to a germline response, to sperm-borne signal(s), and finally to reduced fitness in offspring are likely deeply complex and challenging to unravel. These will involve systems physiology and compound molecular responses that will require new technologies and research strategies to deconvolve and fully explain over coming decades. Nevertheless, whilst we do not claim to have dissected such complex mechanisms, we have uncovered a number of mechanistic components that likely contribute to each stage of the phenomenon, such as paternal endocrine response, altered small RNA payloads in sperm, and paternally-induced placental insufficiency. With respect to the latter, we agree with the reviewer that mice are not an ideal model for studying pre-eclampsia. For this reason, the broader term "placental insufficiency" is now used predominantly in the manuscript instead of pre-eclampsia. We do however identify several molecular indications of a pre-eclampsia-like phenotype. To raise this possibility whilst ensuring accuracy, we occasionally refer to a "preeclampsia-like" phenotype or to "hallmarks of pre-eclampsia" in the revised manuscript.

Referee #4

C1: In the revised manuscript, the authors addressed most of my concerns. However, there are some remaining issues/new ones, as summarized below.

R1: We thank the reviewer for their time in re-assessing our study and recognising the progress we have made. We further respond fully to the remaining issue(s) below.

C2: The newly implemented scRNAseq analysis of mouse testes is critical for distinguishing cellular and molecular changes treated with nABX. However, many details of scRNAseq analysis are missing. The authors should provide more details, including: i) representativeness of their 4 samples from each group. For example, the authors should provide tsne plots for individual samples; also, the authors should provide the cell percentage of each annotated cell subsets from individuals. ii) more specific and more markers should be used to define these cell populations. The authors only used a single marker per each cell subset to define them (Extended data Fig 11B), and some of them are inappropriate. For example, *Dazl* is a pan spermatogonial marker, not specific for spermatogonial stem cells. Likewise, markers used for identifying spermatocytes and spermatids are not specific. iii) the authors used "logfc.threshold=0.25 and p_val_adj<0.1" as cut-off for identifying DEGs from scRNAseq. This is too flexible, especially considering the shallow sequence depth of 10X.

R2: Thank you for the constructive feedback and insightful comments. The four samples were pooled prior to GEM generation, thereby ensuring we utilised enough replicates to capture robust responses, whilst mitigating the (considerable) 10X library prep expense, a cost that isn't affordable for every sample. It is important to note that our primary goal in performing scRNAseq is to verify findings from robust bulk profiling (i.e. total RNA and metabolomics profiling), with the potential to capture single-cell responses. The objective is therefore to confirm the principle that male gut dysbiosis triggers a detectable testicular response at gene expression and cellular levels. Regarding (ii) cell markers for clustering, it is important to note that we did not use a single marker to define clusters, but chose to show representative examples of cluster-specific expression in the Figure 11B. To actually define each (unbiasedly) identified cluster, we searched for multiple markers specifically expressed in the clusters and cross-referenced those genes with previously characterised cell-type-specific genes in classical *and* single-cell transcriptomics studies on the testis. We now provide a more comprehensive output of this analysis as a *new* Extended Data Fig 13B, also appended right.

Following the reviewer final suggestion (iii), this revised manuscript now utilises an considerably more stringent threshold ($p(\text{adj}) < 0.01$; $\log(\text{FC}) > 0.25$). Importantly this results in only a minor change in DEG calling (previous DEG n=232; stringent DEG n=186) and has no impact on the biological conclusions. We have updated Extended Data Fig 13 and the methods to reflect these more stringent parameters.

C3: A following-up concern of scRNAseq analysis: is the post-meiotic germ cell number significantly reduced in the nABX group, as compared to the control group?

R3: As we have shown in the scRNAseq and mature sperm counts (Extended Data Fig 13F and Fig 11F), we observe a trend of reduction in post-meiotic germ cells, though not statistically significant. This is consistent with no change in fertility of nABX-exposed males (*see* Extended Data Fig 1), and with the gut-testis interaction occurring at a physiological level rather than a toxic level. From a biological perspective, this is what makes this study and model interesting. Accordingly, gut microbiota dysbiosis leads to a physiological response of the testis, suggesting that a healthy gut microbiota may play an important role in optimising reproductive processes. It is also important to note that regardless

of whether the gut microbiota affects testis function at a cellular or molecular level, our interpretation and core conclusions regarding the existence of a gut-germline axis remain the same.

C4: In the revised manuscript, the authors used *in situ* sequencing (ISS) to show the expression of Leptin in the wild-type testis. The authors could use ISS to compare Leptin expression between Control vs nABX at single cell level. This is useful for validating their bulk RNA-seq.

R4: Thanks for bringing this to our attention. We have now performed *new* ISS on nABX-exposed and control testis to validate leptin expression changes. A total of 5 control and 5 nABX independent males were used to quantify the number of transcripts expressed in seminiferous tubules from the same developmental stage. This analysis confirmed testis of nABX-treated males have lower *leptin* expression than control, in terms of both transcript count and overall intensity, consistent with bulk RNA-seq data (*see new Extended Data Fig 13H*, also appended right). It is also important to note that we had already validated such a response at the level of leptin hormone (protein) in testis (*see Fig 3H*). We would finally like to point out that, due to technical limitations of ISS, we were unable to compare at the single-cell level. Specifically, identifying each subpopulation and assigning each dot (transcript) to a specific spermatogenesis step is difficult after the staining procedure, as all membranes are either destroyed or opacified. The software currently available cannot allocate each datapoint precisely to a specific cell type in highly heterogeneous cell populations like testis tissue. Nevertheless, the cumulative data from multiple independent assays suggest significant reduction in *leptin* at both RNA (bulk and ISS) and protein (ELISA) levels.

C5: The reviewer does not agree with the author's argument (R6 to C6). While, yes, RNA-seq/small RNA-seq are mature techniques that are “technically equivalent” to qPCR/TaqMan, it IS still critical to validate key findings using independent techniques, especially considering that these authors only provided correlative evidence.

R5: We have now addressed this experimentally by using target-specific and quantitative TaqMan-PCR. These assays were performed on purified sperm samples derived from four *new* nABX-exposed males (quantitated separately). Specifically, we aimed to validate the top two small RNA transcripts mis-expressed in nABX sperm, as judged by our small-RNAseq experiments (miR-141-3p and tRNA-Gly-GCC-2) (*see Figure 4B*). Given the high costs of TaqMan assays, we are limited in number of targets, but nevertheless cover two classes of small RNA. We found both miR-141-3p and tRNA-Gly-GCC are more abundant in nABX sperm relative to control (*see new Extended Data Fig 16D*, appended right), independently confirming previous results. Indeed, the effect-size of nABX exposure on small RNA expression is actually greater in these quantitative experiments. We thank the reviewer for promoting this experiment which reinforces our conclusions using both an orthologous assay and independent samples.

C6: The format of refs for M&M is wrong.

R6: We have corrected this.

Reviewer Reports on the Second Revision:

Referees' comments:

Referee #1 (Remarks to the Author):

Again, an already exceptional body of work and manuscript is made even better with new experiments, analyses and revisions. This work rigorously defines a new axis of phenotype determination, connecting phenotypes in one generation with functions (here microbes) in the previous generation. The problem is first-order, the study design rigorous, and the results clear, and the interpretation appropriately cautious. Like any pioneering study, many new and important questions are raised that remain to be answered - a signal of truly cutting-edge work. My concerns, questions and suggestions have been more than adequately addressed.

Referee #2 (Remarks to the Author):

This manuscript focuses on the very novel and exciting subject of the potential interdependence between paternal pre-mating environment, the father's gut microbiota, and F1 offspring fitness. In particular, the authors propose that paternal gut microbiota perturbation (induced via the administration of antibiotics or laxatives) increases the risk of lower offspring birth weight (reduced fitness) via placental insufficiency. The authors have carried out multiple additional experiments and data analysis to answer each of the referees' questions.

We agree with the authors' responses and the fact that they have provided extensive new data within the scope of this manuscript. We suggest minimal, yet important text/figure edits prior to publication. Please see the suggested edits below in cursive. The other suggestions are lower priority (optional).

The additional sperm viability tests are sufficient and strengthen the manuscript (R2). Please add N= to the Extended data Fig. 1G. Please specify whether N=31 and N=36 in Extended data Fig. 1F indicate the number of pups or the number of litters.

Agree with the stepwise approach to addressing the F0 -> F1 generational phenomenon the authors report (R3). Subsequently, agree with the responses R4-R6, R8-11.

Agree with R7.

Agree with R12. The additional germ-free experiment strengthens the manuscript.

Regarding R14. Since the major measurement of offspring fitness used by the authors is offspring weight, it is vital that the authors demonstrate that the "severe growth restriction" individuals are not from the same litter. This could be done via colouring the data points by father's ID (Fig. 1C) and/or providing statistical analysis that uses father's ID as an explanatory variable while the response variable is pup weight. Even more simply, adding a sentence, e.g. "the 11 pups with severe growth restriction came from X individual litters", would clarify this point. Also, are the individuals in Fig. 1D and Extended Data Fig. 3E all from different fathers? Please specify in both cases.

Agree with R15 and R16.

For future consideration regarding α -diversity (R17), avoiding rarefying would be better. See, for example, <https://doi.org/10.1371/journal.pcbi.1003531>.

Agree with R18.

Referee #3 (Remarks to the Author):

The work is excellent, and illuminates a previously unknown paternal gut-germline axis. I am certain that the work will stimulate new hypotheses and numerous research directions, which will likely have direct relevance to human biology. As previously noted, and also acknowledged by the authors in their summary paragraph, more work needs to be done to untangle the link between the affected male germline (RNAs, metabolites) and the pregnancy phenotype.

The data, along with the new experiments, provide exciting possibilities related to sperm changes that might modify key placental genes, such as leptin (which is highly expressed in the human placenta), PGF, prolactin genes, Syna, Hand1 and others. It might be interesting to assess the level of leptin in the placenta, as it plays a key role in placental development (and invasion, for which the mouse might not be a good model). As a rescue experiment, could the authors add leptin to the pregnant dams? Similarly, sperm piRNA play very important developmental roles. If assessed, can the data be added? Finally, feto-placental vascularization might be better detected using CD31 and Muc1 Abs to delineate distinct vascular spaces.

As noted before, implicating pre-eclampsia (or pre-eclampsia-like) in the pathophysiologic process, merely because of some shared placental analyte patterns, seems unjustified. Obviously, if Plgf is decreased, the ration of sFlt/Plgf will be increased. Was sFlt alone elevated? The identified genes likely represent placental dysfunction. Without additional experiments (measuring blood pressure, showing renal changes, etc.) there is no justification to implicating pre-eclampsia in the findings.

Referee #4 (Remarks to the Author):

The authors fully addressed all my concerns and I'm happy to recommend its publication.

Author Rebuttals to Second Revision:

Response to Reviewers (3rd round) Argaw-Denboba *et al*

Referee #1 (Remarks to the Author):

C1: Again, an already exceptional body of work and manuscript is made even better with new experiments, analyses and revisions. This work rigorously defines a new axis of phenotype determination, connecting phenotypes in one generation with functions (here microbes) in the previous generation. The problem is first-order, the study design rigorous, and the results clear, and the interpretation appropriately cautious. Like any pioneering study, many new and important questions are raised that remain to be answered - a signal of truly cutting-edge work. My concerns, questions and suggestions have been more than adequately addressed.

R1: We greatly appreciate the efforts and input of the reviewer in providing constructive feedback and improving our study during this process.

Referee #2 (Remarks to the Author):

C1: This manuscript focuses on the very novel and exciting subject of the potential interdependence between paternal pre-mating environment, the father's gut microbiota, and F1 offspring fitness. In particular, the authors propose that paternal gut microbiota perturbation (induced via the administration of antibiotics or laxatives) increases the risk of lower offspring birth weight (reduced fitness) via placental insufficiency. The authors have carried out multiple additional experiments and data analysis to answer each of the referees' questions. We agree with the authors' responses and the fact that they have provided extensive new data within the scope of this manuscript. We suggest minimal, yet important text/figure edits prior to publication. Please see the suggested edits below in cursive. The other suggestions are lower priority (optional). The additional sperm viability tests are sufficient and strengthen the manuscript (R2). Please add N= to the Extended data Fig. 1G. Please specify whether N=31 and N=36 in Extended data Fig. 1F indicate the number of pups or the number of litters. Agree with the stepwise approach to addressing the F0 -> F1 generational phenomenon the authors report (R3). Subsequently, agree with the responses R4-R6, R8-11. Agree with R7. Agree with R12. The additional germ-free experiment strengthens the manuscript. Regarding R14. Since the major measurement of offspring fitness used by the authors is offspring weight, it is vital that the authors demonstrate that the "severe growth restriction" individuals are not from the same litter. This could be done via colouring the data points by father's ID (Fig. 1C) and/or providing statistical analysis that uses father's ID as an explanatory variable while the response variable is pup weight. Even more simply, adding a sentence, e.g. "the 11 pups with severe growth restriction came from X individual litters", would clarify this point. Also, are the individuals in Fig. 1D and Extended Data Fig. 3E all from different fathers? Please specify in both cases. Agree with R15 and R16. For future consideration regarding α -diversity (R17), avoiding rarefying would be better. See, for example, <https://doi.org/10.1371/journal.pcbi.1003531>. Agree with R18.

R1: We thank the reviewer for the time and care they have invested in our study, and for the constructive feedback. We have incorporated the relevant textual and figure suggestions into the legends and manuscript, for example, confirming the N=31/36 refers to the number of litters (not pups) and adding "N=" to Extended Data Fig 1G. Moreover, we can clarify that the SGR individuals shown in Fig 1d and ED Fig 3e (now 2e) are each from different fathers (n=6 SGR pups from N=2 fathers), whilst SGR pups from N=3 additional (independent) fathers are shown in ED Fig 2f. To specifically address the question, the 11 SGR pups in the first nABX cohort came from 5 fathers, with subsequent experimental cohorts uncovering further independent fathers. We would also emphasise that the assertion that "the major measurement of offspring fitness used by the authors is weight" is not our view. Rather we suggest offspring survival is the key measurement, since it is unambiguous and captures effects from multiple phenotypes with minimal bias/confounders. With respect to this, we found that pups from 15 of 28 nABX-derived litters died within the test period, compared with pups from only 4 of 26 control litters. In other words, F1 lethality occurred from more than half of nABX-treated fathers.

Referee #3 (Remarks to the Author):

C1: The work is excellent, and illuminates a previously unknown paternal gut-germline axis. I am certain that the work will stimulate new hypotheses and numerous research directions, which will likely have direct relevance to human biology. As previously noted, and also acknowledged by the authors in their summary paragraph, more work needs to be done to untangle the link between the affected male germline (RNAs, metabolites) and the pregnancy phenotype. The data, along with the new experiments, provide exciting possibilities related to sperm changes that might modify key placental genes, such as leptin (which is highly expressed in the human placenta), PGF, prolactin genes, Syna, Hand1 and others. It might be interesting to assess the level of leptin in the placenta, as it plays a key role in placental development (and invasion, for which the mouse might not be a good model). As a rescue experiment, could the authors add leptin to the pregnant dams? Similarly, sperm piRNA play very important developmental roles. If assessed, can the data be added? Finally, feto-placental vascularization might be better detected using CD31 and Muc1 Abs to delineate distinct vascular spaces.

R1: We are highly appreciative of the input and effort put into reviewing our study. With respect to further experimental s suggested: (i) We agree sperm piRNA are interesting and have already included a full analysis of their abundance in control and dysbiotic contexts (see Extended Data Fig 11). (ii) Whilst we show short-term paternal leptin changes appear to influence male reproductive tissues and early F1 development, there is no evidence maternal leptin is involved in any aspects of this phenotype as yet. This is therefore any entirely new line of enquiry that should be investigated in future. (iii) We agree CD31/Muc1 are excellent for delineating vascular spaces, but feel that the multiple and orthogonal approaches we have applied (RNAseq, physiology, placental infarction, feto-placental trophoblasts) as well as the IB4 endothelial cell marker for vascularisation, are sufficient for our conclusions.

C2: As noted before, implicating pre-eclampsia (or pre-eclampsia-like) in the pathophysiologic process, merely because of some shared placental analyte patterns, seems unjustified. Obviously, if Plgf is decreased, the ration of sFlt/Plgf will be increased. Was sFlt alone elevated? The identified genes likely represent placental dysfunction. Without additional experiments (measuring blood pressure, showing renal changes, etc.) there is no justification to implicating pre-eclampsia in the findings.

R2: Whilst we only referred to a “preeclampsia-like” phenotype, based on molecular and physiological datasets, we have now removed reference to preeclampsia-like throughout most of the manuscript including the abstract and discussion. Instead we refer to a more general ‘placental insufficiency’ phenotype, which is strongly supported. We do however, raise the possibility that are data are consistent with a “pre-eclampsia-*like*” phenotype since it is important to consider all possibilities.

Referee #4 (Remarks to the Author):

C1: The authors fully addressed all my concerns and I'm happy to recommend its publication.

R1: We are grateful for the thoughtful discussions and comments on our study over the review period.